# SPARSE DISTRIBUTED MEMORY IS A CONTINUAL LEARNER

**Trenton Bricken**
Systems, Synthetic, and Quantitative Biology
Harvard University
`trentonbricken@g.harvard.edu`

**Xander Davies & Deepak Singh**
Computer Science
Harvard College
`{alexander_davies, tejasvisingh}@college.harvard.edu`

**Dmitry Krotov**
MIT-IBM Watson AI Lab
IBM Research
`krotov@ibm.com`

**Gabriel Kreiman**
Harvard Medical School
Programs in Biophysics and Neuroscience
`Gabriel.Kreiman@childrens.harvard.edu`

## ABSTRACT

Continual learning is a problem for artificial neural networks that their biological counterparts are adept at solving. Building on work using Sparse Distributed Memory (SDM) to connect a core neural circuit with the powerful Transformer model, we create a modified Multi-Layered Perceptron (MLP) that is a strong continual learner. We find that every component of our MLP variant translated from biology is necessary for continual learning. Our solution is also free from any memory replay or task information, and introduces novel methods to train sparse networks that may be broadly applicable.

## 1 INTRODUCTION

Biological networks tend to thrive in continually learning novel tasks, a problem that remains daunting for artificial neural networks. Here, we use Sparse Distributed Memory (SDM) to modify a Multi-Layered Perceptron (MLP) with features from a cerebellum-like neural circuit that are shared across organisms as diverse as humans, fruit flies, and electric fish (Modi et al., 2020; Xie et al., 2022). These modifications result in a new MLP variant (referred to as SDMLP) that uses a Top-K[1] activation function (keeping only the $k$ most excited neurons in a layer on), no bias terms, and enforces both $L^2$ normalization and non-negativity constraints on its weights and data. All of these SDM-derived components are necessary for our model to avoid catastrophic forgetting.

We encounter challenges when training the SDMLP that we leverage additional neurobiology to solve, resulting in better continual learning performance. Our first problem is with "dead neurons" that are never active for any input and which are caused by the Top-K activation function (Makhzani & Frey, 2014; Ahmad & Scheinkman, 2019). Having fewer neurons participating in learning results in more of them being overwritten by any new continual learning task, increasing catastrophic forgetting. Our solution imitates the "GABA Switch" phenomenon where inhibitory interneurons that implement Top-K will *excite* rather than *inhibit* early in development (Gozel & Gerstner, 2021).

The second problem is with optimizers that use momentum, which becomes "stale" when training highly sparse networks. This staleness refers to the optimizer continuing to update inactive neurons with an out-of-date moving average, killing neurons and again harming continual learning. To our

---

[1] Also called "k Winner Takes All" in related literature.

knowledge, we are the first to formally identify this problem that will in theory affect any sparse model, including recent Mixtures of Experts (Fedus et al., 2021; Shazeer et al., 2017).

Our SDMLP is a strong continual learner, especially when combined with complementary approaches. Using our solution in conjunction with Elastic Weight Consolidation (EWC) (Kirkpatrick et al., 2017), we obtain, to the best of our knowledge, state-of-the-art performance for CIFAR-10 in the class incremental setting when memory replay is not allowed. Another variant of SDM, developed independently of our work, appears to be state-of-the-art for CIFAR-100, MNIST, and FashionMNIST, with our SDMLP as a close second (Shen et al., 2021).

Excitingly, our continual learning success is "organic" in resulting from the underlying model architecture and does not require any task labels, task boundaries, or memory replay. Abstractly, SDM learns its subnetworks responsible for continual learning using two core model components. First, the Top-K activation function causes the $k$ neurons most activated by an input to specialize towards this input, resulting in the formation of specialized subnetworks. Second, the $L^2$ normalization and absence of a bias term together constrain neurons to the data manifold, ensuring that all neurons democratically participate in learning. When these two components are combined, a new learning task only activates and trains a small subset of neurons, leaving the rest of the network intact to remember previous tasks without being overwritten

As a roadmap of the paper, we first discuss related work (Section 2). We then provide a short introduction to SDM (Section 3), before translating it into our MLP (Section 4). Next we present our results, comparing the organic continual learning capabilities of our SDMLP against relevant benchmarks (Section 5). Finally, we conclude with a discussion on the limitations of our work, sparse models more broadly, and how SDM relates MLPs to Transformer Attention (Section 6).

## 2 RELATED WORK

**Continual Learning -** The techniques developed for continual learning can be broadly divided into three categories: architectural (Goodfellow et al., 2014; 2013), regularization (Smith et al., 2022; Kirkpatrick et al., 2017; Zenke et al., 2017; Aljundi et al., 2018), and rehearsal (Lange et al., 2021; Hsu et al., 2018). Many of these approaches have used the formation of sparse subnetworks for continual learning (Abbasi et al., 2022; Aljundi et al., 2019b; Ramasesh et al., 2022; Iyer et al., 2022; Xu & Zhu, 2018; Mallya & Lazebnik, 2018; Schwarz et al., 2021; Smith et al., 2022; Le et al., 2019; Le & Venkatesh, 2022).[2] However, in contrast to our "organic" approach, these methods employ complex algorithms and additional memory consumption to explicitly protect model weights important for previous tasks from overwrites.[3]

Works applying the Top-K activation to continual learning include (Aljundi et al., 2019b; Gozel & Gerstner, 2021; Iyer et al., 2022). Srivastava et al. (2013) used a local version of Top-K, defining disjoint subsets of neurons in each layer and applying Top-K locally to each. However, this was only used on a simple task-incremental two-split MNIST task and without any of the additional SDM modifications that we found crucial to our strong performance (Table 2).

The Top-K activation function has also been applied more broadly in deep learning (Makhzani & Frey, 2014; Ahmad & Scheinkman, 2019; Sengupta et al., 2018; Gozel & Gerstner, 2021; Aljundi et al., 2019b). Top-K converges with not only the connectivity of many brain regions that utilize inhibitory interneurons but also results showing advantages beyond continual learning, including: greater interpretability (Makhzani & Frey, 2014; Krotov & Hopfield, 2019; Grinberg et al., 2019), robustness to adversarial attacks (Paiton et al., 2020; Krotov & Hopfield, 2018; Iyer et al., 2022), efficient sparse computations (Ahmad & Scheinkman, 2019), tiling of the data manifold (Sengupta et al., 2018), and implementation with local Hebbian learning rules (Gozel & Gerstner, 2021; Sengupta et al., 2018; Krotov & Hopfield, 2019; Ryali et al., 2020; Liang et al., 2020).

**FlyModel -** The most closely related method to ours is a model of the *Drosophila* Mushroom Body circuitry (Shen et al., 2021). This model, referred to as "FlyModel", unknowingly implements the SDM algorithm, specifically the Hyperplane variant (Jaeckel, 1989a) with a Top-K activation

---

[2]Even without sparsity or the Top-K activation function, pretraining models can still lead to the formation of subnetworks, which translates into better continual learning performance as found in Ramasesh et al. (2022).

[3]Memory replay methods indirectly determine and protect weights by deciding what memories to replay.

function that we also use and will justify (Keeler, 1988). The FlyModel shows strong continual learning performance, trading off the position of best performer with our SDMLP across tasks and bolstering the title of our paper that SDM is a continual learner.

While both models are derived from SDM, our work both extends the theory of SDM by allowing it to successfully learn data manifolds (App. A.4), and reconciles the differences between SDM and MLPs, such that SDM can be trained in the deep learning framework (this includes having no fixed neuron weights and using backpropagation). Both of these contributions are novel and may inspire future work beyond continual learning. For example, learning the data manifold preserves similarity in the data and leads to more specialized, interpretable neurons (e.g. Fig. 26 of App. H).

Training in the deep learning framework also allows us to combine SDMLP with other gradient-based methods like Elastic Weight Consolidation (EWC) that the FlyModel is incompatible with (Kirkpatrick et al., 2017). Additionally, demonstrating how MLPs can be implemented as a cerebellar circuit serves as an example for how ideas from neuroscience, like the GABA switch, can be leveraged to the benefit of deep learning.

## 3 BACKGROUND ON SPARSE DISTRIBUTED MEMORY

Sparse Distributed Memory (SDM) is an associative memory model that tries to solve the problem of how patterns (memories) could be stored in the brain (Kanerva, 1988; 1993) and has with close connections to Hopfield networks, the circuitry of the cerebellum, and Transformer Attention (Kanerva, 1988; Bricken & Pehlevan, 2021; Hopfield, 1982; 1984; Krotov & Hopfield, 2016; Tyulmankov et al., 2021; Millidge et al., 2022). We briefly provide background on SDM and notation sufficient to relate SDM to MLPs. For a summary of how SDM relates to the cerebellum, see App. A.2. We use the continuous version of SDM, where all neurons and patterns exist on the $L^2$ unit norm hypersphere and cosine similarity is our distance metric (Bricken & Pehlevan, 2021).

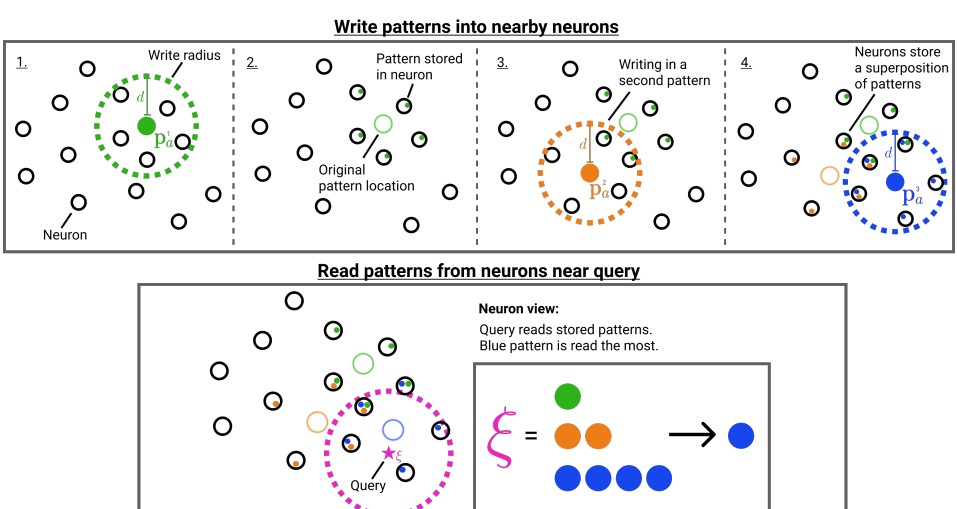

Figure 1: **Graphical Summary of the SDM Write and Read Operations. Top Row -** Three patterns being written into nearby neurons. 1. First write operation; 2. Patterns are stored inside neurons and the original pattern address is shown; 3. Writing a second pattern; 4. Writing a third pattern and neurons storing a superposition of multiple patterns. **Bottom Row -** The SDM read operation. The query reads from nearby neurons with the inset showing the number of times each pattern is read. Blue is read the most as its original pattern location (shown by the hollow circle) is closest to the query. Fig. adapted with permission from Bricken & Pehlevan (2021).

SDM randomly initializes the addresses of $r$ neurons on the $L^2$ unit hypersphere in an $n$ dimensional space. These neurons have addresses that each occupy a column in our address matrix $X_a \in (L^2)^{n \times r}$, where $L^2$ is shorthand for all $n$-dimensional vectors existing on the $L^2$ unit norm hypersphere. Each neuron also has a storage vector used to store patterns represented in the matrix

$X_v \in \mathbb{R}^{o \times r}$, where $o$ is the output dimension. Patterns also have addresses constrained on the $n$-dimensional $L^2$ hypersphere determined by their encoding; encodings can be as simple as flattening an image into a vector or as complex as preprocessing with a deep learning model.

Patterns are stored by activating all nearby neurons within a cosine similarity threshold $c$, and performing an elementwise summation with the activated neurons' storage vector. Depending on the task at hand, patterns write themselves into the storage vector (e.g., during a reconstruction task) or write another pattern, possibly of different dimension (e.g., writing in their one hot label for a classification task). This write operation and the ensuing read operation are summarized in Fig. 1.

Because in most cases we have fewer neurons than patterns, the same neuron will be activated by multiple different patterns. This is handled by storing the pattern values in superposition via the aforementioned elementwise summation operation. The fidelity of each pattern stored in this superposition is a function of the vector orthogonality and dimensionality $n$.

Using $m$ to denote the number of patterns, matrix $P_a \in (L^2)^{n \times m}$ for the pattern addresses, and matrix $P_v \in \mathbb{R}^{o \times m}$ for the values patterns want to write, the SDM write operation is:

$$X_v = P_v b(P_a^T X_a), \qquad b(e) = \begin{cases} 1, & \text{if } e \geq c \\ 0, & \text{else,} \end{cases} \tag{1}$$

where $b(e)$ performs an element-wise binarization of its input to determine which pattern and neuron addresses are within the cosine similarity threshold $c$ of each other.

Having written patterns into our neurons, we read from the system by inputting a query $\boldsymbol{\xi}$, that again activates nearby neurons. Each activated neuron outputs its storage vector and they are all summed elementwise to give a final output $\mathbf{y}$. The output $\mathbf{y}$ can be interpreted as an updated query and optionally $L^2$ normalized again as a post processing step:

$$\mathbf{y} = X_v b(X_a^T \boldsymbol{\xi}). \tag{2}$$

Intuitively, SDM's query will update towards the values of the patterns with the closest addresses. This is because the patterns with the closest addresses will have written their values into more neurons that the query reads from than any competing patterns. For example, in Fig. 1, the blue pattern address is the closest to the query meaning that it appears the most in those nearby neurons the query reads from. SDM is closely related to modern Hopfield networks (Krotov & Hopfield, 2016; Ramsauer et al., 2020; Krotov & Hopfield, 2020) if they are restricted to a single step update of the recurrent dynamics (Tyulmankov et al., 2021; Bricken & Pehlevan, 2021; Millidge et al., 2022).

## 4 TRANSLATING SDM INTO MLPS FOR CONTINUAL LEARNING

A one hidden layer MLP transforms an input $\boldsymbol{\xi}$ to an output $\mathbf{y}$. Using notation compatible with SDM and representing unnormalized continuous vectors with a tilde, we can write the MLP as:

$$\mathbf{y} = \tilde{X}_v f(\tilde{X}_a^T \tilde{\boldsymbol{\xi}} + \tilde{\mathbf{b}}_a) + \tilde{\mathbf{b}}_v, \tag{3}$$

where $\tilde{X}_a \in \mathbb{R}^{n \times r}$ and $\tilde{X}_v \in \mathbb{R}^{o \times r}$ are weight matrices corresponding to our SDM neuron addresses and values, respectively. Meanwhile, $\tilde{\mathbf{b}}_a \in \mathbb{R}^r$, $\tilde{\mathbf{b}}_v \in \mathbb{R}^o$ are the bias parameters and $f(\cdot)$ is the activation function used by the MLP such as ReLU (Glorot et al., 2011).

Using this SDM notation for the MLP, it is trivial to see the similarity between the SDM read Eq. 2 and Eq. 3 for single hidden layer MLPs that was first established in (Kanerva, 1993). However, the fixed random neuron address of SDM makes it unable to effectively model real-world data. Using ideas from Keeler (1988), we resolve this inability with a biologically plausible inhibitory interneuron that is approximated by a Top-K activation function. App. A.3 explains how SDM is modified and the actual Top-K activation function used is presented shortly in Eq. 4. This modification makes SDM compatible with an MLP that has no fixed weights. As for the SDM write operation of Eq. 1, this is related to an MLP trained with backprop as outlined in App. A.6.

**The Dead Neuron Problem -** An issue with the Top-K activation function is that it creates dead neurons (Ahmad & Scheinkman, 2019; Rumelhart & Zipser, 1985; Makhzani & Frey, 2014; Fedus

et al., 2021). Only a subset of the randomly initialized neurons will exist closest to the data manifold and be in the top $k$ most active, leaving all remaining neurons to never be activated. In the continual learning setting, when a new task is introduced, the model has fewer neurons that can learn and must overwrite more that were used for the previous task(s), resulting in catastrophic forgetting.

Rather than pre-initializing our neuron weights using a decomposition of the data manifold (Rumelhart & Zipser, 1985; McInnes & Healy, 2018; Strang, 1993) we ensure that all neurons are active at the start of training so that they update onto the manifold. This approach has been used before but in biologically implausible ways (see App. B.1) (Makhzani & Frey, 2014; Ahmad & Scheinkman, 2019; van den Oord et al., 2017). We instead extend the elegant solution of Gozel & Gerstner (2021) by leveraging the inhibitory interneuron we have already introduced. After neurogenesis, when neuronal dendrites are randomly initialized, neurons are excited by the inhibitory GABA neurotransmitter (see App. B.2). This means that the same inhibitory interneuron used to enforce competition via Top-K inhibition at first creates cooperation by exciting every neuron, allowing them all to learn. As a result, every neuron first converges onto the data manifold at which point it is inhibited by GABA and Top-K competition begins, forcing each neuron to specialize and tile the data manifold.

We present the full GABA switch implementation in App. B.3 that is the most biologically plausible. However, using the positive weight constraint that makes all activations positive, we found a simpler approximation that gives the same performance. This is by linearly annealing our $k$ value from the maximum number of neurons in the layer down to our desired $k$. A similar approach without the biological justification is used by Makhzani & Frey (2014) but they zero out all values that are not the $k$ most active and keep the remaining $k$ untouched. Motivated by our inhibitory interneuron, we instead subtract the $k + 1$-th activation from all neurons that remain on. This change leads to a significant boost in continual learning (see Table 2 and App. B.3). Formally, letting the neuron activations pre inhibition be $\mathbf{a} := X_a^T \boldsymbol{\xi}$ and $\mathbf{a}^*$ post inhibition:

$$\mathbf{a}^* = [\mathbf{a} - I]_+ \tag{4}$$
$$I = \text{descending-sort}([\mathbf{a}]_+)_{(k_t+1)}$$
$$k_t = \max\left(k_{\text{target}}, \lfloor k_{\max} - E_t(k_{\max} - k_{\text{target}})/s \rfloor\right),$$

where $[\cdot]_+$ is the ReLU operation, $\lfloor \cdot \rfloor$ is the floor operator to ensure $k_t$ is an integer, and descending-sort$(\cdot)$ sorts the neuron activations in descending order to find the $k + 1$-th largest activation. $E_t$ is an integer representing the current training epoch and subscript $t$ denotes that $k_t$ and $E_t$ change over time. Hyperparameter $s$ sets the number of epochs for $k$ to go from its starting value $k_{\max}$, to its target value $k_{\text{target}}$. When $k_t = k_{\text{target}}$ our approximated GABA switch is fully inhibitory for every neuron. An algorithm box in App. A.1 summarizes how each SDMLP component functions.

**The Stale Momentum Problem -** We discovered that even using the GABA switch, some optimizers continue killing off a large fraction of neurons, to the detriment of continual learning. Investigating why, we have coined the "stale momentum" problem where optimizers that utilize some form of momentum (especially Adam and RMSProp) fail to compute an accurate moving average of previous gradients in the sparse activation setting (Kingma & Ba, 2015; Geoffrey Hinton, 2012). Not only will these momentum optimizers update inactive neurons not in the Top-K, but also explode gradient magnitudes when neurons become activated after a period of quiescence. We explain and investigate stale momenta further in App. D and use SGD without momentum as our solution.

**Additional Modifications -** There are five smaller discrepancies (expanded upon in App. A.5) between SDM and MLPs: (i) using rate codes to avoid non differentiable binary activations; (ii) using $L^2$ normalization of inputs and weights as an approximation to contrast encoding and heterosynaptic city (Sterling & Laughlin, 2015; Rumelhart & Zipser, 1985; Tononi & Cirelli, 2014); (iii) removing bias terms which assume a tonic baseline firing rate not present in cerebellar granule cells (Powell et al., 2015; Giovannucci et al., 2017); (iv) using only positive (excitatory) weights that respect Dale's Principle (Dale, 1935); (v) using backpropagation as a more efficient (although possibly non-biological (Lillicrap et al., 2020)) implementation of Hebbian learning rules (Krotov & Hopfield, 2019; Sengupta et al., 2018; Gozel & Gerstner, 2021).

## 5 SDM Avoids Catastrophic Forgetting

Here we show that the proposed SDMLP architecture results in strong and organic continual learning. In modifying only the model architecture, our approach is highly compatible with other contin-

ual learning strategies such as regularization of important weights (Kirkpatrick et al., 2017; Aljundi et al., 2018; Zenke et al., 2017) and memory replay (Zhang et al., 2021; Hsu et al., 2018), both of which the brain is likely to use in some fashion. We show that combinations of SDMLP with weight regularization are positive sum and don't explore memory replay.

**Experimental Setup -** Trying to make the continual learning setting as realistic as possible, we use Split CIFAR10 in the class incremental setting with pretraining on ImageNet (Russakovsky et al., 2015). This splits CIFAR10 into disjoint subsets that each contain two of the classes. For example the first data split contains classes 5 and 2 the second split contains classes 7 and 9, etc. CIFAR is more complex than MNIST and captures real-world statistical properties of images. The class incremental setting is more difficult than incremental task learning because predictions are made for every CIFAR class instead of just between the two classes in the current task (Hsu et al., 2018; Farquhar & Gal, 2018). Pretraining on ImageNet enables learning general image statistics and is when the GABA switch happens, allowing neurons to specialize and spread across the data manifold. In the main text, we present results where our ImageNet32 and CIFAR datasets have been compressed into 256 dimensional latent embeddings taken from the last layer of a frozen ConvMixer that was pre-trained on ImageNet32 (step #1 of our training regime in Fig. 2) (Trockman & Kolter, 2022; Russakovsky et al., 2015).[4]

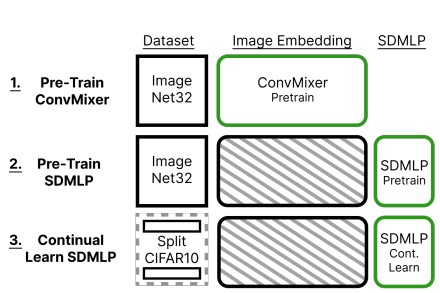

| Method | Neurons | $k$ | Val. Acc. |
|---|---|---|---|
| SDMLP | 10K | 10 | 0.71 |
| FlyModel | 10K | 32 | 0.82 |
| MAS | 10K | NA | 0.67 |
| SI | 10K | NA | 0.44 |
| ReLU | 10K | NA | 0.21 |
| EWC | 10K | NA | 0.65 |
| SDMLP+MAS | 10K | 10 | 0.84 |
| SDMLP+EWC | 10K | 10 | **0.86** |
| Oracle | 10K | NA | 0.93 |

Figure 2: **Left**, the three-step training regime. Green outlines denote the module currently being trained. Gray hatches on the ConvMixer denote when it is frozen. Table 1: **Right**, Split CIFAR10 final validation accuracy. We highlight the best performing SDMLP, baseline, and **overall performer** in the 10K neuron setting. Oracle was trained on the full CIFAR10 dataset. All results are the average of 5 random task splits.

Using the image embeddings, we pre-train our models on ImageNet (step #2 of Fig. 2). We then reset $X_v$ that learns class labels and switch to continual learning on Split CIFAR10, training every model for 2,000 epochs on each of the five data splits (step #3 of Fig. 2). We use such a large number of epochs to encourage forgetting of previous tasks. We test model performance on both the current data split and every data split seen previously to assess forgetting. The CIFAR dataset is split into disjoint sets using five different random seeds to ensure our results are independent of both data split ordering and the classes each split contains.

We use preprocessed image embeddings because single hidden layer MLPs struggle to learn directly from image pixels. This preprocessing is also representative of the visual processing stream that compresses images used by deeper brain regions (Pisano et al., 2020; Li et al., 2020; Yamins et al., 2014). However, we consider two ablations that shift the validation accuracies but not the rank ordering of the models or conclusions drawn: (i) Removing the ConvMixer embeddings and instead training directly on image pixels (removing step #1 of Fig. 2, App. E.1); (ii) Testing continual learning without any ImageNet32 pre-training (removing step #2 of Fig. 2, App. E.2). We also run experiments for CIFAR100 (App. F.2), MNIST (App. F.3) and FashionMNIST (App. F.4).

**Model Parameters and Baselines -** All models are MLPs with 1,000 neurons in a single hidden layer unless otherwise indicated. When using the Top-K activation function, we set $k_{\text{target}} = 10$ and also present $k_{\text{target}} = 1$. We tested additional $k$ values and suggest how to choose the best $k_{\text{target}}$ value

---

[4]This version of ImageNet has been re-scaled to be 32 by 32 dimensions and is trained for 300 epochs (Russakovsky et al., 2015).

in App. C.2. Because the $k$ values considered are highly sparse – saving on FLOPs and memory consumption – we also evaluate the 10,000 neuron setting which improves SDM and the FlyModel continual learning abilities in particular.

The simplest baselines we implement are the ReLU activation function, L2 regularization, and Dropout, that are all equally organic in not using any task information (Goodfellow et al., 2014). We also compare to the popular regularization based approaches Elastic Weight Consolidation (EWC), Memory Aware Synapses (MAS), and Synaptic Intelligence (SI) (Kirkpatrick et al., 2017; Aljundi et al., 2018; Zenke et al., 2017; Smith et al., 2022). These methods infer model weights that are important for each task and penalize updating them by using a regularization term in the loss. To track each parameter, this requires at least doubling memory consumption plus giving task boundaries and requiring a coefficient for the loss term.[5]

In the class incremental learning setting, it has been shown that these regularization methods catastrophically forget (Hsu et al., 2018; Gurbuz & Dovrolis, 2022). However, we found these results to be inaccurate and get better performance through careful hyperparameter tuning. This included adding a $\beta$ coefficient to the softmax outputs of EWC and SI that alleviates the problem of vanishing gradients (see App. G.1).We are unaware of this simple modification being used in prior literature but use it to make our baselines more challenging.

We also compare to related work that emphasizes biological plausibility and sparsity. The FlyModel performs very well and also requires no task information (Shen et al., 2021). Specific training parameters used for this model can be found in App. G.2. Additionally, we test Active Dendrites (Iyer et al., 2022) and NISPA (Neuro-Inspired Stability-Plasticity Adaptation) (Gurbuz & Dovrolis, 2022). However, these models both use the easier task incremental setting and failed to generalize well to the class incremental setting, especially Active Dendrites, on even the simplest Split MNIST dataset (App. F.3).[6] The failure for task incremental methods to apply in a class incremental setting has been documented before (Farquhar & Gal, 2018). These algorithms are also not organic, needing task labels and have significantly more complex, less biologically plausible implementations. We considered a number of additional baselines but found they were all were lacking existing results for the class incremental setting and often lacked publicly available codebases.[7]

**CIFAR10 Results -** We present the best continual learning results of each method in Fig. 3 and its corresponding Table 1. SDMLP organically outperforms all regularization baselines, where its best validation accuracy is 71% compared to 67% for MAS. In the 1K neuron setting (shown in Table 5 of App. F.1) SDMLP with $k = 1$ ties with the FlyModel at 70%. In the 10K neuron setting, the FlyModel does much better achieving 82% versus 71% for SDMLP. However, the combination of SDMLP and EWC leads to the strongest performance of 86%, barely forgetting any information compared to an oracle that gets 93% when trained on all of CIFAR10 simultaneously.[8]

SDMLP and the regularization based methods are complimentary and operate at different levels of abstraction. SDMLP learns what subset of neurons are important and the regularization method learns what subset of weights are important. Because the FlyModel combines fixed weights with Top-K, it can be viewed as a more rigid form of the SDMLP and regularization method combined. The FlyModel also benefits from sparse weights making neurons more orthogonal to respond uniquely to different tasks. We attempted to integrate this weight sparsity into our SDMLP so that our neurons could learn the data manifold but be more constrained to a data subset, hypothetically producing less cross-talk and forgetting between tasks. Interestingly, initial attempts to prune weights either randomly or just the smallest from the ImageNet pretrained models resulted in catastrophic forgetting. We leave more in depth explorations of weight sparsity to future work.

**Ablations -** We ablate components of our SDMLP to determine which contribute to continual learning in Table 2. Deeper analysis of why each ablation fails is provided in App. H but we highlight

---

[5]We acknowledge that task boundaries can be inferred in an unsupervised fashion but this requires further complexity and we expect the best performance to come from using true task boundaries (Aljundi et al., 2019a).

[6]Because we do not test memory replay approaches, we ignore the modified version of NISPA developed for the class incremental setting (Gurbuz & Dovrolis, 2022).

[7]All of our code and training parameters can be found in our publicly available codebase: `https://github.com/TrentBrick/SDMContinualLearner`.

[8]We found that SGDM for the combined SDMLP+EWC does slightly better than SGD and is used here while SGD worked the best for all other approaches.

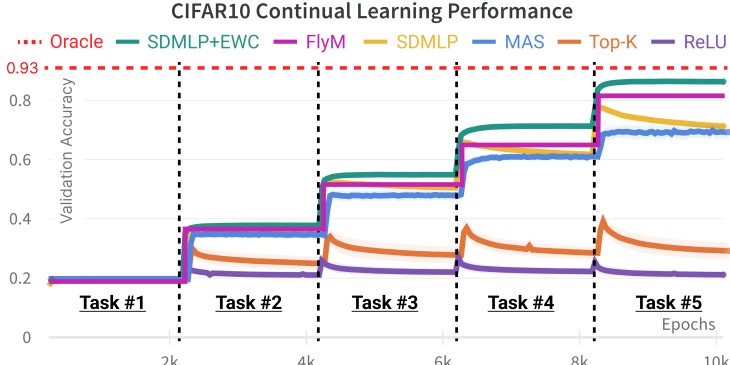

Figure 3: **SDM outperforms regularization methods and their combination is positive sum.** We plot validation accuracy across tasks for the best performing methods and baselines on Split CIFAR10. SDMLP+EWC (green line) does the best, then the FlyModel (magenta), then SDMLP (yellow) with MAS (blue) close behind. The Top-K (orange) without SDM modifications and ReLU (purple) baselines do poorly. The FlyModel was only trained for one epoch on each task as per Shen et al. (2021) but we visually extend the validation accuracy on each task to make method comparison easier. App. F.1 visualizes how SDMLP gradually forgets each task compared to catastrophic forgetting of the ReLU baseline. We use the average of 5 random seeds and error bars to show standard error of the mean but the variance is small making them hard to see.

the two most important here: First, the Top-K activation function prevents more than $k$ neurons from updating for any given input, restricting the number of neurons that can forget information. A small subset of the neurons will naturally specialize to the new data classes and continue updating towards it, protecting the remainder of the network from being overwritten. This theory is supported by results in App. C.2 where the smaller $k$ is, the less catastrophic forgetting occurs. We provide further support for the importance of $k$ by showing the number of neurons activated when learning each task in App. H.

Second, the $L^2$ normalization constraint and absence of a hidden layer bias term are crucial to keeping all neurons on the $L^2$ hypersphere, ensuring they all participate democratically. Without both these constraints, but still using the Top-K activation function, we found that a small subset of the neurons become "greedy" in having a larger weight norm or bias term and always out-compete the other neurons. This results in a small number of neurons updating for every new task and catastrophically forgetting the previous one as shown in the Table 2 ablations. Evidence of manifold tiling is shown in Fig. 6 with a UMAP (McInnes & Healy, 2018) plot fit on the SDM weights (orange) that were pre-trained on ImageNet32 embeddings. We project the embedded CIFAR10 training data (blue) to show that pre-training has given SDM subnetworks that cover the manifold of general image statistics contained in ImageNet32. This manifold tiling is in sharp contrast to the other methods shown in App. H and enables SDM to avoid catastrophic forgetting. As further evidence of manifold tiling, we show in the same appendix that SDM weights are often highly interpretable when trained directly on CIFAR10 images.

As an additional ablation to our SDMLP defined by Eq. 2, we find that introducing a bias term in the output layer breaks continual learning. This is because the model assigns large bias values to the classes within the current task over any classes in previous tasks, giving the appearance of catastrophic forgetting. Interestingly, this effect only applies to SDM; modifying all of our other baselines by removing the output layer bias term fails to affect their continual learning performance.

## 6 DISCUSSION

Setting out to implement SDM as an MLP, we introduce a number of modifications resulting in a model capable of continual learning. These modifications are biologically inspired, incorporating components of cerebellar neurobiology that are collectively necessary for strong continual learning.

| Name | Val. Acc. |
|------|-----------|
| SGD | 0.63 |
| SGDM | 0.54 |
| Adam | 0.23 |
| RMSProp | 0.20 |
| Linear Subtract | 0.63 |
| Linear Mask | 0.38 |
| No Linear Anneal | 0.35 |
| Negative Weights | 0.57 |
| No $L^2$ Norm | 0.20 |
| Hidden Layer Bias Term | 0.20 |
| Output Layer Bias Term | 0.20 |

Table 2: **SDMLP Ablations.** Validation accuracy for ablated versions of SDMLP on CIFAR10 embeddings and $k = 10$ with 1,000 neurons (the same task as Table 1). See main text for details.

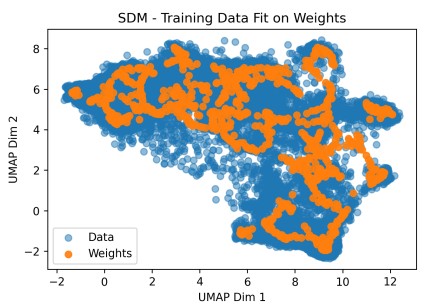

Fig. 6: **SDM learns the data manifold.** UMAP representation of CIFAR10 embeddings (blue) and SDM weights (orange) pretrained on ImageNet32. The overlap between weights and data is lacking in other methods (see App. H).

Our new solution to the dead neuron problem and identification of the stale momentum problem are key to our results and may further generalize to other sparse neural network architectures.

Being able to write SDM as an MLP with minor modifications is interesting in light of SDM's connection to Transformer Attention (Bricken & Pehlevan, 2021). This link converges with work showing that Transformer MLP layers perform associative memory-like operations that approximate Top-K by showing up to 90% activation sparsity in later layers (Geva et al., 2020; Sukhbaatar et al., 2019; Nelson et al., 2022). Viewing both Attention and MLPs through the lens of SDM presents their tradeoffs: Attention operates on patterns in the model's current receptive field. This increases the fidelity of both the write and read operations because patterns do not need to rely on neurons that poorly approximate their original address location and store values in a noisy superposition. However, in the MLP setting, where patterns are stored in and read from neurons, an increase in noise is traded for being able to store patterns beyond the current receptive field. This can allow for learning statistical regularities in the patterns' superpositions to represent useful prototypes, likely benefiting generalization (Irie et al., 2022).

**Limitations -** Our biggest limitation remains ensuring that SDM successfully avoids the dead neuron problem and tiles the data manifold in order to continually learn. Setting $k_{\text{target}}$ and $s$ decay rate remain difficult and data manifold dependent (App. C.2). Avoiding dead neurons currently requires training SDM on a static data manifold, whether it is image pixels or a fixed embedding. Initial experiments jointly training SDM modules either interleaved throughout a ConvMixer or placed at the end resulted in many dead neurons and failure to continually learn. We believe this is in large part due to the manifold continuing to change over time.

Constraining neurons to exist on the data manifold and only allowing a highly sparse subset to fire is also suboptimal for maximizing classification accuracy in non continual learning settings. Making $k$ sufficiently large can alleviate this problem but at the cost of continual learning abilities. However, the ability for hardware accelerators to leverage sparsity should allow for networks to become wider (increasing $r$), rather than decreasing $k$, and help alleviate this issue (Wang, 2020; Gale et al., 2020). Finally, while our improved version of SDM assigns functionality to five different cell types in the cerebellar circuit, there is more of the circuitry to be mapped such as Basket cells and Deep Cerebellar Nuclei (Sezener et al., 2021).

**Conclusion -** We have shown that SDM, when given the ability to model correlated data manifolds in ways that respect cerebellar neurobiology, is a strong continual learner. This result is striking because every component of the SDMLP, including $L^2$ normalization, the GABA switch, and momentum free optimization are necessary for these continual learning results. Beyond setting a new "organic" baseline for continual learning, we help establish how to train sparse models in the deep learning setting through solutions to the dead neuron and stale momentum problems. More broadly, this work expands the neurobiological mapping of SDM to cerebellar circuitry and relates it to MLPs, deepening links to the brain and deep learning.

## ACKNOWLEDGEMENTS

Thanks to Dr. Beren Millidge, Joe Choo-Choy, Miles Turpin, Blake Bordelon, Stephen Casper, Dr. Mengmi Zhang, Dr. Tomaso Poggio, and Dr. Pentti Kanerva for providing invaluable inspiration, discussions, and feedback. We would also like to thank the open source software contributors that helped make this research possible, including but not limited to: Numpy, Pandas, Scipy, Matplotlib, PyTorch, and Anaconda.

## AUTHOR CONTRIBUTIONS

- Trenton Bricken conceived of the project and theory, conducted all experiments, and wrote the paper with suggestions from co-authors.
- Xander Davies participated in discussions, contributed to early investigations of Dead Neurons and Stale Momentum, and reviewed related work.
- Deepak Singh reviewed robustness and Top-K related work and participated in discussions.
- Dmitry Krotov advised on the continual learning experiments, Top-K learning theory, and relations to associative memory models.
- Gabriel Kreiman supervised the project providing guidance throughout on theory and experiments.

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

# App.

## Table of Contents

## A  SDM

### A.1  SDM TRAINING ALGORITHM

Here we present in full the training loop used for the SDMLP.

---

**Algorithm 1** SDMLP Training Algorithm

---

**Input:** Model weights: $(X_a, X_v)$, Tasks: $(T_1, \ldots, T_n)$, Top-K params: $k_{\max}$, $k_{\text{target}}$, Train steps: $s$
**Output:** Trained model weights: $(X_a, X_v)$
$X_v \leftarrow \text{LayerInit}(X_v)$                    // Reset neuron output weights
**for** *i in (1,...,n)* **do**
    $\mathbf{X}, \mathbf{Y} \leftarrow T_i$          // Obtain data and class labels for current task
    **for** *j in (1,...,s)* **do**
        $\mathbf{x}, \mathbf{y} \sim \mathbf{X}, \mathbf{Y}$                    // Sample a data point from the task
        $\mathbf{x} \leftarrow \mathbf{x}/\|\mathbf{x}\|$                           // L2 normalize the data
        $\mathbf{a} \leftarrow X_a^T \mathbf{x}$                            // Get neuron activations
        $\mathbf{a} \leftarrow \text{Top-K}(\mathbf{a})$                               // Apply Top-K Eq.4
        $\hat{\mathbf{y}} \leftarrow X_v^T \mathbf{a}$                            // Get model predictions
        $\nabla_\theta X_a, \nabla_\theta X_v \leftarrow \text{ComputeGradients}(\hat{\mathbf{y}}, \mathbf{y})$     // Compute Loss and Gradients
        $X_a, X_v \leftarrow \text{GradientStep}(\nabla_\theta X_a, \nabla_\theta X_v)$            // Update model weights
        $X_a, X_v \leftarrow [X_a]_+, [X_v]_+$           // Clamp all weights to be positive
        $X_a \leftarrow X_a/\|X_a\|$              // L2 normalize the neuron addresses
    **end**
**end**
**return** $X_a, X_v$

---

## A.2 SDM Biological Plausibility

Here we summarize the biological foundations of SDM.[9] The biological plausibility of our new modifications to SDM that allow it to learn the data manifold and be implemented as a deep learning model can be found in App. A.5. The biological plausibility of the GABA switch can be found in App. B.2.

Fig. 5 overlays SDM notation and operations on the cerebellar circuitry. Mossy Fibers represent incoming queries $\boldsymbol{\xi}$ and pattern addresses $\mathbf{p}_a$ through their firing activity. Each Granule cell represents an SDM neuron $\mathbf{x}$, with its dendrites as the neuron address vector $\mathbf{x}_a$ and post-synaptic connections with Purkinje cells as the storage vector $\mathbf{x}_v$. In branching perpendicularly, each Granule cells axon (called a Parallel Fiber), forms contacts with thousands of Purkinje cells with many granule cells (Hoxha et al., 2016). The strength of each contact represents a counter recording the value of a particular element in the neuron storage vector $(\mathbf{x}_v)_i$, where $i$ indexes this vector element.

During SDM read operations, this Purkinje cell can thus sum over index $i$ of all activated neuron storage vectors and use its firing rate to represent the value of $\mathbf{y}_i$. During SDM write operations, the Climbing Fibers are used to update the Purkinje cell counter of each active neuron with the pattern value at this index position $(\mathbf{p}_v)_i$. The most demanding part of SDM's implementation in a biological circuit that is strikingly satisfied by the cerebellum is the three way interface and specificity that exists between Purkinje cells, Climbing Fibers, and Parallel Fibers. This allows for Granule cells to have storage vectors that are precisely written to and read from. We also provide Fig. 6 to give an additional perspective on the biological mapping.

While our improved version of SDM assigns functionality to five different cell types in the cerebellar circuit, there is more of the circuitry to be mapped such as the Basket cells and Deep Cerebellar Nuclei (Sezener et al., 2021). There is also additional neuroscientific evidence required to confirm that the cerebellum operates as SDM predicts, for example that granule cells fire sparsely (Kanerva, 1988; Bricken & Pehlevan, 2021; Giovannucci et al., 2017). Within the scope of our model, the weakest biological plausibility is how well Top-K actually approximates the Golgi inhibitory interneuron (App. C.1) and how $L^2$ normalization is implemented (App. A.5).

---

[9]A more extensive treatment of how SDM may be implemented by the cerebellum can be found in Chapter 9 of the SDM book (Kanerva, 1988) and (Kanerva, 1993).

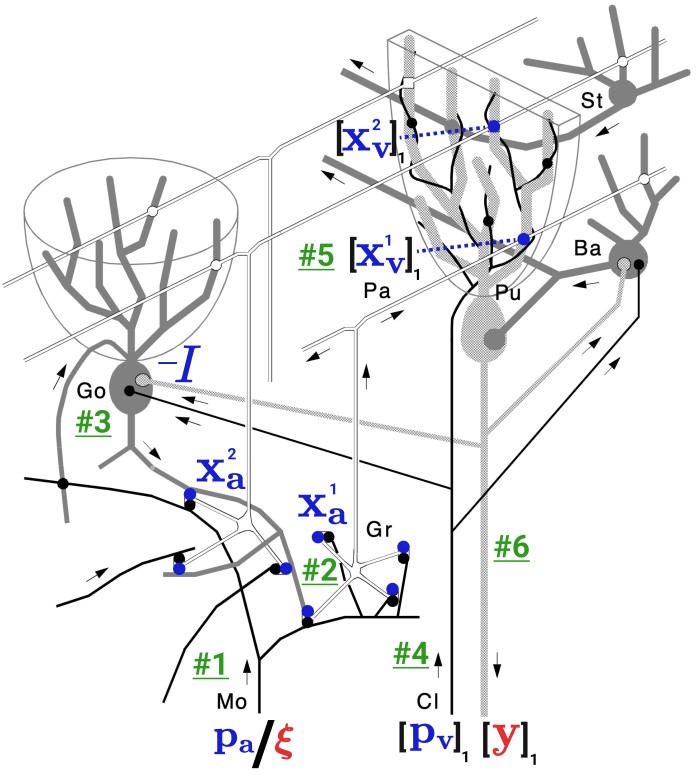

Figure 5: **SDM Mapping to the Cerebellum.** The operations are as follows: #1. A pattern $\mathbf{p}_a$ (blue) or query $\boldsymbol{\xi}$ (red) enters via firing of the Mossy Fibers (Mo). #2. Mossy fibers activate Granule Cells (Gr), with the Gr dendrites representing $\mathbf{x}_a$. Here we show two different Granule cells and their addresses $\mathbf{x}_a^1$ and $\mathbf{x}_a^2$. These Granule Cells project their axons (called Parallel Fibers (Pa)) upwards and across Purkinje Cells (Pu) where their synaptic connections encode their storage vectors. Each contact with a Purkinje Cell encodes a single element of the storage vector. Here we show one of these elements with the first Purkinje Cell with the subscript 1, $[\mathbf{x}_v^1]_1$ and $[\mathbf{x}_v^2]_1$. We will come back to these. #3. A Golgi (Go) inhbitory interneuron gets input from a number of neuron types including Granule cells, it inhibits these Granule cells with a value of $-I$, implementing an approximate Top-K. #4. If and only if we are writing in a pattern $\mathbf{p}_v$, it comes through a climbing fiber (Cl) that wraps around a single Purkinje Cell to produce large action potentials that induce long term potentiation/depression (LTP/LTD). Here we show the Climbing fiber that writes to the Purkinje Cell representing the first element of the storage vectors with $[\mathbf{p}_v]_1$. #5. The Granule Cells that are firing excite the Purkinje cells in proportion to the strength of their synaptic connections that represent their storage vectors. In the case of writing, this updates these synapses. #6. If we are reading from the system, the Purkinje Cell integrates signals from all activate neurons' storage vectors and determines whether or not to fire, implementing a non-linear activation function that outputs $[\mathbf{y}]_1$ (the first element of the vector in this case). The neuron types not mentioned previously are, going clockwise starting in the top right: St=Stellate cell (inhibitory interneuron) and Ba=Basket cell (inhibitory interneuron). Displayed originally without the SDM notation overlayed as Fig. 3.11 of (Kanerva, 1993), reprinted with permission from Pentti Kanerva to whom all rights belong.

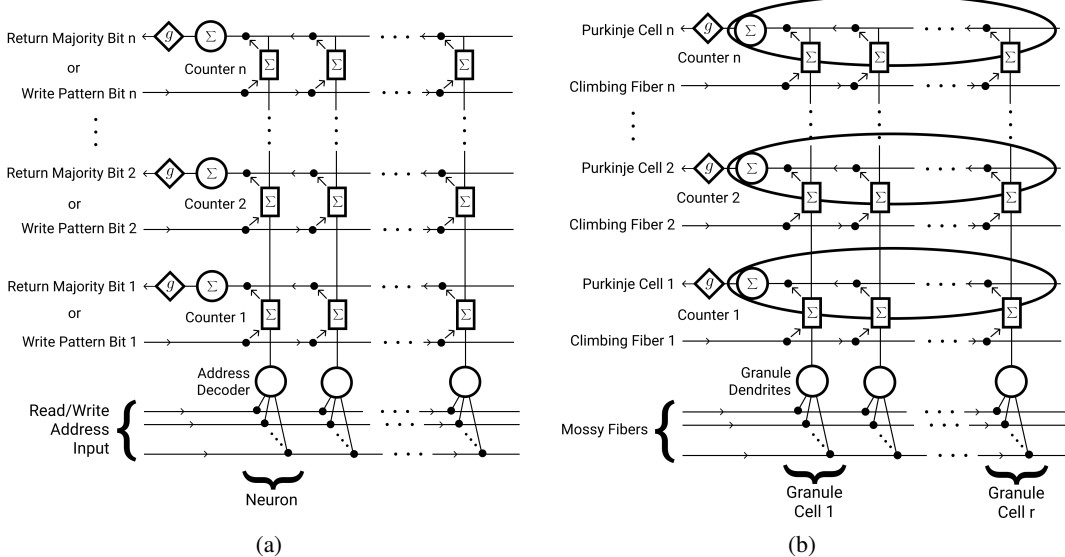

(a)                                                          (b)

Figure 6: **Another perspective on the mapping from SDM to the cerebellar circuitry.** **(a)** The computational circuitry necessary to implement SDM. **(b)** The biological implementation of SDM where the labels of (a) have been replaced by their biological components. The Purkinje cell is shown as an oval to represent the synaptic contacts with the granule cells. See the text for explanations of the architectural features.

### A.3 LEARNING SDM NEURON ADDRESSES WITH THE TOP-K ACTIVATION FUNCTION

SDM was built on the assumption that its neuron addresses, $X_a$, are randomly initialized and fixed in place (Kanerva, 1988). This was for reasons of both biological plausibility and analytical tractability when determining the maximum memory capacity and convergence dynamics of the model (see App. A.4). However, these random and fixed neuron addresses will often not be within cosine similarity $c$ of the real-world patterns existing on some lower dimensional manifold in the vector space. As a result, these neurons will often never be written to or read from.

Keeler (1988) outlined how SDM could both retain its theoretical results and allow for neuron addresses to learn the data manifold. This is by replacing the fixed cosine similarity activation threshold $c$ that activates a variable number of neurons, with a variable threshold that activates a fixed $k$ neurons. Intuitively, with random neuron and pattern addresses, a fixed $c$ would result in a pattern or query activating some $k$ neurons in expectation, keeping neuron utilization constant. However, using the same fixed $c$ with non-random addresses would vary the number of neurons being activated and utilized. Over-utilized neurons would store too many pattern values in their storage vector superposition, harming pattern fidelity. Instead, we can achieve the same constant neuron utilization as in the random address setting by varying $c$ such that only $k$ neurons are always activated.

How can $c$ be dynamically adjusted to keep a constant $k$ neurons active in a biologically plausible fashion? Via an inhibitory interneuron that creates a negative feedback loop: the more neurons that are active, the more activated the interneuron becomes, and the more it inhibits, keeping only $k$ active at convergence.[10] Sparsity inducing interneurons are ubiquitious across layers of cortical processing in the brain, particularly relevant to SDM is the cerebellar Golgi interneuron (Marr, 1969; Keeler, 1988; Paiton et al., 2020).

### A.4 WHY SDM ORIGINALLY REQUIRED FIXED NEURON ADDRESSES

In attempting to be biologically plausible, SDM was designed to respect Dale's Principle, whereby a synaptic connection can be either excitatory or inhibitory but not transition between them (Dale, 1935). Using the original binary vector formulation, neurons could compute if they were near enough to an incoming pattern or query to be activated (within some Hamming distance threshold) through the following steps:

1. Converting the neuron's binary address to bipolar weights $\{+1, -1\}$ corresponding to being excitatory or inhibitory, respectively.

2. Activating those weights where the incoming query has a 1 in its address and summing over them. This corresponds to how many 1s in the query and neuron address vectors agree – if the addresses match then an excitatory weight is activated, giving a +1, if they disagree then a negative weight is activated, giving a -1.

3. Rescaling the binary Hamming distance threshold if the neuron is activated by the total number of 1s in the neuron address. See (Kanerva, 1988; 1993) for details beyond this summary.

Formally, the binary neuron address can be converted into weights $w_i = \pm 1$ with:

$$w_i = \big(2 * [\mathbf{x}_a^\tau]_i\big) - 1, \forall i \in \{1, ..., n\}. \tag{5}$$

Where $\tau$ is used to denote a specific neuron address, the query vector is $\boldsymbol{\xi}$, and the function to determine if the neuron should be activated is $a(\cdot, \cdot)$ that returns a binary action potential. We can write:

$$a(\boldsymbol{\xi}, \mathbf{x}_a^\tau) = I\bigg[\bigg(\sum_{i=1}^n w_i \xi_i\bigg) > d\bigg] = I\bigg[\bigg(\sum_{i=1}^n \big((2 * [\mathbf{x}_a^\tau]_i) - 1\big)\xi_i\bigg) > d\bigg],$$

where there is the action potential threshold $d$ for the neuron to fire and $I[\cdot]$ is the indicator function. The interval of values for address decoding is the number of 1s in the address, $[-\sum_i^n [\mathbf{x}_a^\tau]_i, \sum_i^n [\mathbf{x}_a^\tau]_i]$. By adjusting our $d$ value to account for the possible number of matches, we can implement the Hamming distance threshold for each neuron.

---

[10]App. C.1 discusses of how well inhibitory interneurons can be approximated by Top-K.

The potential for changing the magnitude of neuron weights was considered but a change of their sign was banned due to Dale's Principle. It was acknowledged in (Kanerva, 1988) that changing the magnitudes of the neuron weights results in a weighted sum where the matching of some bits in the addresses matters more than others for calculating if the neuron is activated. However, the crucial feature is the sign of the weights, determining if there is a match or mismatch. This sign change is what was banned by Dale's Principle, making the neuron addresses fixed both to their sign and to their specific value by the use of binary vectors that cannot represent nuances in weight magnitude (Kanerva, 1988; 1993).

However, the original SDM approach still in fact violated Dale's Principle by mapping the binary neuron address to bipolar because of what this means for the pre-synaptic input neurons. Consider the 3rd element of the input vector (either a pattern or a query) being on (a 1 value). The neuron that is active and represents this value will not have a mixture of both excitatory and inhibitory efferent connections with the SDM neuron addresses. In the cerebellar mapping, mossy fibers represent the pattern/query inputs and release glutamate. This means they are always excitatory to granule cells and the 0s in a neuron address should correspond to having no weight (no dendritic connection) rather than a negative weight (inhibitory connection).

An additional issue with the original SDM formulation is that the neuron addresses are dense vectors; this is not the case for the granule cells that they are mapped onto in the cerebellum where each has only $\sim 4$ dendrites (Litwin-Kumar et al., 2017) (see (Jaeckel, 1989a) for a solution that makes the weights of SDM sparse).

Taken together, these issues can be resolved by having the binary neuron addresses be sparse and staying binary rather than becoming bipolar. We can then allow our neuron addresses to use positive real values, simultaneously allowing for changes in weight strengths and respecting Dale's Principle. In this work, we use positive real values but do allow for our weights to be dense, leaving high degrees of weight sparsity to future exploration (Jaeckel, 1989a;b).

## A.5 Additional Modifications to SDM

The five modifications made to SDM for it to be implemented in a deep learning framework and that differentiate it from a vanilla MLP are explained in full here. These modifications are:

1. Using continuous instead of binary neuron activations
2. $L^2$ normalization of inputs and weights
3. No bias term
4. Only positive weights
5. Backpropagation

**1. Continuous Activations -** SDM originally modelled neurons as having binary action potentials by using a Heaviside step function. However, this is non-differentiable and we want to use backpropagation for training our model. This can be resolved by viewing a neuron's action potentials over a time interval, referred to in the neuroscience literature as a rate code (Dayan & Abbott, 2001). We believe this is compatible with SDM, whereby neurons with addresses closer to an incoming pattern or query will receive stronger stimulation and fire more action potentials by having more dendritic connections stimulated by excitatory neurotransmitters.

We can represent neuronal activation as an expected firing rate with a real positive number $a \in [0, t]$ where $t$ is some maximum firing rate. This may look like a rectified tanh function where, due to refractory periods, there are diminishing returns to more stimulation (Glorot et al., 2011). However, because our neurons are already constrained by $L^2$ normalization to be in the cosine similarity bounds of $[-1, 1]$ we simply use a ReLU activation function.

Implementing weighted read and write operations in our original SDM Eqs. 1 and 2, we would replace our binarizing function $b(\cdot)$ with a weight coefficient proportional to the distance between the input and neuron addresses. We show in App. A.7 that this modification has minimal impact on how SDM weights different patterns. This means it should have minimal effect on memory capacity and is still approximately exponential, maintaining its relationship to Transformer Attention (Bricken & Pehlevan, 2021).

**2.** $L^2$ **Normalization -** SDM requires a valid distance metric to compute if neurons and patterns/queries are sufficiently close to each other to produce an activation. We make SDM continuous so that it is differentiable by following (Bricken & Pehlevan, 2021) and replacing its original Hamming distance with cosine similarity.

In SDM's mapping to the cerebellum (outlined in App. A.2), mossy fibers represent input patterns/queries and granule cell dendrites represent neuron addresses. It remains to be experimentally established if and how these specific cell types enforce $L^2$ normalization. However, contrast encoding and heterosynapticity are both ubiquitous mechanisms that can approximate $L^2$ normalization for the mossy fiber activations and granule dendrites, respectively (Sterling & Laughlin, 2015; Rumelhart & Zipser, 1985; Tononi & Cirelli, 2014). From a deep learning perspective, LayerNorm and BatchNorm are both used ubiquitously and can also be viewed as approximations to $L^2$ normalization (Ioffe & Szegedy, 2015; Ba et al., 2016).

**3. No Bias Term -** In order to have our neuron activations represent cosine similarities, we must remove the bias term. We also remove the bias term from the output layer so that outputs represent only a summation of the neuron storage vectors that are activated. The absence of both of these bias terms is clear in the SDM Eq. 2 compared to the MLP Eq. 3.

Bias terms can be viewed biologically as representing a neuron activation threshold (if negative) or a baseline tonic firing rate (if positive). In SDM's cerebellar mapping, granule cells that represent the neurons do not have a tonic baseline firing rate meaning that positive bias terms should not be allowed (Powell et al., 2015; Giovannucci et al., 2017). Purkinje cell firing that represents the output layer is much more complex such that keeping or removing the bias term is hard to justify biologically (Zeeuw, 2020) but we follow the SDM equation in also removing it.

While removing positive bias terms from the granule cells fits neurobiology, removing negative bias terms corresponding to an activation threshold is less justified. However, as it relates to learning dynamics and the ability for Top-K to still be approximated, the ordinary differential equations of (Gozel & Gerstner, 2021) (summarized in C.1), still maintain approximately $k$ neurons firing while using activation thresholds. This is because the fewer neurons that are firing, the less the inhibitory interneuron is activated, keeping more neurons active. As a result, and in line with SDM Eq. 2, we make the simplifying assumption of not allowing for negative bias terms either, thus removing bias terms entirely.

An additional benefit of SDM having no bias term, in conjunction with positive weights, is that all neurons will have positive activations by default. This allows for $k$ annealing instead of the GABA switch that is otherwise needed to inject positive current into each neuron when learning the data manifold (also discussed in App. B.3). Having all neurons active also guarantees that there will always be at least $k$ active neurons to use in the Top-K.

It is noteworthy that the removal of the bias terms is seeing a resurgence in state of the art models such as the 540B parameter PaLM Transformer language model (Chowdhery et al., 2022), which noted that removal of bias terms resulted in more stable training.

**4. Positive Weights -** The connection between mossy fibers and granule cells is excitatory so the weights should be positive. Allowing for both positive and negative weights would violate Dale's Principle (Dale, 1935). As discussed in the last section, the combination of only positive weights and no bias term gives the added benefit of ensuring that all neuronal activations are positive, allowing for the use of the simpler $k$ annealing algorithm.

Coincidentally, having only positive weights also gives better continual learning performance as shown in the ablations of Table 2. A final benefit is the creation of sparse weights but this advantage has its limitations. In models trained on our ConvMixer embeddings, we found only $\sim$20% weight sparsity. Meanwhile, when trying to jointly train a ConvMixer with SDM on top, too many weights were set to 0, resulting in failed training runs.

**5. Backpropagation -** Gradient descent via backpropagation used to train MLPs is likely to be biologically implausible (Lillicrap et al., 2020; Goodfellow et al., 2015). However, there exist a number of local Hebbian learning rules associated with inhibitory interneurons and manifold tiling that we see as viable alternatives (Sengupta et al., 2018; Krotov & Hopfield, 2019; Gozel & Gerstner,

2021). In addition, the cerebellum receives supervised learning signals, via climbing fibers[11], that are known to be capable of encoding heteroassociative relationships such as an eyeblink response to a tone. This circuitry is considered in (Sezener et al., 2021) and presents yet another alternative to backpropagation for training the cerebellum.

## A.6 SDM WRITE OPERATION RELATION TO MLP BACKPROPAGATION

The original SDM write operation directly updates the neuron value vector with the pattern value $\mathbf{x}_v = \mathbf{x}_v + \alpha \mathbf{p}_v$ where $\alpha$ weights the pattern by the amount each neuron was activated. Meanwhile, the backpropgation write operation updates $\mathbf{x}_v$ with the error between the model output and the true class one-hot (using cross entropy loss). During model training all inputs are considered write operations. After training, during inference, all inputs are considered queries that perform the SDM read operation. $\mathbf{p}_a$ corresponds to a CIFAR image and $\mathbf{p}_v$ is a one hot label of its encoding.

There are a few reasons why this difference is compatible with SDM:

First, an optimal solution for $\mathbf{x}_v$ that will result in zero error is the one hot encoding used by the original SDM write operation.

Second, the original SDM write is only appropriate when the neuron addresses are fixed and $\mathbf{x}_v$ is the only thing learnt. Otherwise, as neurons update their address $\mathbf{x}_a$, this changes the patterns they are activated by and what $\mathbf{p}_v$ they should store in $\mathbf{x}_v$. Using the backpropagation approach to continuously update $\mathbf{x}_v$ as a function of the patterns it is currently activated by is a viable solution.

Finally, from a biological perspective, the error signal used by backpropagation is a closer approximation to how the cerebellar circuit that SDM maps to updates $\mathbf{x}_v$ (Bidirectional learning in upbound and downbound microzones of the cerebellum, De Zeeuw, 2021). While backpropagation through multiple layers of a deep network has been argued to be biologically implausible, this update to the output layer is directly connected to the error computation making it possible.

In summary, SDM will explicitly write in $\mathbf{p}_v$ while the MLP with backprop will compute a delta between $\mathbf{p}_v$ and the network output but this approach is compatible with the same solution, works better when also learning neuron addresses, and is likely to be more biologically plausible.

## A.7 RATE CODE ACTIVATIONS MAINTAIN SDM'S APPROXIMATION TO TRANSFORMER ATTENTION

It was shown in (Bricken & Pehlevan, 2021) that the Attention operation of Transformers closely approximates the SDM read operation (Vaswani et al., 2017). This is because the weighting assigned to each pattern in SDM is approximately exponential, resulting in the softmax rule.

However, this result is derived where SDM has binary activations of neurons when reading and writing, not weighting each pattern by its distance. Here, we show that with linear or exponential pattern weightings, SDM remains a close approximation to Attention. This makes the results of previous work continue to hold in our case where SDM is written as an MLP, giving an interesting relation between MLPs and Attention (see Section 6).

The algorithm derived in (Bricken & Pehlevan, 2021) for the size of each circle intersection is summarized before showing how it can implement the weighting coefficient and the effects of this weighting.

As in the original SDM formulation, we are using $n$ dimensional binary vectors with a Hamming activation threshold of $d$ and the distance between the addresses of a query $\boldsymbol{\xi}$ and pattern $\mathbf{p}_a$ is $d_v := d(\boldsymbol{\xi}, \mathbf{p}_a)$. We can group the elements of $\boldsymbol{\xi}$ and $\mathbf{p}_a$ into two disjoint sets: the $n - d_v$ elements

---

[11]It is an open question if the climbing fiber signals encode errors or the target to be learnt but we use "supervised" here as a superset of the two and distinct from unsupervised learning without any "teacher" signal.

where they agree and the $d_v$ elements where they disagree[12]:

$$\boldsymbol{\xi} = [\overbrace{1,\ldots,1\mid 0,\ldots,0}^{n-d_v}\mid\overbrace{1,\ldots,1\mid 0,\ldots,0}^{d_v}] \tag{6}$$
$$\mathbf{p}_a = [\underbrace{1,\ldots,1\mid 0,\ldots,0}_{a+z}\mid\underbrace{0,\ldots,0\mid 1,\ldots,1}_{b+c}]$$

Now imagine a third vector, representing a potential neuron address. This neuron has four possible groups that the elements of its vector can fall into:

- $a$ - agree with *both* $\boldsymbol{\xi}$ and $\mathbf{p}_a$
- $z$ - disagree with *both* $\boldsymbol{\xi}$ and $\mathbf{p}_a$
- $b$ - agree with $\boldsymbol{\xi}$ and disagree with $\mathbf{p}_a$
- $c$ - agree with $\mathbf{p}_a$ and disagree with $\boldsymbol{\xi}$

We want to constraint the values of $a$, $z$, $b$ and $c$ such that the neuron address exists inside the circle intersection between $\boldsymbol{\xi}$ and $\mathbf{p}_a$. This produces the following constraints:

$$a + b + c + z = n$$
$$a + b \geq n - d$$
$$a + c \geq n - d$$
$$a + z = n - d_v$$
$$b + c = d_v$$

Using the notation of (Vaswani et al., 2017), we can write the total number of neurons that exist in the intersection of the read and write circles as:

$$\sum_{a=n-d-\lfloor\frac{d_v}{2}\rfloor}^{n-d_v}\sum_{c=\max(0,n-d-a)}^{d_v-(n-d-a)} w_{\text{Type}}(a,c,d_v,n)\left(\binom{n-d_v}{a}\cdot\binom{d_v}{c}\right),$$

where introduce the weight coefficient $w_{\text{Type}}()$ that can be:

$$w_{\text{Binary}}(a,c,d_v,n) = 1 \tag{7}$$

$$w_{\text{Linear}}(a,c,d_v,n) = \frac{a+c}{n}\cdot\frac{a+(d_v-c)}{n} \tag{8}$$

$$w_{\text{Exp}}(a,c,d_v,n) = \exp\left(-\beta\big(n-(a+c)\big)\right)\cdot\exp\left(-\beta\big(n-(a+(d_v-c))\big)\right), \tag{9}$$

where $w_{\text{Binary}}$ is the original weighting of 1 for everything; $w_{\text{Linear}}$ applies a linear decay from 1 for a perfect vector match to 0 for the maximum distance allowed between vectors; and $w_{\text{Exp}}$ is an exponential decay weighting that uses a $\beta$ coefficient for its decay rate.

As we will now show, the linear weighting applies the most weight to patterns right in the middle of the pattern and query with a gradual, symmetric decline around this point. Meanwhile, the exponential weighting cancels out to apply a constant weight re-scaling to everything. Letting $x$ represent the distance of a neuron to the read and write vectors it is weighted by, $x := (n-z)-(a+c) = b-z$. Because we want to know the weighting coefficient that applies to neurons at all possible distances from the pattern and query, without loss of generality we can set $z = 0$ for the analysis that follows:

$$\begin{aligned}w_{\text{Linear}}(a,c,d_v,n) &= \frac{n-x}{n}\cdot\frac{n-((d_v+z)-x)}{n}\\ &= \frac{n-x}{n}\cdot\frac{n-(\mathcal{C}-x)}{n}\\ &= \frac{n^2-x^2-n\mathcal{C}+x\mathcal{C}}{n^2}\\ &= Z + \frac{-x(x-\mathcal{C})}{n^2}\end{aligned} \tag{10}$$

---

[12]This formulation was first inspired by (Jaeckel, 1989b;a).

where $\mathcal{C} = (d_v + z)$ is a constant as is $Z = \frac{n^2 - n\mathcal{C}}{n^2}$. We can take the first and second derivatives of Eq. 10 to know that the most weight is applied at the distance right between the read and write vectors, $\mathcal{C}/2$. This linear weighting applies different weights to neurons at different distances. As a result, the patterns stored in superposition will also have different weightings. However, empirically, this has a negligible effect on the SDM exponential approximation, as shown in Fig. 7. We hypothesize that this is due to two factors: (i) the difference in weight values is not particularly large; (ii) neurons receiving the largest weights are the most numerous. Therefore, the weighting is correlated with the approximately exponential decay in the number of neurons that exist in the circle intersection as vector distance changes.

As for the exponential, it cancels to a constant term that depends upon the choice of $\beta$:

$$w_{\text{Exp}}(a, c, d_v, n) = \exp\left(-\beta x\right) \cdot \exp\left(-\beta(\mathcal{C} - x)\right)$$
$$= \exp\left(-\beta\mathcal{C}\right). \tag{11}$$

This constant term modification to the weighting of all patterns is then removed by the normalization term in the softmax operation, resulting in no effect on the output.

We confirm our results empirically for the three optimal SDM hamming distances $d^*$ and $n = 64$ dimensional canonical Transformer Attention setting used throughout (Bricken & Pehlevan, 2021) in Fig. 7.

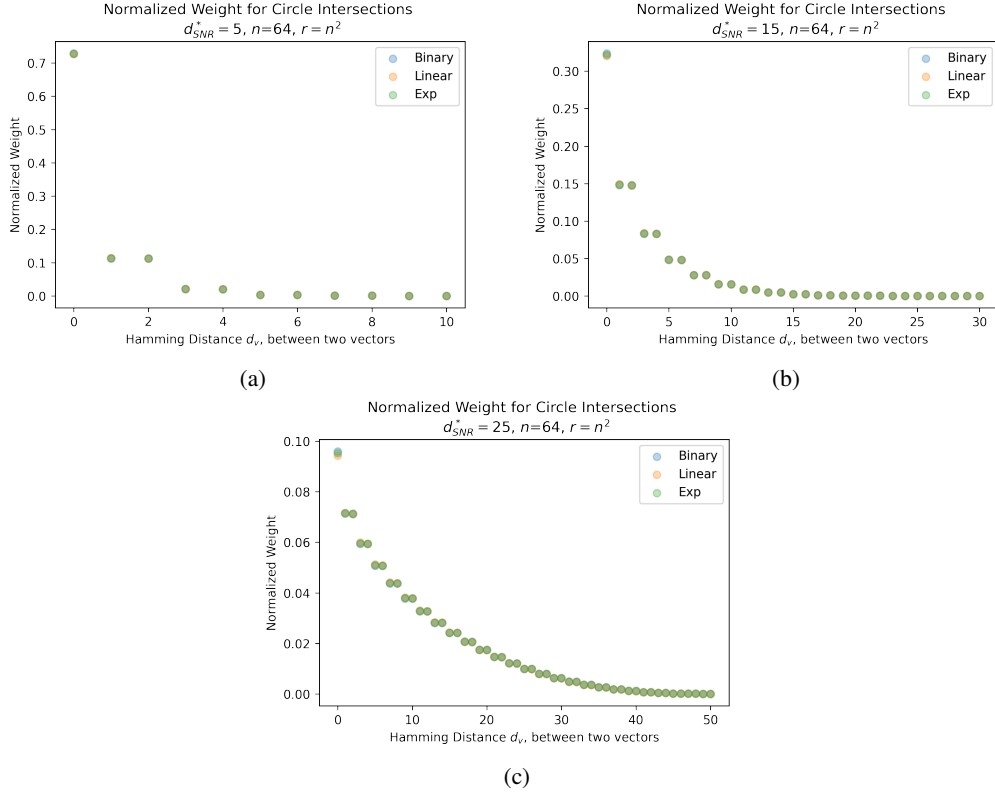

Figure 7: **Weighting neuron read and write operations has a negligible effect on SDM.** The Exponential weighting (green) as expected gives the same result as if there was a Binary weighting (blue) making it overlap perfectly. The Linear weighting (orange) results in a negligible difference that is barely visible due to high overlap.

## B   GABA Switch

### B.1   Biologically Implausible Solutions to the Dead Neuron Problem

The dead neuron problem is where neurons are never activated by any input data and are a waste of resources. This is particularly a problem with the Top-K activation function, especially at initialization where it is possible for a small subset of neurons that happen to already exist closest to the data manifold to always be activated and in the Top-K. This means that none of the other neurons are ever activated, and thus never receive weight updates, preventing them from learning the data manifold and becoming useful.

The first set of strategies to solve the dead neuron problem initializes neurons immediately on or near the data manifold, for example, using PCA or SVD (Rumelhart & Zipser, 1985; McInnes & Healy, 2018). Biologically, this would assume that priors on the location of the data manifold are encoded in our genome and pre-determine the structure of neuronal dendrites. However, dendrites often form in a highly dynamic and stochastic way (Dhar et al., 2018; Purves et al., 2001).

In addition, this solution requires the genome to store a large amount of information and would limit an organism's adaptability to learn new data manifolds during its development, for example by moving to a new environmental niche. It is popular in deep learning to initialize weights not using the data directly but with optimized strategies such as Xavier (Glorot & Bengio, 2010) that empirically work well, even for the sparse ReLU activation function. However, as evidenced by the continued development of new initialization strategies, this approach remains heuristic and does not generalize across datasets, networks, and training techniques (Aguirre, 2019; Chang et al., 2020b; Goodfellow et al., 2015).

The second set of approaches to solve the dead neuron problem ensures that all neurons, independent of their initialization, are active and update to learn the data manifold. This is a major cited reason why ReLU is often avoided (Goodfellow et al., 2015; Hendrycks & Gimpel, 2016), even though in practice it often does not seem to significantly harm performance (Pedamonti, 2018; Trockman & Kolter, 2022). The Top-K model of (Ahmad & Scheinkman, 2019), VQ-VAE (van den Oord et al., 2017), and Switch Transformer (Fedus et al., 2021) all address their dead neuron problems in biologically implausible ways. When using the Top-K activation function, the work of (Ahmad & Scheinkman, 2019) artificially "boosted" the ranking for each neuron to be in the Top-K, with more inactive neurons getting higher rankings. This solution makes the biologically implausible assumption that neurons fire as a function of how inactive they are and undergo ubiquitous anti-Hebbian plasticity. Meanwhile, the VQ-VAE and Switch Transformer both used terms in their loss function to increase the utilization of "dead" code vectors and expert modules, respectively.

### B.2   GABA Switch Biology

We summarize how the GABA switch works biologically in Fig. 8 (Gozel & Gerstner, 2021). Neurons are excited by GABA early in development before being inhibited by it due to changes in intracellular $CL^-$ concentration. After neurogenesis, neurons express NKCC1 which imports $CL^-$ *into* the cell causing the $CL^-$ reversal potential to be more positive than resting potential (Gozel & Gerstner, 2021; Heigele et al., 2016). When GABA is present, $CL^-$ flows towards its reversal potential, resulting in depolarization. Over time, this GABA activation indirectly results in increased KCC2, which instead pumps $CL^-$ *out* of the cell (Ganguly et al., 2001; Connor et al., 1987). This makes the $CL^-$ reversal potential more negative than the resting potential and hyper-polarization during GABA activation.

This biological mechanism suggests that our GABA switch implementation in Eq. 12 could be made more sophisticated. Rather than counting up the number of times the neuron is simply active (a binary outcome), it could switch as a function of the actual activation amount that the neuron experiences or size of gradient updates.

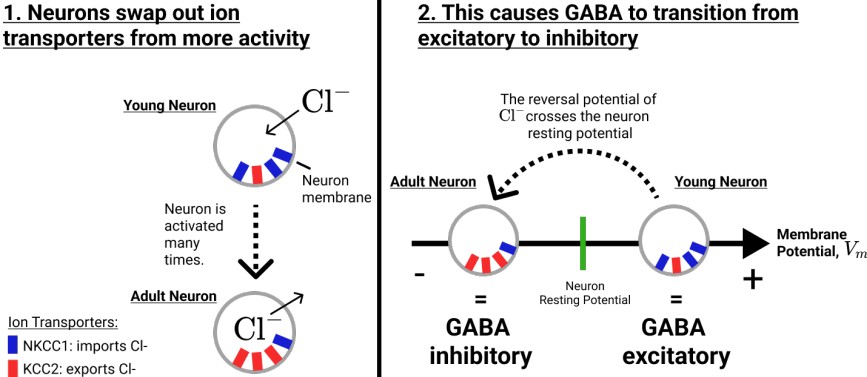

Figure 8: **Summary of GABA Switch Biology.** Here we visualize the ion transporters inside young versus adult neurons and how they change over time. The left side of the figure shows how as a neuron gets activated more times it swaps out the concentration of its ion transporters from NKCC1 (blue) to KCC2 (red). This causes the neuron to go from importing chloride ions (Cl-) to exporting them. The right side of the figure shows the consequence of this transition where the reversal potential of chloride goes from being more positive than the neuron's resting potential – resulting in GABA being excitatory – to more negative – resulting in GABA being inhibitory.

### B.3 GABA Switch Practical Considerations

There are three conclusions from this section on when and how to use the GABA switch to avoid dead neurons, improving continual learning:

- If the positive weight constraint is being used then annealing $k$ gives the same performance as the full GABA switch. If you allow for negative weights then the GABA switch should be used.
- The subtraction operation works much better for continual learning than masking. However, the inverse is true in a non continual learning setting. We provide some initial analysis of why this is the case.
- The value of $s$ for the GABA switch should consider the complexity of the data manifold, optimizer, and learning rate. When in doubt there is no harm in setting $s$ larger aside from requiring more training epochs.

We first give the full GABA switch algorithm that was developed and is used for the SDMLP throughout the paper. We then outline why the $k$ annealing approximation works just as well when we constrain our weights and inputs to be only positive. Next, we discuss why subtracting to enforce Top-K instead of masking leads to better continual learning performance. We conclude with other considerations on how to avoid dead neurons with interactions between the data manifold, learning rate, and hyperparameter $s$.

**The GABA Switch Algorithm**

Formally, the GABA switch algorithm is implemented as:

$$
\begin{aligned}
a_i^* &= [a_i - \lambda_i I]_+ \\
I &= \text{descending-sort}([\mathbf{a}]_+)_{(k_{\text{target}}+1)} \\
\lambda_i &= \min\left(1, \max\left(-1, -1 + 2C_i/s\right)\right)
\end{aligned}
\tag{12}
$$

This algorithm is slightly more complex than the K annealing one presented in Eq. 4 of the main text. $C_i$ is a counter for each neuron, recording the number of times that it has been activated since the start of training. $\lambda_i$ linearly increases from -1 to 1 as a function of $C_i$, and $[\cdot]_+$ is the ReLU operation. $s$ is a hyperparameter that determines the number of activations required for this neuron to *switch* from being excited by GABA to being inhibited. When training on top of the ConvMixer embeddings, we found that $s = 250,000$ ensured all neurons could move onto the data manifold, resulting in none being dead. As an example, if we assume all neurons are activated before the GABA switch, with this $s$ value the switch occurs after just 2.5 epochs of training on CIFAR10 (there are 50,000 training examples). However, this value depends significantly upon the learning rate and complexity of the manifold where in the case of training on raw CIFAR10 images, to ensure there are no dead neurons we set $s = 5,000,000$ requiring 100 epochs for the GABA switch to occur. However, in this case we did not try to find the lower bound on $s$.

**Why $k$ annealing works just as well as the GABA switch when weights are positive**

Because of Dale's Principle, we implement SDM with positive weights and inputs. Empirically, this also boosts continual learning performance and introduces weight sparsity[13] that would increase computational efficiency if unstructured sparsity can be taken advantage of (Wang, 2020).

When positive weights are used, we found no change in the number of dead neurons or continual learning performance using Eq. 4 linear annealing. This was true for both the ImageNet32 ConvMixer embeddings and raw pixel datasets. However, when negative weights are allowed this was no longer the case and only the full GABA switch algorithm avoided dead neurons. We speculate that this is because when GABA is excitatory, the full algorithm will inject positive activation into neurons that may otherwise have negative activity and not fire thus failing to get a gradient update. Meanwhile, with positive weights and inputs the activation for every neuron is always positive resulting in every neuron being active unless Top-K is enforced. Thus the positive weights and inputs

---

[13]We found that $\sim 20\%$ of weights in two models checked (trained on the ImageNet32 ConvMixer embedding and directly on pixels) had values less than 0.01 meaning they could likely be pruned, however, we leave further investigations of sparsity including the introduction of $L^1$ losses, etc to future work.

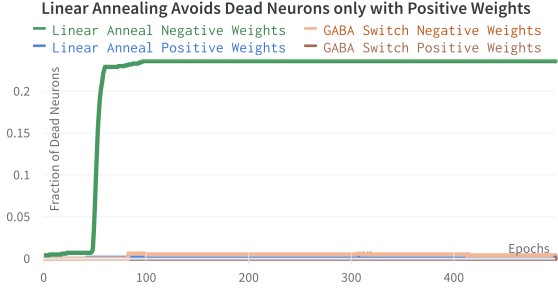

Figure 9: **Linear $k$ Annealing needs Positive Weights**. All lines overlap at the bottom of the plot aside from Linear Annealing when negative weights are allowed (green). The GABA Switch excites neurons that can otherwise be inactivated due to their negative weights. This ensures all neurons update onto the data manifold and are not dead. However, linear $k$ annealing presented in Eq. 4 does not excite leaving these neurons dead. Training is directly on the CIFAR10 manifold which is more difficult to learn than the ConvMixer embeddings.

ensure that every neuron receives gradient updates and moves onto the data manifold. In Fig. 9, we present this result training directly on CIFAR10 pixels, a more difficult manifold to learn than the ConvMixer embeddings.

Note that while positive weights allow for linear $k$ annealing, it can also result in all weights be set to 0, leading to dead neurons and model training to fail.

**Why *subtracting* works better than *masking* for Continual Learning**

Empirically, masking results in much higher validation accuracies during pretraining than subtraction (Fig. 10b). However, during continual learning, subtraction works better as shown in the ablations of Table 2 and Fig. 10a. To work out why subtracting is better for continual learning we analyzed the learning dynamics of both in App. H and discuss them here. We leave an analysis of why masking works better during pretraining to future work as we do not care about maximizing validation accuracy for our pretraining task here.

Top-K with subtraction utilizes more neurons during continual learning as shown in Fig. 10c. This result is supported by App. H Fig. 23 where there are fewer neurons that are consistently activated and they are more polysemantic, being used for multiple tasks.

Motivated by the stale momentum results of App. D, we wondered if the larger activation values preserved by masking may lead to more dead neurons when using SGDM as our momentum based optimizer. However, using SGD did not lead to any change in results and in hindsight our gradients being less than 1 means that the results in App. D hold independent of the gradient values.

This means that the gradients must actually be smaller for subtraction and lead to slower updating. At first glance, the subtraction operation just reduces the activity of the $k$ neurons that remain active by a constant term. This will scale the size of each gradient, equivalent to modifying the learning rate.

However, the situation is more complex. The activity of the $k$ neurons and their learning rate is conditioned upon the activity of the $k + 1$th neuron. If this neuron is almost as active as those in the $k$ subset, all of their activities will be very close to zero. Meanwhile, if the $k + 1$th neuron is much less active than the $k$ subset, their activities will remain large and they will receive a larger gradient update. These two situations correspond to the input being in densely and sparsely tiled regions of the data manifold, respectively.

We can think of this situation as approximating a Bayesian one where we have a mixture of Gaussians defining a posterior distribution over the data manifold. The number of neurons in a region is inversely proportional to the variance in the distribution at that location and thus changes the amount the likelihood of our current data point updates the distribution. In other words, the subtraction operation introduces a dynamic learning rate.

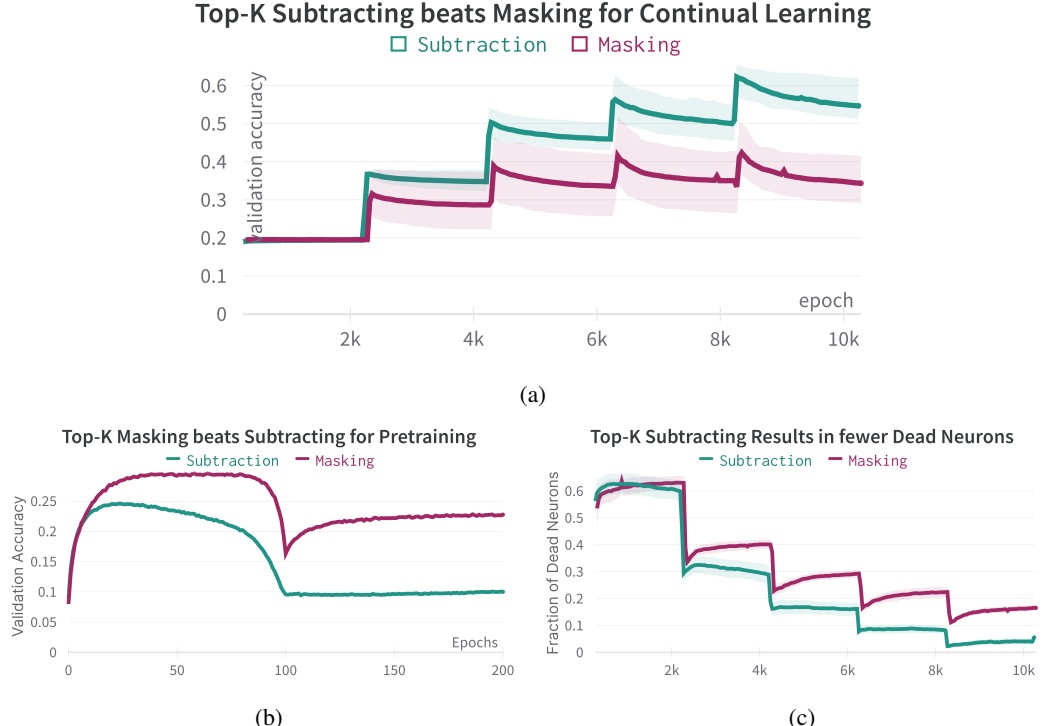

Figure 10: **(a)** Subtraction beats Masking for Continual Learning. However, as shown in **(b)** it is worse for pretraining. During pretraining, neither method creates any dead neurons. **(c)** During continual learning, across each task, there are fewer dead neurons when subtracting that may explain the better memory retention of previous tasks. Because these plots are not crowded, error bars show the min and max range across the five random continual learning data splits instead of the standard error of the mean.

During continual learning when the task changes and a new subregion of the data manifold is presented, we want those neurons closest to the subregion and only those neurons to update and move onto it. In our stochastic optimization setting, it is possible that neurons not necessarily closest to new subregion are updated towards it instead, leading to an over-allocation of neurons to the new task. We hypothesize that the subtraction operation helps avoid this problem by making the learning rate both lower and dynamic. However, we believe the dynamic learning effect is more influential than stochasticity because even when increasing the batch size from 128 to 10,000, we see that subtraction is still more robust to catastrophic forgetting.

It is likely that the reduced and dynamic learning rate is a doubled edged sword where while it is better at remembering, it is also worse at learning new information. Both in pretraining and within each data split, the subtraction achieves worse training and validation accuracies. Why the subtraction operation does worse during the pretraining on a single task we leave to future work.

As an interesting aside, the Top-K subtraction operation implemented by an inhibitory interneuron when $k = 1$ is the same as a Vickrey second price auction used in (Chang et al., 2020a) to model neurons as individual agents with their own utility functions. The winning neuron places a "bid" represented by its activity amount. It has its activity subtracted by that of the second most active neuron (thus paying the second highest price) and its remaining activity determines its exposure to the next gradient update.

**Other GABA Switch Considerations - Data Manifolds, Optimizers, and Learning Rates**

Note that for the ConvMixer embedding, dead neurons are not as much of a problem and either the GABA switch or $k$ annealing can avoid any dead neurons within just a few epochs and with or without positive weights. This is because the manifold is much easier to learn as it is lower

dimensional both in the size of vectors (256 versus 3072) and the data manifold being far more meaningfully correlated.

Fig. 11 shows how a learning rate that is too low will cause SGD to produce dead neurons. The number of neurons dying being inversely proportional to the learning rate and the sudden jump in dead neurons before it plateaus just after the GABA switch both support the theory that the learning rate is too low. This means that neurons do not update quickly enough onto the data manifold before the GABA switch occurs. This sudden jump and plateau in dead neurons is in contrast to Adam and RMSProp optimizers shown in Fig. 15 where neurons will continue dying over time due to the stale momentum problems (Kingma & Ba, 2015; Geoffrey Hinton, 2012). Meanwhile, SGDM does not suffer from Stale Momentum problems to the same extent and is much more robust to choice of learning rate making it a good choice for pretraining. However, it is still suffers somewhat from the stale momentum problem in the continual learning setting as shown in the ablations of Table 2.

To avoid dead neurons when using SGD either the learning rate can be tuned, the GABA switch value $s$ can be increased, or a sparse optimizer that avoids the stale momentum problem can be used (however, the success of these sparse optimizers remains to be validated in future work). Also keep in mind that the learning rate chosen will be affected by the $L^2$ normalization.

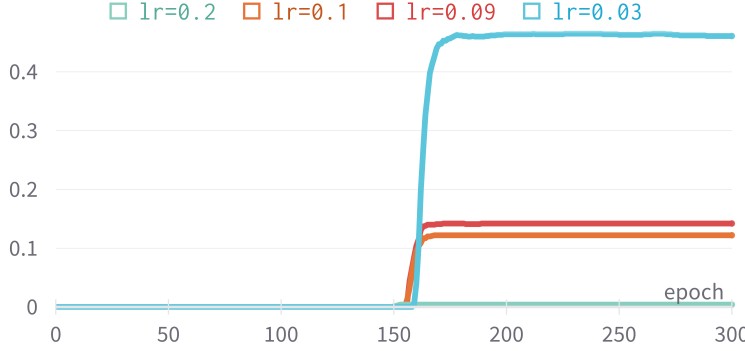

Figure 11: **The GABA switch needs to account for the learning rate**. We train SDM directly on CIFAR10 pixels to test its ability to learn the manifold and avoid dead neurons. The GABA switch occurs at epoch ∼150 and each line denotes a different learning rate "lr". For reasons outlined in App. D, SGD should result in the fewest dead neurons. However, the learning rate needs to be carefully set in relation to the GABA switch threshold, $s$. If the learning rate is too small, neurons won't update onto the data manifold quickly enough and will die instead.

## C   TOP-K

### C.1   INHIBITORY INTERNEURONS APPROXIMATE TOP-K

At a high level, inhibitory interneurons exist to regulate the firing of excitatory neurons, introducing sparse firing and keeping only the most active on.[14] This makes the brain likely to be highly sparse in the number of neurons firing at any given time. Estimates suggest that "an average neuron in the human brain transmits a spike about 0.1-2 times per second." (Impacts, 2022), while action potentials and refractory periods happen on a time interval of roughly 10ms (Sterling & Laughlin, 2015). Assuming neurons fire randomly within this 10ms time interval, this gives a back of the envelope calculation that a maximum of 10ms/100ms = 10% neurons will fire within the time interval. This aligns well with the prediction of 15% by (Attwell & Laughlin, 2001). Sparse firing also makes sense from the perspective of metabolic costs where action potentials are expensive, consuming

---

[14]Inhibitory interneurons can also inhibit each other resulting in disinhibition but this is still used to perform more sophisticated forms of excitatory neuron regulation.

~20% of the brain's energy (Sterling & Laughlin, 2015; Attwell & Laughlin, 2001; Sengupta et al., 2010).

However, in practice, it is unrealistic to assume that an inhibitory neuron can keep exactly $k$ neurons active for any given input because its afferent and efferent synaptic connections are heterogeneous. The inhibitory interneuron sums together the activations from all neurons, weighted by their presynaptic strengths and then outputs an inhibitory value that is scaled by the strengths of post synaptic connections. The weighted summation of inputs to the interneuron removes information on how many neurons are firing and at what rate. The heterogenous post synaptic connections will weight the effects of inhibitory on each neuron differently.

Keeping these concerns in mind, there are a few reasons that we draw from for optimism that Top-K can be approximated. First, inhibitory horizontal cells in the retina compute the mean activation value of thousands of nearby cone photoreceptors and inhbits them so that they encode contrast (Chapter 13 of (Sterling & Laughlin, 2015)). Not only do horizontal cells have carefully tuned weights for their gap junction connections to compute this mean, but also will dynamically change its pre and post synaptic connectivity strengths to accurately compute mean activity under different lighting conditions. In a low light setting, the horizontal cell will average over a larger number of neurons to reduce the variance in its estimate of the mean (page 252 of (Sterling & Laughlin, 2015)). If inhibitory interneurons in general come close to the level of sophistication shown by horizontal cells in calibrating their synaptic connectivity, then it is possible the interneuron can compute how much it should inhibit to implement Top-K reliably.

Second, looking at biological evidence of Top-K from cerebellum-like structures, there is strong evidence of Top-K being approximated in the Drosophila Mushroom Body by its Golgi interneuron analog (Lin et al., 2014). In mammals, the evidence is more complicated with some papers finding dense granule cell activations (Giovannucci et al., 2017). However, other experimental evidence and theory suggests that granule cell sparsity in the mammalian cerebellum may be a function of task complexity (Lanore et al., 2021; Xie et al., 2022). The lower complexity the task, the more dense the representations can afford to be as there are fewer stimuli that must be unique encoded. Meanwhile, high complexity tasks require a larger number of orthogonal codes resulting in the need for sparser representations that lose fidelity as a result (Xie et al., 2022). Ultimately, the density of granule cell firing remains an open question in need of lower latency voltage indicators and the ability to record from more neurons simultaneously across a more diverse range of tasks (Lanore et al., 2021).

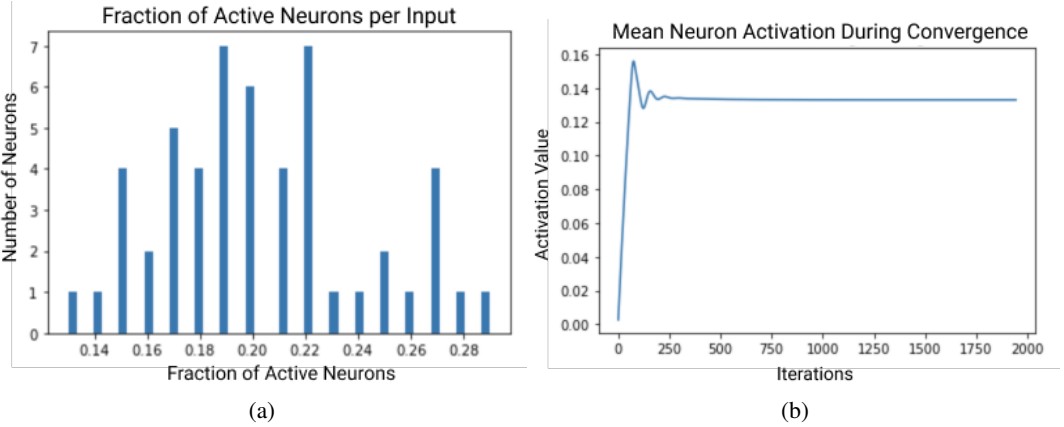

(a)  (b)

Figure 12: **Inhibitory Interneuron Approximates Top-K**. We reproduce the Hebbian learning rule of (Gozel & Gerstner, 2021). **(a)** We show that it approximates having $k \approx 200$ of the 1000 neurons active for 50 different MNIST inputs. **(b)** We show an example of the average neuron activity running the ODEs given in Eq. 13 until convergence.

Finally, we tried to model the ability for an inhibitory interneuron to approximate Top-K under the simplifying assumption that its pre and post synaptic weights are fixed and homogenous. We first analyzed the $k$ neurons still on using the Hebbian learning dynamics of (Gozel & Gerstner, 2021). We then looked at our SDMLP during learning and the relationship between the total sum of excitatory activations and the $k+1$-th inhibitory value used in Eq. 4 to keep only $k$ neurons still on.

The work that introduces the GABA switch (Gozel & Gerstner, 2021) uses a Hebbian learning rule with an inhibitory interneuron. An MNIST digit is presented for the network to learn and it obeys the following ordinary differential equations to determine the activities of the excitatory and inhibitory neurons:

$$\mathbf{e}^0 = W_{\text{inp}}^T \mathbf{x}$$
$$\mathbf{i}^0 = 0$$

$$\mathbf{p}_a^t = \mathbf{e}^0 + W_{IE}\mathbf{i}^{t-1}$$
$$\mathbf{p}_i^t = \mathbf{e}^0 + W_{EI}\mathbf{e}^{t-1}$$

$$\mathbf{g}_a = \frac{1}{\tau_a}\big(-\mathbf{e}^{t-1} + \tanh\left([(\mathbf{p}_a^t - \mathbf{b})/L]_+\right)\big)$$
$$\mathbf{g}_i = \frac{1}{\tau_i}\big(-\mathbf{i}^{t-1} + [(\mathbf{p}_i^t - b_i)]_+\big)$$

$$\mathbf{e}^t = \mathbf{e}^{t-1} + \Delta\mathbf{g}_a$$
$$\mathbf{i}^t = \mathbf{i}^{t-1} + \Delta\mathbf{g}_i, \tag{13}$$

where $\mathbf{e}$ is a vector describing excitatory neuron activities, $\mathbf{i}$ the inhibitory neurons, and $\Delta$ is the learning rate. The excitation and inhibition time constants are $\tau_a$ and $\tau_i$, respectively. $L$ is a smoothing term, $\mathbf{b}$ is a firing threshold learnt for each neuron. $W_{IE}$ and $W_{EI}$ are the weight matrices from excitatory to inhibition neurons and vice versa; they are fixed and have homogenous weights. $W_{\text{inp}}$ are the weights that respond to the input $\mathbf{x}$. $b_i$ is the firing threshold for the inhibitory interneurons that a fixed scalar and can be thought of as resulting in a noisy approximate $k$ value. During training, we use a Hebbian learning rule (not given) to train $W_{\text{inp}}$ and run the ODEs shown to converge to approximately $k$ neurons remaining on.

These dynamics are simplifications because the weights between excitatory and inhbitory neurons are fixed and homogenous. $W_{EI}$ is initialized where every excitatory neuron is connected to each inhibitory with a probability of 90% and strength of 1. $W_{IE}$ is initialized with the same connectivity probability but with a strength of $1/(0.9r_I)$ where $r_I$ is the number of inhibitory neurons. This makes the synaptic weights connecting inhibitory to excitatory neurons sum to 1 in expectation.

Putting aside these simplifying assumptions, when we run the model and analyse the number of neurons that remain active, it is approximately a constant $k$ neurons as shown in Fig. 12a. Given the inhibitory activation threshold $b_i$, approximately 20% of the neurons remain on for each input giving $k \approx 200$. We also show how the average activity of the excitatory neurons evolves over time in Fig. 12b to emphasize that the inhibitory interneuron does not have to make a single guess for how to correctly inhibit all but $k$ neurons. Instead, this is a dynamic process defined by the ODEs of Eq. 13.

We also looked for our SDMLP at the $k + 1$-th highest activity value used by the inhibitory neuron to implement Top-K and how this relates to the total unweighted sum of neuron activations entering the inhibitory interneuron. After the GABA switch when neurons tile the data manifold and form subnetworks, this relationship is largely linear where the $k + 1$-th highest activity value is the total sum of all neuron activations before inhibition, divided by the total number of neurons. This is again a simplification by assuming all input and output weights are homogenous and equal to one. However, again, if the interneuron can modify its pre and post synaptic strengths with the same degree of precision displayed by horizontal cells (Sterling & Laughlin, 2015) then these results support the possibility of the interneuron approximating Top-K, using a simple function to respond to the sum of its inputs.

## C.2 Optimized Top-K

The $k$ value is a parameter which must be chosen in our model and can significantly affect performance. We present empirical results for ablating $k$ on both pretraining and continual learning across a number of different data manifolds.

To summarize our findings:

- A larger $k$ gives better pretraining but worse continual learning and vice versa.

- $k$ depends upon the complexity of the data manifold and number of neurons, $r$.

- SDM will approximately tile whatever data manifold it is given. Joint training of SDM with other model components such as a CNN will improve pretraining. However, learning a non static manifold in this joint training setting is difficult for SDM.

**Optimal $k$ values for Pretraining or Continual Learning?**

Sparsity in the SDM setting presents a fundamental tradeoff between performing well in continual learning (CL) and non continual learning (NCL) settings. This tradeoff exists because sparsity limits the representational capacity of the model and should be expected to reduce NCL accuracy. Meanwhile, the more sparse the model is, the better it is able to form subnetworks that are not overwritten by future CL tasks. Additionally, SDM will tile the data manifold it is given, which depending on the manifold, can be poorly correlated with maximizing classification accuracy.

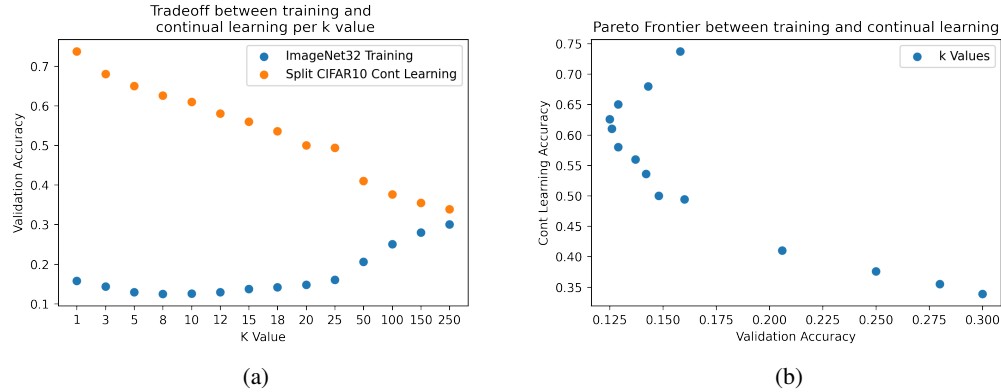

(a)                                               (b)

Figure 13: Two perspectives on how the $k$ value influences the ability to learn the ImageNet32 pretraining task and the Split CIFAR10 continual learning task. **(a)** Validation Accuracies for final NCL ImageNet pretraining (blue) and final CL Split-CIFAR10 (orange) over all tasks as a function of $k$ values. **(b)** The pareto frontier between ImageNet pretraining accuracy (x-axis) and CIFAR10 continual learning (y-axis). Each blue dot is a different k value. This plot makes evident the better pretraining performance for $k < 8$.

We see this tradeoff in Fig. 13 that summarizes ablating $k$ during pretraining on ImageNet32 embeddings from the ConvMixer and testing on Split CIFAR10. As the $k$ value decreases, continual learning accuracy improves while pretraining accuracy declines. However, it is unclear why performance on the original ImageNet32 training is parabolic, slightly improving for $k < 8$. While we are indifferent to the NCL pretraining task here, using it as a way for the neurons to specialize across an arbitrary manifold of real-world images, it is worth acknowledging the performance is worse. That this performance decline is caused by sparsity is even more evident during a single training run as GABA switches and $k$ is slowly annealed. Fig. 14a shows this for smaller $k$ values where performance declines.

To emphasize how the optimal $k$ value changes as a function of the number of neurons and data manifold, we describe a number of additional results:

For the same ConvMixer embeddings of ImageNet32 pretraining and Split CIFAR10 testing, if we use 10K neurons instead of 1K then there is the same linear decline in pretraining performance as

in Fig. 13 but $k = 5$ outperforms all $k \leq 50$, it is also the best for continual learning. Meanwhile, $k = 1$ becomes the worst for pretraining and the second best for continual learning.

With the same setup again but operating directly on pixels without using the ConvMixer embeddings, $k = 10$ is the best for pretraining but $k = 1$ is the best at continual learning as shown in Fig. 18 of App. E.1.

In all of the above cases we are given a static manifold for SDM to learn. This always results in worse pretraining accuracy than an equivalent ReLU MLP, even when using large $k$ values. This is because the ReLU MLP is not constrained to the data manifold, having a bias term and no $L^2$ normalization. This allows it to learn weights and biases that better maximize NCL validation accuracy.

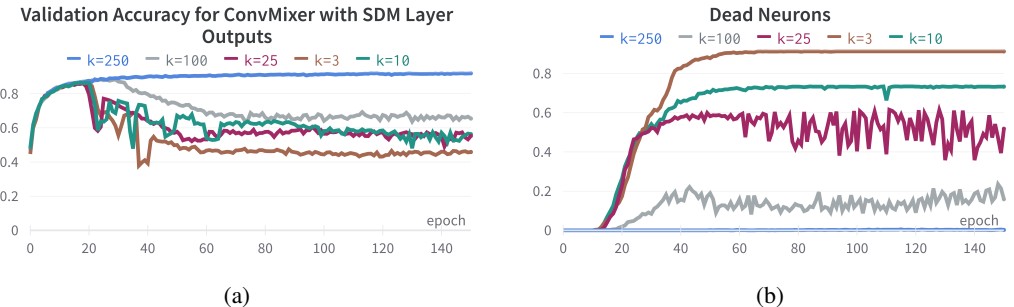

(a)             (b)

Figure 14: **Sufficiently small $k$ values harm network training**. ConvMixers with an SDM layer at the end trained on CIFAR10. All networks perform the same for the first 20 epochs at which point Top-K is approximately fully implemented (the GABA switch occurs around epoch 10). **(a)** Validation accuracy for all models. The smaller the $k$ value, the worse accuracy is. **(b)** The fraction of neurons that are not active for any of the validation inputs (this is an upper bound on the true number of dead neurons that may be inactive for the training data). Training the CNN portion of the network along with the SDM neurons leads to many dead neurons. We think this is because the manifold is still updating. As more neurons die, the representational capacity of the model declines.

If we perform joint training of the ConvMixer and SDM module then the ConvMixer can learn to create a manifold for SDM to tile that does maximize validation accuracy, performing on par with the ReLU MLP. This is what happens in a test where we train on the whole CIFAR10 dataset in the NCL setting, as long as $k \geq 250$, but independent of if there are 1,000 or 10,000 neurons in the SDM layer. This result is shown in Fig. 14a and suggests that the ReLU network only needs at least 250 neurons in its final layer to backpropagate gradients and achieve high prediction accuracy. There is evidence that artificial neural networks are overparameterized at the start of training and functionally sparse, as supported by results from network pruning, most notably the Lottery Ticket Hypothesis (Frankle & Carbin, 2019).

However, this result is again manifold dependent whereby training instead on ImageNet32, even with $k = 2,500$ and 10,000 neurons still harms performance compared to a ReLU network. We believe this is because ImageNet32 has a dramatically more complex data manifold with $\sim 1.2$M images from 1,000 different classes, compared to 50,000 images in 10 classes for CIFAR10. This means that even a $k$ of 2500 is too small and harms the model's representational capacity.

An additional difficulty with joint training is the timing of the GABA switch to avoid dead neurons. This is because the manifold SDM learns is dynamically changing over time and we believe it explains the neuron death that is inversely proportional to $k$ shown in Fig. 14b.

We leave it to future work to further investigate optimal $k$ values and joint training versus using frozen, pretrained models. We are optimistic that the best way to resolve the tradeoff from sparsity limiting representational capacity is to make the network layers wider rather than the number of neurons that can be active, $k$, fewer. This is feasible because sparse activations are computationally cheap – while computing the $k$ most active neurons requires multiplying the input with each neuron (this can be parallelized) and then sorting them, a constant $k << r$ neurons produce outputs and receive gradient updates. Computing which neurons are active would also be cheaper with sparse weights (Jaeckel, 1989a; Ahmad & Scheinkman, 2019).

**SDM Theory to derive optimal $k$ values**

Before concluding this section, we want to flag to readers that SDM, with a few simplifying assumptions, is able to analytically derive optimal $k$ values under a few different conditions of optimality: (i) Optimising the signal to noise ratio of every pattern; (ii) Optimizing for the maximum number of patterns that can be stored, within a certain retrieval probability; (iii) Maximizing the distance a query can be from its target pattern while still converging correctly. These derivations and further discussion be found in (Kanerva, 1993; Bricken & Pehlevan, 2021).

While flagging that these analytical results exist and are interesting, there are all concerned with maximizing the information content of the patterns stored that is most relevant to a reconstruction task. In this paper, we care about two different objective functions somewhat uncorrelated with maximizing information content: (i) Classification accuracy, where better performance can be achieved without trying to model the data manifold; (ii) Continual learning, where while we care about modelling the data manifold, we also want the formation of unique subnetworks that can output the correct classification label. In addition, SDM's analytical results assume that the patterns are random. For correlated real-world datasets, we need ways to quantify the complexity of the data manifold and a full exploration of this problem and solutions is beyond the scope of this work.

## D    STALE MOMENTUM

Imagine a neuron is activated for three training batches, then inactive for the next 10 batches, and finally activated two more times. During the first set of activations there are no problems, the neuron will update its weights and its momentum term, using up to date gradient information. During inactivity, problems begin as the inactive neuron will still update its weights and do so using an increasingly out of date momentum term. Problems continue when this neuron is activated again because the momentum term decays slowly and will significantly boost the gradient in an out of date direction.

These stale momenta are especially harmful in the Top-K competitive activation setting where a neuron needs to be the $k$ most active for at least one input to continue receiving gradient updates, otherwise it will permanently die. Using Stochastic Gradient Descent (SGD) that is free from any momentum term removes this problem. SGD with momentum works worse than SGD without momentum, but still better than Adam or RMSProp (Kingma & Ba, 2015; Geoffrey Hinton, 2012; Goodfellow et al., 2015).

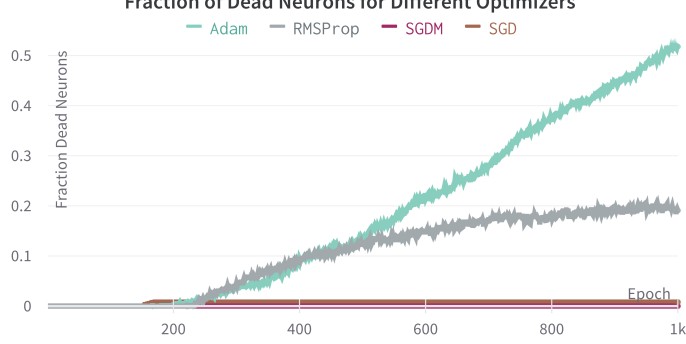

Figure 15: The GABA switch occurs by epoch ∼100 and Top-K is fully implemented by ∼200 epochs. At this point, the RMS and Adam optimizers begin continuously killing off neurons such that they are never activated for any training or validation data. These results are for training directly on CIFAR10 images. See the text for a discussion of training on ConvMixer embeddings.

We discovered the Stale Momentum effect when training our models directly on CIFAR10 images that have a much more complex data manifold than the ConvMixer embeddings. Fig. 15 shows that after the GABA switch, when Top-K is fully implemented around epoch 200, Adam and RMSProp continue killing off neurons (Kingma & Ba, 2015; Geoffrey Hinton, 2012). These neurons are dead

in that they are never activated by any training or validation data. Our results are robust across learning rates for the different optimizers.

While these dead neurons from Adam and RMSProp don't appear when pretraining on ImageNet32 embeddings, the Stale Momentum problem still leads to incorrect gradient updates that harm continual learning as shown in the ablations of Table 2. This is because fewer neurons are covering the data manifold and able to retain memories of separate tasks.

Notably, a ReLU network trained with Adam or RMSProp also kills off up to 95% of its neurons with only minor effects on train and validation accuracy when trained directly on CIFAR10 pixels. SGDM and SGD again do not kill off any neurons for this network.

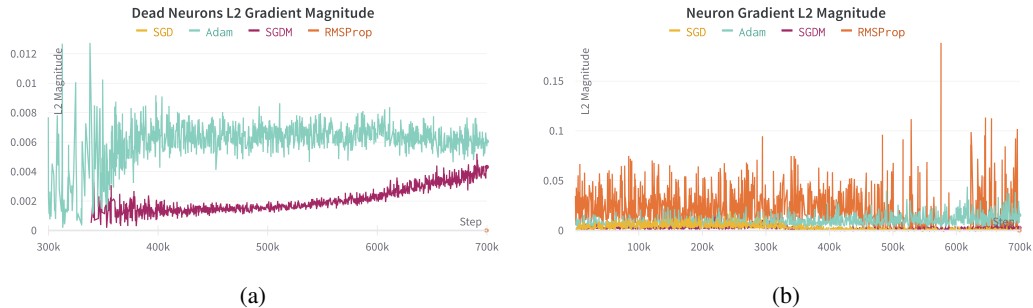

(a)                                                                (b)

Figure 16: **Empirical Stale Momenta Seen During Training. (a)** We take all neurons that are dead during a batch and record the mean $L^2$ magnitude that its weights experience during the update. Adam (teal) has the largest updates to dead neurons. Note that these magnitudes are still 5-10x smaller than the average for alive neurons. SGDM (purple) also updates dead neurons but to a lesser degree. SGD (yellow) and RMSProp (orange) are not visible because they do not apply any updates to dead neurons. We start our x-axis at training step 300k because this is when there are enough dead neurons for their gradient updates to be meaningfully calculated. **(b)** We track all gradient updates applied to ten random neurons during the course of training and pick a representative to display here. We track gradient updates starting right as the GABA switch occurs around epoch 100 and all training steps from this point on are shown along the x-axis. RMSProp (orange) has the highest variance and magnitude in gradient updates followed by Adam (teal).

We logged the gradients of neuron weights during training to confirm that neurons are receiving gradient updates even while they are dead as shown in Fig. 16a. This figure is initially confusing because, while Adam and RMSProp kill off the most neurons in Fig. 15, it is Adam and SGDM that keep updating dead neurons in Fig. 16a (Kingma & Ba, 2015; Geoffrey Hinton, 2012). Looking at every gradient update applied to a single neuron right after GABA switches in Fig. 16b provides a different perspective. Here, the highest variance and largest gradient magnitudes (as quantified by their $L^2$ norm) are produced by RMSProp and Adam. These gradient spikes appear particularly in cases where the neuron is dead for some time and then activated again.

To investigate further, we implemented each optimizer in a toy environment tracking a single weight that receives sparse gradient updates to see how the moving averages respond.[15] We were able to reproduce the much larger gradient spikes displayed by Adam and RMSProp upon the first few gradient updates after a period of quiescence. These results are shown in Fig. 17a, where we show the $\Delta_{\text{Optimizer}}$ term for each optimizer, independent of the learning rate. Initializing their respective terms shown in Eqs. 14 with zeros, we first introduce two gradient updates and then have periods of quiescence before injecting gradient updates.

In Fig. 17a we show four different gradient injection periods and that RMSProp has the largest response, followed by Adam and then SGDM, reproducing the gradient spikes seen in Fig. 16b. Fig. 17b explains these larger gradients by plotting the Adam and RMSProp numerator and denominators for only two gradient injection periods. Note in particular that for Adam, its numerator $m_t$ (dark orange) declines somewhat quickly, having a decay of $\beta = 0.9$, however, its denominator $\sqrt{v_t}$ (light

---

[15]Further details for this toy experiment can be found and reproduced in our github repository in the Jupyter Notebook in the notebooks/ directory with the filename StaleGradients.ipynb

orange) has a much slower decay of $\beta_2 = 0.999$, staying large for much longer and causing the new gradient inputs to explode in size.

Finally, Fig. 17c quantifies how large the gaps in gradient update magnitude are between the three optimizers and the actual gradient when we vary the time between gradient injections.

To summarize, we believe it is more the gradient explosions after quiescence rather than updating of neurons currently dead that causes Adam and RMSProp to kill off neurons as in Fig. 15. The denominators present in these optimizers, given in Eqs. 16 and 17 are to blame because they are such slow moving averages and inflate the sparse gradients.

The update rules for SGDM, Adam, and RMSProp are as follows where we let $\lambda$ be the learning rate, $\theta_t$ are the parameters at time $t$, $g_t$ is the gradient, and $\gamma = 0.9$ $\beta_1 = 0.9$, $\beta_2 = 0.999$, $\alpha = 0.99$ are hyperparameters for the various algorithms with their standard values.

$$\theta_t = \theta_{t-1} - \lambda \Delta_{\text{Optimizer}} \tag{14}$$

$$\Delta_{\text{SGD}} = g_t$$

$$\Delta_{\text{SGDM}} = \gamma \theta_{t-1} + g_t \tag{15}$$

$$\Delta_{\text{Adam}} = \frac{\hat{m}_t}{\sqrt{\hat{v}_t}} \tag{16}$$
$$\hat{m}_t = \frac{m_t}{1 - \beta_1}$$
$$\hat{v}_t = \frac{v_t}{1 - \beta_2}$$
$$m_t = \beta_1 m_{t-1} + (1 - \beta_1) g_t$$
$$v_t = \beta_2 v_{t-1} + (1 - \beta_2) g_t^2$$

$$\Delta_{\text{RMSProp}} = \frac{g_t}{\sqrt{v_t}} \tag{17}$$
$$v_t = \alpha v_{t-1} + (1 - \alpha) g_t^2$$

Note that these results are robust for any gradient updates $g_t < 1$, otherwise SGDM has larger gradient spikes than Adam. However, this is never the case in our training regime because of the $L^2$ normalization term for our weights and inputs which means that no weight or neuron activation value is ever larger than 1. If this were not the case then we predict that SGDM would have sufficiently large jumps in its gradients to also result in dead neurons.

**Practical Takeaways** Below we give our reasoning for why SGD is the best solution but needs careful hyperparameter tuning of the learning rate and GABA switch $s$. Sparse optimizers (Spa) are another alternative that should be investigated in future work.

Because SGD has no momentum term it does not suffer from stale momenta and is the most principled solution. However, lacking a momentum term, SGD is also the most sensitive to learning rate choice and can introduce either slow or poor convergence if it is too high or low, respectively. Setting the learning rate too low is particularly problematic in our setting with the excitatory period of the GABA switch because neurons can fail to receive enough gradient updates to move onto the data manifold, resulting in dead neurons. In Fig. 11 of App. B.3 we show how a learning rate that is too low will result in dead neurons that never learn the manifold before the GABA switch occurs.

Given these hyperparameter tuning requirements for SGD, we recommend using SGDM to pretrain SDMLP while using SGD for continual learning. SGDM pretraining does not result in dead neurons as shown in Fig. 15 and is more robust to choice of learning rate. For example in this instance of SGDM on CIFAR10 pixels, varying the learning rate from 0.01 to 0.09 did not affect either

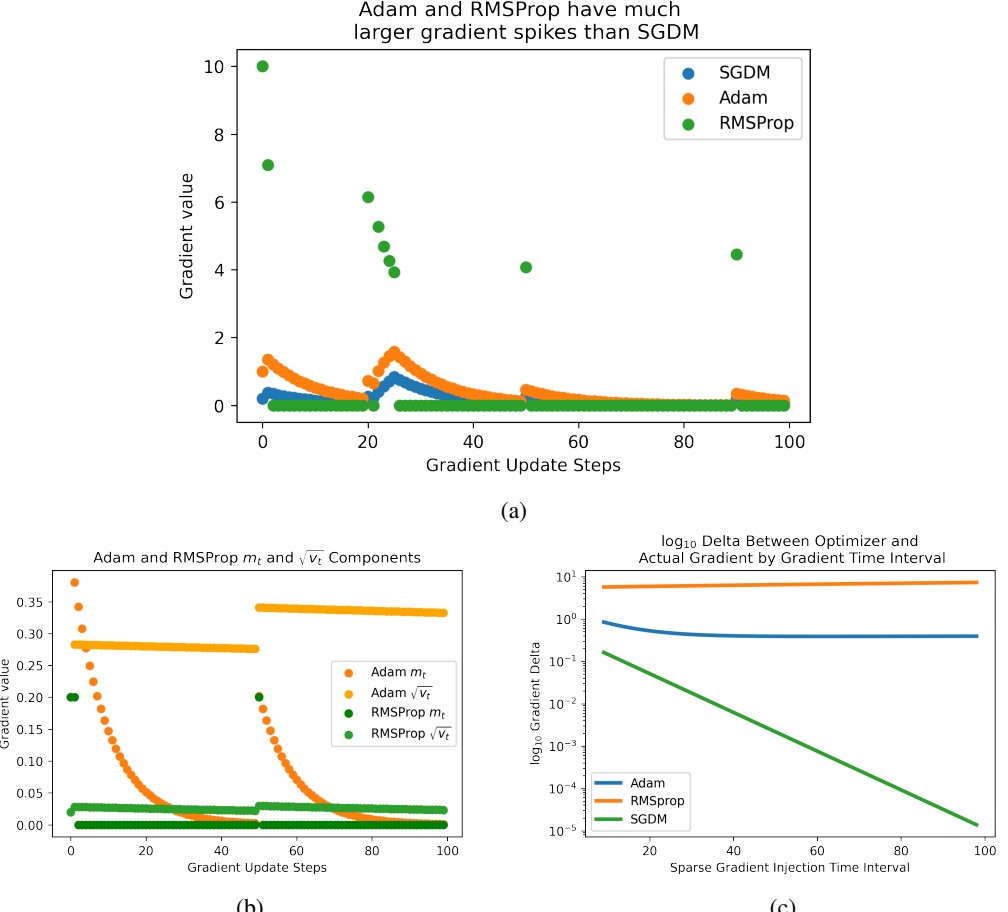

(a)

(b)

(c)

Figure 17: **Toy experiment of how Adam, RMSProp and SGDM introduce stale momenta. (a)** Gradient updates of 0.2 are injected at four different time points visible by the large RMSProp green dots. We first apply two gradient updates, then four, then one and one. Note how RMSProp amplifies the gradient magnitudes but does not apply any gradient update when the neuron is inactive. Meanwhile, Adam (orange) and SGDM (blue) will keep applying gradient updates that slowly decay over time. Also note that the Adam gradient magnitudes are larger than those for SGDM. **(b)** We inject gradients at single time points twice and observe how the numerators (darker color) and denominators (lighter color) for Adam and RMSProp change. The denominators decay very slowly. **(c)** We see how the delta between the actual gradient value of 0.2 and the gradient value applied by the different optimizers changes depending on the interval between the last gradient update. Note that the y-axis is $\log_{10}$. RMSProp has a $10/0.2 = 50$x amplified gradient and Adam is a $1/0.2 = 5$x. Note that SGDM without a denominator term is the only optimizer that over longer periods of time stops amplifying the gradient value. All results generalize for gradient values that are $< 1$.

convergence speed or kill off any additional neurons. Meanwhile, as shown in the ablations of Table 2, in the continual learning setting, SGD is the best choice.

Another alternative are sparse optimizers like "sparse Adam" which does not apply gradient updates to any neuron that is dead (Spa). However, failing to update the momentum term in a novel way will theoretically still result in what we believe to be the larger problem of exploding re-activated neuron gradients. The only way to use sparse Adam in Pytorch is to implement sparse weight layers and we leave it to future work to empirically test sparse momentum based optimizers.

This section lacks many citations because we are unaware of existing literature around other models in the sparse activation regime, using either Top-K or Mixture of Experts (Fedus et al., 2021; Roller et al., 2021; Shazeer et al., 2017) that discuss the stale momentum problem and issues with

momentum based optimizers.[16] We also could not find an academic citation for the sparse Adam implementation provided in PyTorch or the motivations for implementing it. Suggestively, in Tensor-Flow the algorithm is called "LazyAdam" and advertised as being advantageous for computational efficiency reasons without any mention of dead neurons. This is unsurprising in light of the fact that the dead neuron problem is only an issue in the sparse regime and even in this case can go unnoticed if validation accuracy is the only metric of interest. For example, training on CIFAR10 pixels with a ReLU network and Adam results in 95% dead neurons with minor effects to validation accuracy.

## E  TRAINING REGIME ABLATIONS

### E.1  TRAINING DIRECTLY ON IMAGE PIXELS

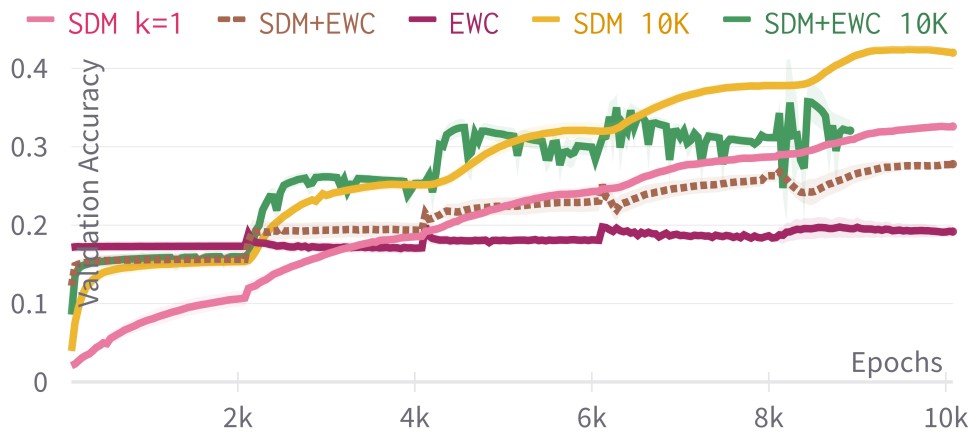

Figure 18: **Rank ordering of continual learning algorithms remains approximately the same for training directly on CIFAR10.** Here SDMLP with 10K neurons (yellow) actually does better than SDMLP+EWC (green) that was the best performer in the main text Fig. 3(a). However, we did not tune the hyperparameters for the EWC loss coefficient or the $\beta$ parameter for training directly on pixels. SDMLP+EWC with 10K neurons fails to terminate within the 15 hours of GPU time allocated however it is clear that SDM outperforms it. Results are for the 5 random seed splits of Split CIFAR 10.

We remove the ConvMixer preprocessing and train models directly on ImageNet32 pixels before testing their continual learning abilities on Split CIFAR10 pixels. This is a way to address concerns that the ConvMixer preprocessing was manipulating the CIFAR10 data manifold to make the continual learning task too easy. Fig. 18 shows the results and Table 3. We only tested the best performing models with reference to Table 1 and chose EWC over MAS because it gives the best model overall and otherwise performs similarly. We did not test the FlyModel because it assumes the presence of preprocessed latent embeddings rather than the 3072 dimensional CIFAR10 flattened image vectors. All training settings and parameters are kept the same as in the Table 1 tests.

Interestingly, all EWC methods fail to perform as well as they did before. This is likely because we did not tune our loss coefficient or $\beta$ parameter for training directly on pixels. As a result, the best performing model is now SDM with 10K neurons (yellow) instead of SDMLP+EWC (green). Also note that while SDMLP with $k = 1$ (pink) does worse, it fails to learn each task within the 2,000 epochs, e.g. at the end of 2,000 epochs the validation accuracy is $\sim$10% while the other methods maximize the accuracy within task of $\sim$20%. This means that SDM with $k = 1$ could potentially do better with more epochs or a higher learning rate.

---

[16]Interestingly, (Shazeer et al., 2017) modifies Adam to reduce its parameter count by setting $\beta_1 = 0$ and taking averages over $v_t$, both of which may have inadvertently helped avoid stale momenta.

We use the same 5 random seeds and 2,000 epochs per task. Training takes longer because the images are 3x32x32=3072 pixels rather than the 256 ConvMixer embeddings and the most complicated model, SDMLP+EWC with 10K neurons fails to terminate in the 15 hours of GPU time allocated to each run but makes it to the last learning task where it's performance can be inferrred.

**Table 3: Pixel Training - Split CIFAR10 Validation Accuracy**

| Method | Neurons | $k$ | Val. Acc. |
|---|---|---|---|
| SDMLP | 1K | 1 | 0.33 |
| SDMLP | 10K | 10 | **0.42** |
| EWC | 1K | NA | 0.19 |
| SDMLP+EWC | 1K | 10 | 0.28 |
| SDMLP+EWC* | 10K | 10 | 0.32 |
| Oracle | 1K | NA | 0.53 |
| Oracle | 10K | NA | 0.53 |

The most competitive models, pretrained on ImageNet32 and tested on Split CIFAR10 without any ConvMixer embedding. SDMLP+EWC with 10K neurons has a * to denote that it did not finish training. Average of 5 random task splits.

## E.2 NO PRETRAINING

**No Pretraining - Split CIFAR 10**

Figure 19: **Rank ordering of continual learning algorithms remains for no ImageNet32 pre-training.** Solid lines denote different algorithms. Dotted lines denote variants of an algorithm. SDMLP+EWC (green) with 10K neurons does the best then SDMLP with 10K neurons (yellow). The 1K algorithms all do approximately the same including EWC (purple), SDMLP+EWC (dotted brown), and SDMLP k=1 (pink). Results are for the 5 random seed splits of Split CIFAR 10. TopK and ReLU benchmarks both do poorly and were not run for the full set of random seeds or shown as a result.

Fig. 19 shows the results for the most competitive continual learning algorithms without any ImageNet32 pretraining and tested on the ConvMixer embedded Split CIFAR10. See Table 4 for the final validation results.

We train for 2,000 epochs on each task and 10,000 epochs in total. The GABA switch occurs well within the first task where $s = 5,000,000$ with GABA switching at epoch $\sim 50$ and Top-K being fully implemented by epoch $\sim 100$. Performances are lower than with the ImageNet32 pretraining, for example the best performing SDMLP+EWC is 84% versus the 86% in Table 1. Most notable is the worse performance for the 1K neuron models that lack sufficient capacity to learn the data manifolds of new tasks without the benefits of pretraining.

**Table 4: No Pretraining - Embedded Split CIFAR10 Validation Accuracy**

| Method | Neurons | $k$ | Val. Acc. |
|---|---|---|---|
| SDMLP | 1K | 1 | 0.56 |
| SDMLP | 10K | 10 | 0.77 |
| FlyModel | 1K | 32 | 0.69 |
| FlyModel | 10K | 32 | 0.82 |
| EWC | 1K | NA | 0.52 |
| SDMLP+EWC | 1K | 10 | 0.54 |
| SDMLP+EWC | 10K | 10 | **0.84** |
| Oracle | 1K | NA | 0.93 |
| Oracle | 10K | NA | 0.93 |

## F    ADDITIONAL DATASETS

### F.1    CIFAR10 EXTRA FIG. AND TABLE

Table 5 shows the Split CIFAR10 results for the 1K neuron setting. Fig. 20 shows the extent to which the SDMLP forgets previous tasks when compared to the baseline ReLU model. This figure gives extra context to the headline continual learning results of Fig. 3.

**Table 5: Split CIFAR10 - 1K Neurons - Validation Accuracy**

| Method | Neurons | $k$ | Val. Acc. |
|---|---|---|---|
| SDMLP | 1K | 1 | 0.70 |
| SDMLP | 1K | 10 | 0.63 |
| Top-K | 1K | 10 | 0.29 |
| FlyModel | 1K | 64 | 0.70 |
| MAS | 1K | NA | 0.69 |
| EWC | 1K | NA | 0.67 |
| SI* | 1K | NA | 0.34 |
| NISPA | ~1K | NA | 0.19 |
| L2 | 1K | NA | 0.23 |
| Dropout | 1K | NA | 0.21 |
| SDMLP+MAS | 1K | 10 | 0.83 |
| SDMLP+EWC | 1K | 10 | **0.83** |
| Oracle | 1K | NA | 0.93 |

**Split CIFAR10 Final Validation Accuracy** - We highlight the best performing SDMLP, baseline, and **overall performer** in the 1K neuron setting. Oracle was trained on the full CIFAR10 dataset. SI has a * denoting some of its runs failed. NISPA uses a three hidden layer network with 400 hidden units per layer but this is close in parameter count to one hidden layer with 1K neurons. All results are the average of 5 random task splits.

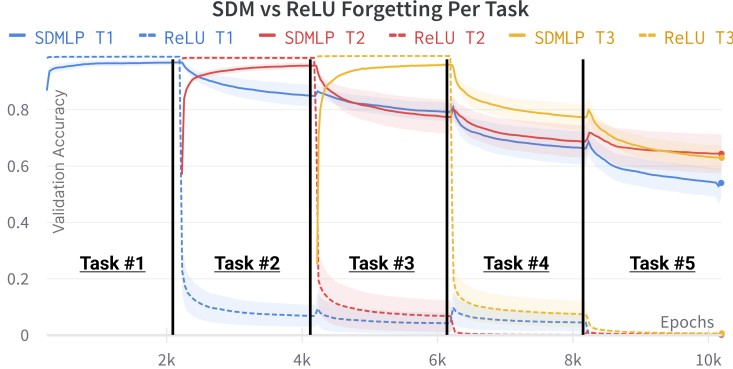

Figure 20: **SDMLP gradually forgets previous tasks.** Solid lines along the top show SDMLP forgetting previous tasks over time in comparison to ReLU that catastrophically forgets (dashed lines along the bottom). Validation accuracy is now computed within tasks and to avoid clutter we only show the learning curves for the first three tasks. Both plots use the average of 5 random seeds and error bars show standard error of the mean.

### F.2    CIFAR100

To better assess the true memory capacity of our models, we use the same pretraining on ImageNet32 and then test continual learning on 50 splits of CIFAR100, shown in Fig. 21 and Table 6. We allow

for 500 epochs per task and 25,000 epochs in total. We used three random seeds instead of five, the same hyperparameters found for CIFAR10, and only tested the best performers from Table 1.

It is with CIFAR100 that the SDM models including the FlyModel and SDMLP+EWC really shine over the regularization baselines. In the 1K neuron setting, SDMLP+EWC is the best performer with 42%. Meanwhile, in the 10K setting the FlyModel takes the lead getting 58% and SDMLP+EWC behind it at 51% against the oracle of 72%. The next closest baseline is 40% for MAS. The large performance jumps in the FlyModel going from 1K to 10K neurons for CIFAR10 and CIFAR100 highlight the importance of dimensionality for the fixed neuron addresses that perform a random projection. Interestingly, when looking at the validation accuracy on Task 1 for each of the models, which shows how it forgets the task over time, SDMLP+EWC performs better than the FlyModel, however the FlyModel retains later tasks better.

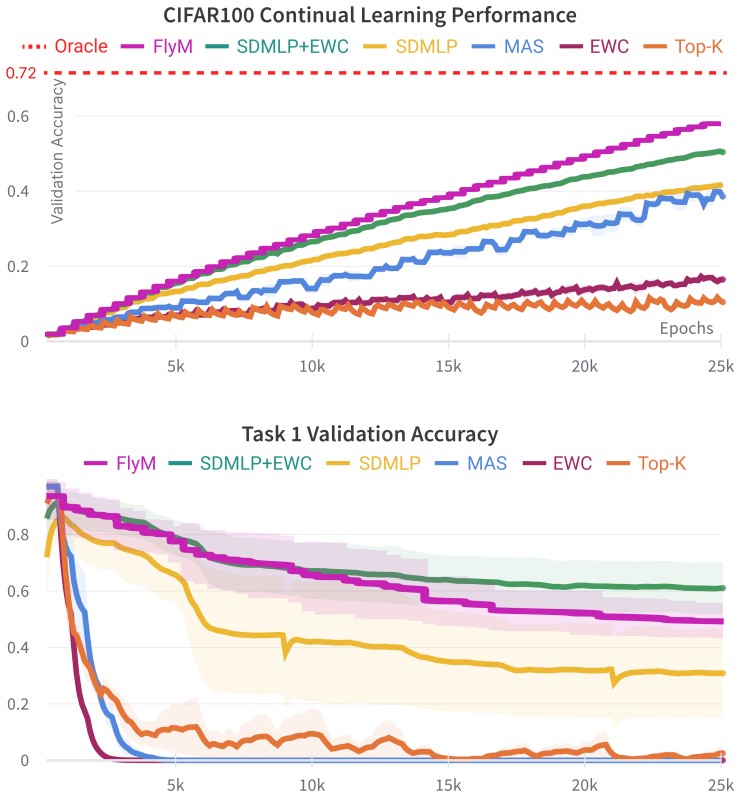

Figure 21: **Top - Rank ordering of continual learning algorithms remains similar for CI-FAR100.** The FlyModel with neurons (magenta) now outperforms SDMLP+EWC (green). The SDM algorithm does the next best (yellow) with MAS (blue) again close behind and then EWC (purple) and Top-K (orange) performing poorly. **Bottom - SDM methods are robust to forgetting.** The first task of CIFAR100 and its validation accuracy for this task over the entire course of training on 49 other tasks is shown for each method. SDMLP+EWC is the most robust to forgetting (green), then the FlyModel (magenta) and SDMLP (yellow). It is interesting that the SDMLP+EWC forgets less for this first task than the FlyModel. Top-K shows some retained memory but only a small amount while MAS and EWC both catastrophically forget over time within $\sim 5$ tasks (2,500 epochs).

**Table 6: Split CIFAR100 Validation Accuracy**

| Method | Neurons | $k$ | Val. Acc. | Method | Neurons | $k$ | Val. Acc. |
|--------|---------|-----|-----------|--------|---------|-----|-----------|
| SDMLP | 1K | 1 | 0.32 | SDMLP | 10K | 10 | 0.43 |
| SDMLP | 1K | 10 | 0.39 | FlyModel | 10K | 32 | **0.58** |
| Top-K | 1K | 10 | 0.11 | MAS | 10K | NA | 0.40 |
| FlyModel | 1K | 32 | 0.36 | EWC | 10K | NA | 0.16 |
| MAS | 1K | NA | 0.24 | SDMLP+EWC | 10K | 10 | 0.51 |
| EWC | 1K | NA | 0.12 | | | | |
| SDMLP+EWC | 1K | 10 | **0.42** | Oracle | 10K | NA | 0.72 |

We bold the best performing model within the 1K and 10K neuron settings. Oracle was trained on the full CIFAR10 dataset. Average of 3 random task splits.

### F.3 SPLIT MNIST

For the sake of completeness and providing an easier benchmark for other methods, we use the Split MNIST dataset but keep the class incremental setting. Results are presented in Table 7 and approximate the general rank ordering of performance whereby the FlyModel with 10K neurons does the best, SDMLP+EWC comes second, then SDMLP on its own and then the parameter importance regularization methods. We did extensive hyperparameter tuning of the EWC and MAS regularization coefficients and $\beta$ parameter along with the FlyModel parameters (App. G.2).

We do not use any pretraining and train directly on MNIST pixels. We train for 500 epochs on each task meaning 2,500 epochs in total. We use just three random seeds to initialize model weights but always use naive split where task 1 contains digits 0 and 1, task 2 contains digits 2 and 3, etc.

As one of our benchmarks we used Active Dendrites (Iyer et al., 2022). We used the code provided but found that it failed to generalize beyond the easier Permuted MNIST task incremental benchmark used in the paper.

NISPA has a * in the number of neurons column of Table 7 because it used three hidden layers of 400 neurons each as in its original implementation. This is approximately the same number of parameters as the 1K neurons in a single layer (637600 vs 794000 ignoring bias terms).[17]

**Table 7: Split MNIST Validation Accuracy**

| Method | Neurons | $k$ | Val. Acc. |
|---|---|---|---|
| SDMLP | 1K | 1 | 0.69 |
| SDMLP | 1K | 10 | 0.53 |
| SDMLP | 10K | 10 | 0.53 |
| FlyModel | 1K | 64 | 0.77 |
| FlyModel | 10K | 32 | **0.91** |
| EWC | 1K | NA | 0.61 |
| EWC | 10K | NA | 0.67 |
| MAS | 1K | NA | 0.49 |
| MAS | 10K | NA | 0.58 |
| SI | 1K | NA | 0.36 |
| Top-K | 1K | 10 | 0.22 |
| NISPA | * | NA | 0.40 |
| Active Dendrites | 1K | NA | 0.20 |
| ReLU | 1K | NA | 0.21 |
| SDMLP+EWC | 1K | 10 | 0.83 |
| SDMLP+EWC | 10K | 10 | 0.86 |
| Oracle | 1K | NA | 0.98 |
| Oracle | 10K | NA | 0.99 |

Models trained directly on MNIST pixels and without any pretraining. We bold the best performing method. We run these results on three random seeds and using 500 epochs for each split (this is much higher than other baselines (Hsu et al., 2018)).

Interestingly, the SDMLP with 1K neurons does better than with 10K neurons. We also found that the SDMMLP+EWC models were still learning the last task at the end of training. However, increasing training times did not result in the last task being learnt better, suggesting that the model has run out of new neurons to avoid overwriting, or that the regularization coefficient for EWC should be reduced for the combination with SDM.[18]

---

[17]NISPA is 80% weight sparse here but this aids its continual learning and we do not consider the activation sparsity of the Top-K models in our parameter counts either. It also takes many more FLOPs to train because of the iterative weight growth and pruning.

[18]This effect was also observed for the FashionMNIST results in App. F.4.

### F.4 SPLIT FASHIONMNIST

We also evaluate our most successful models and baselines on the FashionMNIST dataset that has ten different classes of fashion items as grayscale 28x28 images. This task is slightly harder than MNIST with our oracles getting 90% instead of 99% for MNIST. Table 8 shows these results as the average of 3 random seeds used to initalize the model weights. The rank ordering of results agrees with that of Split MNIST Table 7.

**Table 8: Split FashionMNIST Validation Accuracy**

| Method | Neurons | $k$ | Val. Acc. |
|---|---|---|---|
| SDMLP | 1K | 1 | 0.73 |
| SDMLP | 1K | 10 | 0.53 |
| SDMLP | 10K | 10 | 0.52 |
| FlyModel | 1K | 64 | 0.67 |
| FlyModel | 10K | 64 | **0.76** |
| EWC | 1K | NA | 0.68 |
| EWC | 10K | NA | 0.72 |
| MAS | 1K | NA | 0.33 |
| MAS | 10K | NA | 0.34 |
| Top-K | 1K | 10 | 0.23 |
| ReLU | 1K | NA | 0.21 |
| SDMLP+EWC | 1K | 10 | 0.74 |
| SDMLP+EWC | 10K | 10 | 0.72 |
| Oracle | 1K | NA | 0.90 |
| Oracle | 10K | NA | 0.90 |

Models trained directly on FashionMNIST pixels and without any pretraining. We bold the best performing method.

## G BASELINE IMPLEMENTATIONS

### G.1 BETA COEFFICIENT FOR ELASTIC WEIGHT CONSOLIDATION AND SYNAPTIC INTELLIGENCE

The paper (Hsu et al., 2018) considers a number of baseline continual learning algorithms with different continual learning settings. This includes the most realistic class incremental setting that we use. The paper also open sourced their implementations of these different algorithms and found that the regularization methods of EWC, MAS and SI (Kirkpatrick et al., 2017; Aljundi et al., 2018; Zenke et al., 2017) all catastrophically forget Split MNIST.

However, in our hands by more carefully tuning the loss coefficient for the regularization term in the loss function we were able to improve performance. Tuning the loss coefficient for EWC and SI did not increase their performance. But when we looked at the EWC and SI learning dynamics, we found that because they were getting 100% accuracy on each task split, there was no gradient from the loss that could be used to infer the importance of each weight to be regularized for future tasks. In order to give the model gradient information, we modified its cross entropy loss, introducing a $\beta < 1$ coefficient that made the model less confident in its prediction. Formally:

$$p_i = \frac{\exp\left(\beta l_i\right)}{\sum_{i=1}^{o} \exp\left(\beta l_i\right)}, \tag{18}$$

where $l_i \in \mathbb{R}$ are the real valued outputs (logits) for each of the $o$ output classes, indexed by $i$. And when put through the softmax distribution this gives a probability distribution where $\sum_{i=1}^{o} p_i = 1$. In the original algorithm of (Hsu et al., 2018), $\beta = 1$ by not being a hyperparameter to tune and this $\beta = 1$ is large enough that it results in the model output for the correct class getting a probability of $\sim 1$ with values of 0 for all other classes, giving a loss of 0 and no gradient. By having $\beta < 1$, we force our model to be less confident in its prediction of the correct class, creating a loss and gradient information to infer which parameters are important. Note that this $\beta$ term is only used when inferring parameter importance, not when training the model. By tuning the $\beta$ value in conjuction with the regularization loss coefficient, we were able to avoid catastrophic forgetting and again exceed the baseline reported in (Hsu et al., 2018) for EWC and SI. While this is another hyperparameter that must be tuned, and is not a modification mentioned in the original algorithms of (Kirkpatrick et al., 2017; Zenke et al., 2017), or other literature that we are aware of, it is a straightforward modification that boosts performance and retains the original essence of the algorithm. Therefore, we feeling that it is an appropriate modification to create more meaningful baselines.

**Table 9: Improved Regularization Baselines on Split MNIST**

| Name | Loss Coef. | $\beta$ | New Val. Acc. | Original |
|------|-----------|---------|---------------|----------|
| EWC (Kirkpatrick et al., 2017) | 200 | 0.005 | 63.23 | 19.80 |
| SI (Zenke et al., 2017) | 1500 | 0.005 | 35.76 | 19.67 |
| MAS (Aljundi et al., 2018) | 0.5 | NA | 24.99 | 19.52 |
| L2 (Goodfellow et al., 2014) | 10 | NA | 36.77 | 22.52 |

Testing a 1,000 neuron single hidden layer MLP on Split MNIST with 10 epochs per split we get the final validation accuracies after hand trying a few different hyperparameters shown in Table 9.[19] We show the hyperparameters used and present results from the class incremental setting of Table 2 in (Hsu et al., 2018) for comparison.

Note that the number of epochs here is fewer than in the full Split MNIST analysis of App. F.3. This is to relate our results to other baselines that use only a small number of epochs per task (Hsu et al., 2018). If you compare Table 9 with Table 7 of F.3 you will see that training for more epochs does affect the performance of EWC and MAS.

---

[19]For our full experiments using Split CIFAR10 in the pretraining setting shown in Table 1, we do more extensive Bayesian hyperparameter searches to maximize the performance of each baseline.

## G.2 FLYMODEL PARAMETERS

The FlyModel is unique in being trained for only one epoch and not using backpropagation. The authors outlined a method for the model to be trained for more than one epoch on each task, however, this introduces synaptic decay that would likely reduce performance via forgetting. It was never implemented in (Shen et al., 2021) and so we present the strongest version of the algorithm trained for one epoch on each task here.

We used the parameters outlined in (Shen et al., 2021) as a starting point. Because the dimensionality of our ConvMixer embeddings is 256, half of the 512 used in the original paper, we were able to cut in half the number of Kenyon cells from 20,000 to 10,000, conveniently fitting the number of neurons considered in our other experiments. We also varied the number of projection neuron connections between 64 and 3, finding that 32 performed better in both the 10,000 and 1,000 neuron settings. We experimented with the learning rate and found that the value of 0.005 worked best for the MNIST and CIFAR10 experiments. For CIFAR100, we got better performance using a learning rate of 0.2 but only for the 10,000 Kenyon cell model. The best results for each run across the random seeds is what is presented in the text.

## H INVESTIGATING DIFFERENCES IN CONTINUAL LEARNING ABILITIES

We investigated why SDM is robust to catastrophic forgetting while baseline models and SDMLP models that ablate specific features all fail.

To summarize our findings:

- ReLU - neurons never learn to tile the data manifold and every neuron is activated by almost every input.
- Top-K - the lack of $L^2$ normalization and the use of a bias term means neurons don't tile the data manifold. This means they are not only activated by more tasks, forgetting previous ones. Additionally, there are more dead neurons reducing memory capacity.
- SDM Top-K Mask - masking leads to more dead neurons and the neurons that are alive being more polysemantic, failing to have subnetworks that avoid being overwritten by future tasks.
- SDM No $L^2$ Norm - there are few neurons with massive weight norms that are active for up to 50% of all inputs. These neurons are active for all tasks, resulting in memory overwriting and catastrophic forgetting.

During Split CIFAR10 continual learning we checkpoint our model every 100 epochs, training on each data split for 300 epochs for a total of 1,500. We also track the number of times that each neuron is active across the entirety of training and visualize this on the $\log_{10}$ plots shown in Fig. 22.

Fig. 22a compares SDM to Top-K and ReLU where it is clear that ReLU neurons (green) are all activated many times without the Top-K activation function. While Top-K has fewer activations (orange), its neurons lack the bimodal distribution of SDM (blue) that we believe corresponds to the unique subnetworks only activated for a specific task. Fig. 22b looks at SDMLP ablations that use a Top-K Mask instead of subtraction and no $L^2$ normalization. Note the small blip of neurons for no $L^2$ norm (green) that corresponds to the greedy neurons always in the Top-K for all tasks. Also note that the Top-K mask (orange) is slightly shifted towards larger activation values.

For all of our analysis that follow we do not show dead neurons. These dead neurons are only really a problem for Top-K that has 55% dead neurons and 18% for SDM without an $L^2$ norm.

To look at how specialized each neuron is to specific tasks and data classes, we take the final models after Split CIFAR10 continual learning and pass all of the CIFAR10 training data through them, recording which neurons are in the Top-K for each input. For each neuron, we count the number of times it is active for each of the 10 input classes and use this to create a probability distribution over the classes the neuron is activated by. We take the entropy of this distribution as a metric for how polysemantic each neuron is. We plot the entropy of each neuron against the percentage of time it is in the Top-K in Fig. 23. We also combine these two metrics to weight each neurons' entropy by the amount of time it is in the Top-K to compute the average entropy of activated neurons and

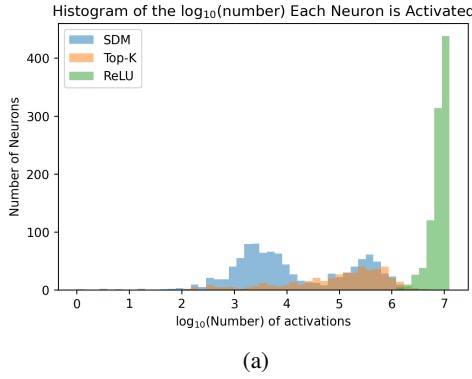 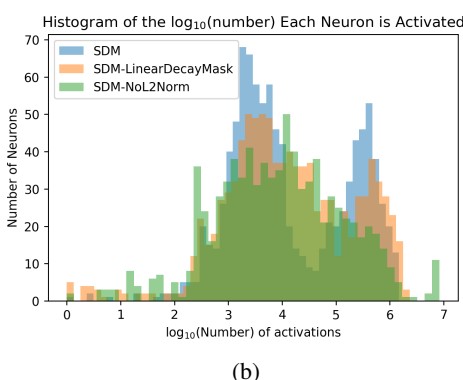

| (a) | (b) |

Figure 22: **SDM Neurons are more specialized than Top-K or ReLU.** We track the number of times every neuron is activated during continual learning and present this as a histogram with a $\log_{10}$ scale. **(a)** We compare SDM (blue) to the Top-K (orange) and ReLU (green) baselines. **(a)** We compare SDM (blue) to two ablations, SDM with a Top-K mask (orange) and SDM without $L^2$ normalization (green).

present this in Table 10 alongside the continual learning performance of each method. There is a clear inverse correlation between the average entropy of activated neurons and continual learning.[20]

**Table 10: Mean Neuron Entropy Weighted by Top-K Presence**

| Name | Mean Weighted Entropy | Val. Accuracy |
|------|----------------------|---------------|
| SDMLP | 0.99 | 0.54 |
| SDMLP Linear Mask | 1.36 | 0.35 |
| SDMLP No $L^2$ Norm | 1.96 | 0.20 |
| Top-K | 1.48 | 0.29 |
| ReLU | 2.25 | 0.21 |

Fig. 23 and the summary in Table 10 effectively convey the subnetwork formation by SDM that enables strong continual learning performance. SDM (Fig. 23a) has the lowest mean entropy of activated neurons followed by SDM with the Top-K mask (Fig. 23b). The SDM with Top-K Mask Fig. 23b shows how the most polysemantic (highest entropy) neurons are also the most active which will result in forgetting across tasks. Fig. 23c strikingly shows 13 "greedy" neurons in the top right that are not only highly polysemantic but active for many inputs (compare the y-axis going up to 6% compared to 1.4% for SDM). Top-K in Fig. 23d looks like the SDM with Top-K Mask but the average polysemantism of each neuron is much higher. Finally, the ReLU network in Fig. **??**, while haved a very even distribution of neuron activations, having the lowest y-axis range of 0.16%, the neurons are all highly polysemantic.

Notably, the fact that SDM in Fig. 23a does not have every neuron activated and in the Top-K the same percentage of the time means that it's neurons do not perfectly tile the data manifold in proportion to the density of the data. However, this may be because we are optimizing the network for classification performance instead of reconstruction loss.

---

[20]Note that in these experiments we used the SGDM optimizer instead of SGD. This means the validation accuracies for SDM and the linear mask are lower than they otherwise would be, however, we believe the insights we drawn here are unaffected and SGD would have just resulted in fewer dead neurons.

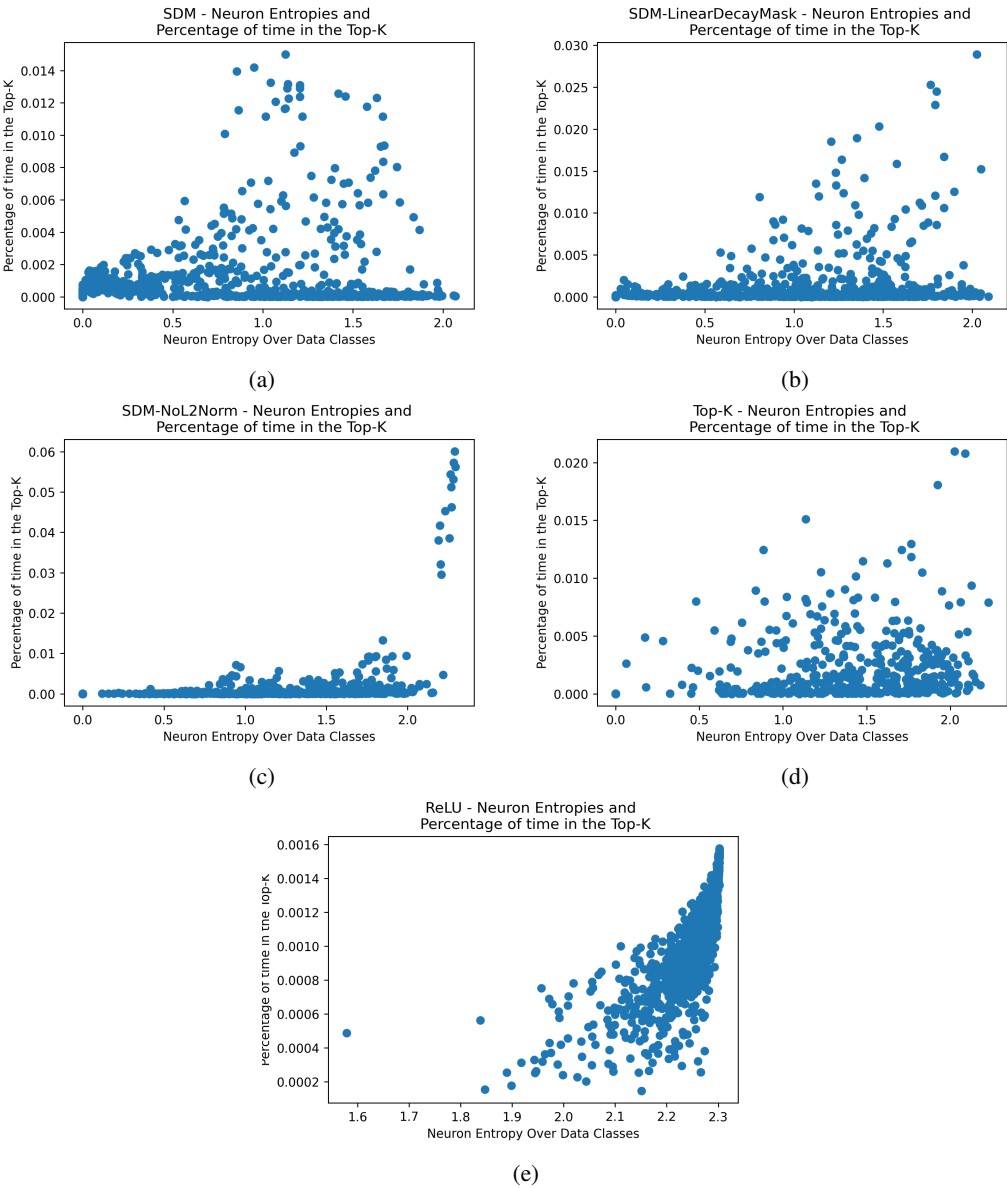

Figure 23: **SDM Neurons are specialized and participate democratically in learning.** Plotting the entropy of each neuron (the distribution of data classes that the neuron is activated by) against the percentage of time that it is in the Top-K for all inputs.

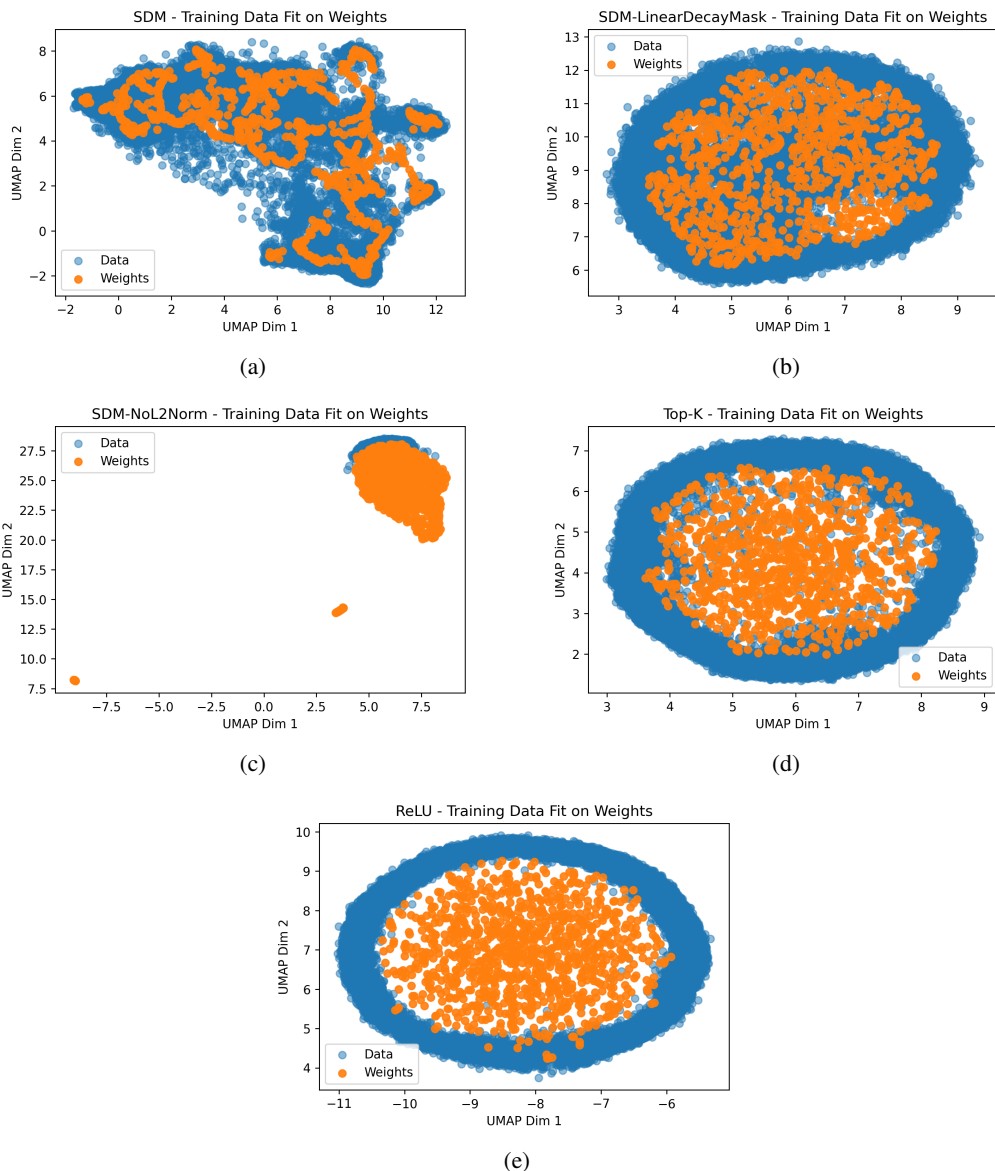

Figure 24: **CIFAR10 projected onto a UMAP embedding of the SDM weights tiles the manifold the best. (a)** is shown as Fig. 6 of the main text and the only approach that manages to tile the manifold such that the UMAP plot has meaningful structure.

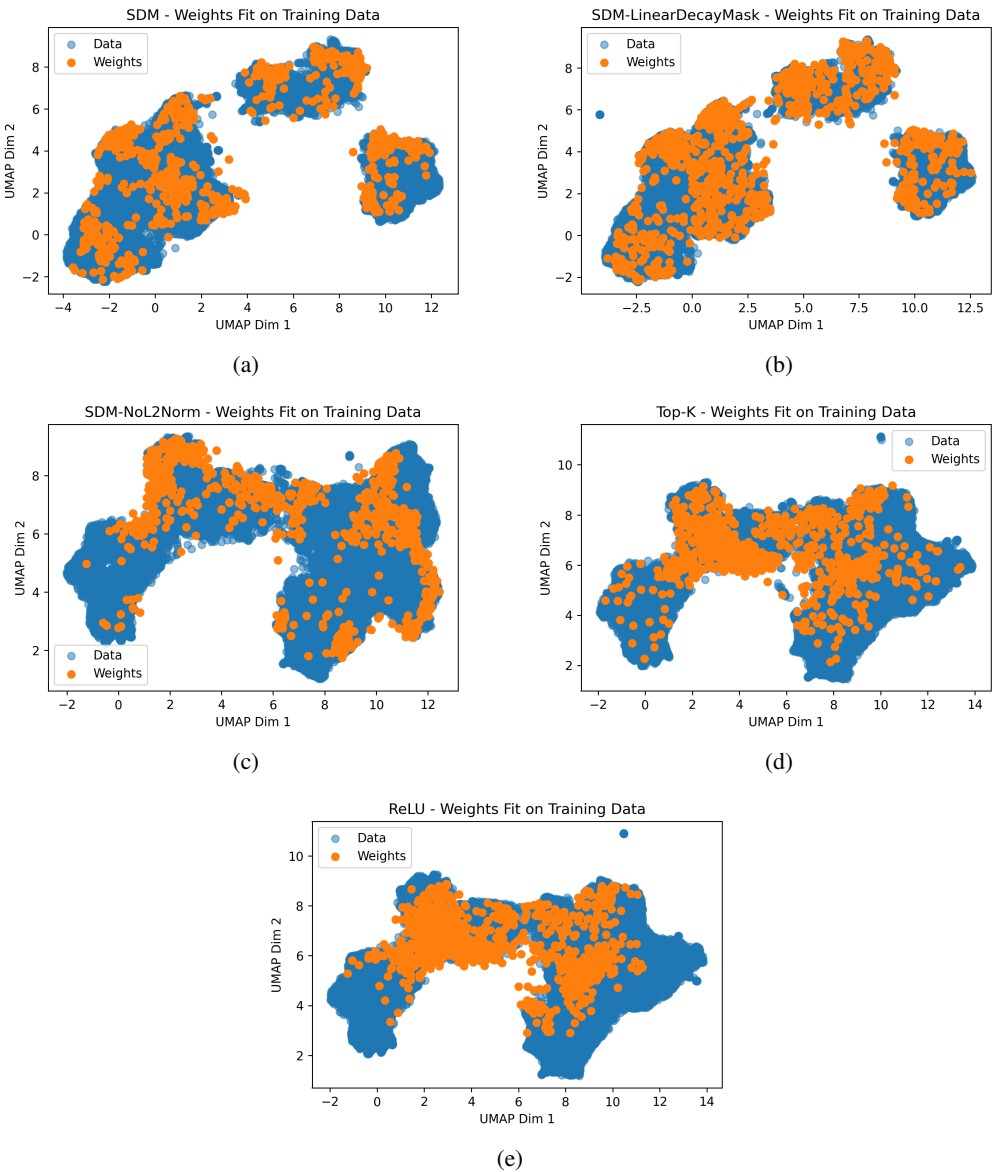

Figure 25: **SDM Weights Tile the CIFAR10 UMAP Embedding the Best.** The two SDMLP models in the top row (**(a)** and **(b)**) do the best tiling of the manifold. This is unlike the previous Fig. 24 where only SDM in **(a)** showed good tililng. The CIFAR10 data manifold looks different here for the top row, forming three distinct blobs because of the $L^2$ normalization operation.

We present a number of additional plots that show the manifold tiling abilities of SDM in comparison to the other methods. Fig. 24 fits the UMAP projection on the weights of the pretrained on ImageNet32 models and then uses this projection for the CIFAR10 data. Fig. 25 does the inverse where it fits a UMAP (McInnes & Healy, 2018) projection to ConvMixer embedded CIFAR10 training data and uses this projection for the trained model weights. Both Fig.s show that SDM learns to tile the data manifold most effectively with the full SDMLP being the only method to tile the manifold in both figures. This shows that upon pretraining, the neurons of SDM have learnt to differentiate across the manifold of general image statistics, forming subnetworks that will be useful for continual learning.[21]

We emphasize that while there is manifold tiling and subnetwork formation, the ImageNet32 pretraining does not result in SDM already knowing the CIFAR10 data or perfectly tiling the CIFAR10 manifold. Fig. 27 shows a UMAP plot with the projection fit to the SDMLP trained directly on CIFAR10 pixels and projecting the same CIFAR10 data, note how much tighter the manifold tiling is here. Further evidence of manifold tiling is evident in this pixel based training where in Fig. 26 we take the weights of ten random SDMLP neurons and reshape them into 3x32x32 dimensions to reveal the neurons have specialized to specific classes in the data. This is in stark contrast to the weights of the ReLU neurons shown adjacently. SDM neurons specialize to not only specific image classes but even specific examples within the class, analogous to the hypothetical "grandmother neuron" (Gross, 2002; Quiroga et al., 2005). This figure shows the trained weights of randomly chosen neurons from an SDMLP trained on CIFAR10 pixels and reshaped into their image dimensions. A bird, frog, and multiple deer and horses are visible.

Note that if we directly visualize these $L^2$ normalized weights and our $L^2$ normalized CIFAR10 images, they all look black because the pixel values denoting RGB colors are between 0 and 1 while our normalization makes them much smaller. We rescale our weights back into pixel values by first subtracting the minimum pixel value $\mathbf{x} - \min(\mathbf{x})$ and then multiplying $\mathbf{x} * 1/(\max(\mathbf{x}))$ that is standard practice.

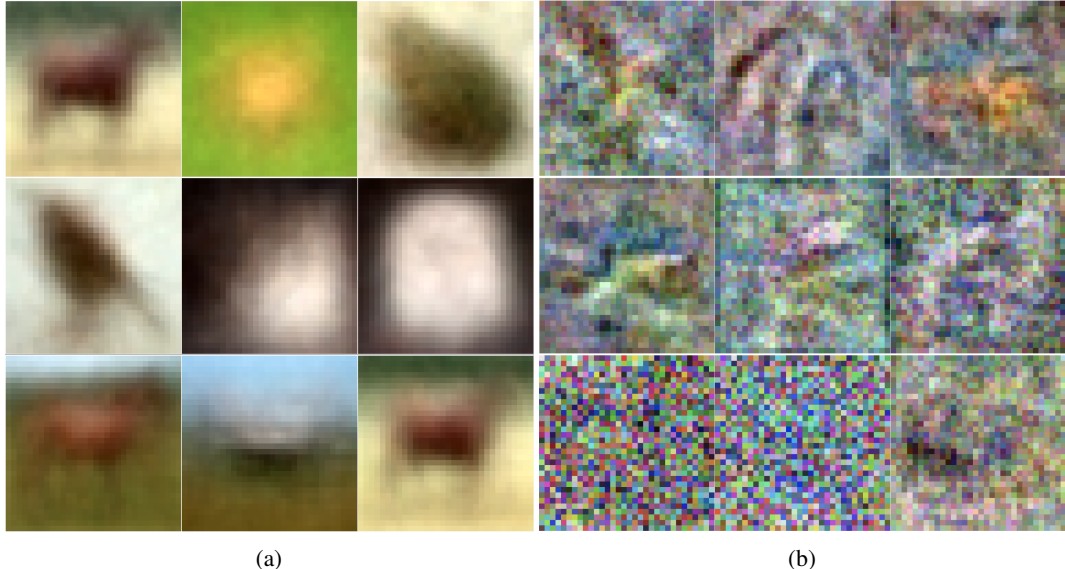

(a)                                          (b)

Figure 26: **Learned Neuron Weights for SDM (a) versus ReLU (b).** We train our models directly on CIFAR10 and visualize the weights of nine randomly chosen neurons by reshaping them into their image dimensions. SDM results in significantly more interpretable receptive fields for the neurons.

To fully emphasize that these neurons and their interpretable receptive fields are on meaningful parts of the manifold, we take a neuron that has a frog receptive field and show where it is located on the manifold. We then scan the CIFAR10 training data and take the 3 images that maximally activate

---

[21]The fact that the neurons are unique and dispersed across the data manifold, when combined with the Top-K activation function, ensures that subnetworks of neurons of will be active for different input classes.

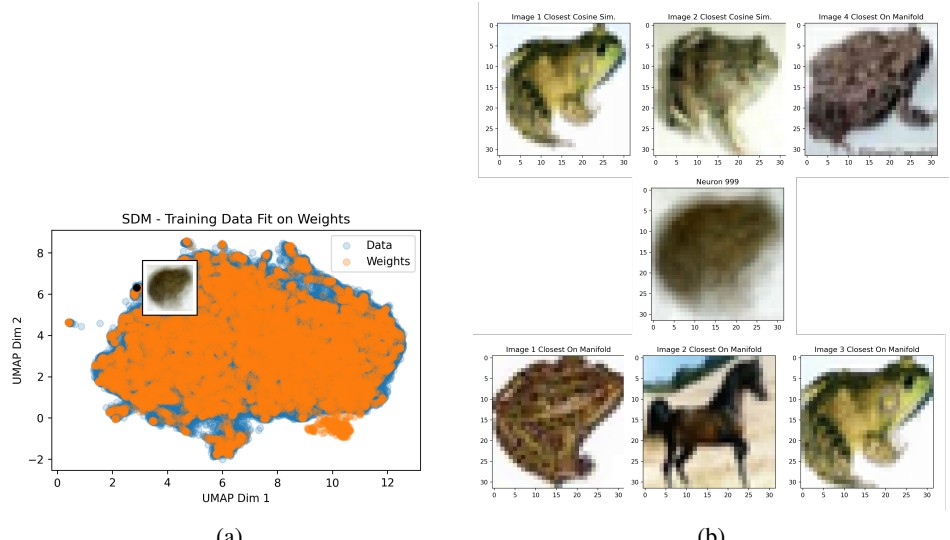

(a)                                                    (b)

Figure 27: **Analyzing neuron receptive fields of SDM in relation to the data manifold.** **(a)** A UMAP plot fit on neuron weights trained on CIFAR10 pixels (orange) with the CIFAR10 data (blue). The black dot in the top left shows the location of the neuron weights that give the frog image displayed as an inset. **(b)** The neuron weights are shown in the center, in the top row above are the three CIFAR10 images that maximally activate this neuron showing that the neuron weights learn a superposition of similar frog images. In the bottom row are the three closest images determined by euclidian distance on the UMAP embedding. This embedding will be less precise but still shows that the manifold largely captures similar classes, aside from the horse.

this neuron, plotting them in the top row of Fig. 27. We also look at the UMAP embedded images that have the smallest euclidian distance to the UMAP embedding of our neuron weights and show the closest three images. This UMAP apporach does contain a horse but this approach will be less precise due to the UMAP embedding mapping the 3072 dimensional images into 2 dimensions.

We use our neuron UMAP embeddings to visualize the number of times that each neuron is activated in Fig. 28. This uses the same overall activation values across continual learning presented in the histograms of Fig. 22 with yellow indicating the most activations and purple the fewest.

Finally, we show the data manifold tiling of SDM in Fig. 29 where we assign each CIFAR10 class a different color in the left plot and then show the positions of each neuron assigning the color that activates this neuron the most.

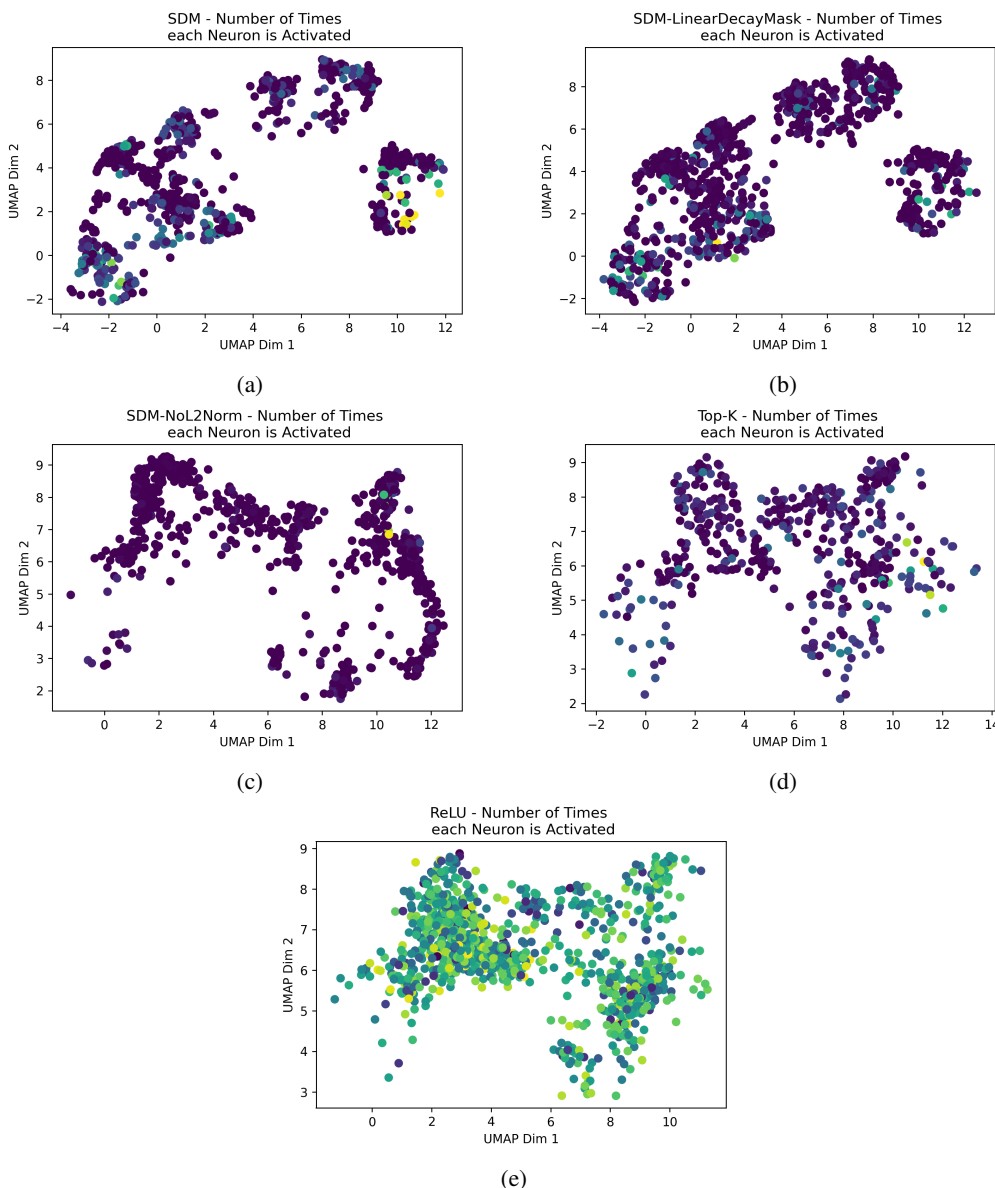

Figure 28: **Neurons colored by the number of times they have been activated.** This plot makes it clear how SDM with no $L^2$ norm (c) only activates a few neurons to do all of its learning, resulting in catastrophic forgetting. Meanwhile, ReLU in (e) activates almost all of its neurons. All models that use Top-K: SDM (a), SDM with Top-K masking (b) and Top-K (d) activate subsets of neurons making it harder to distinguish their learning dynamics without the additional analyses presented in this section.

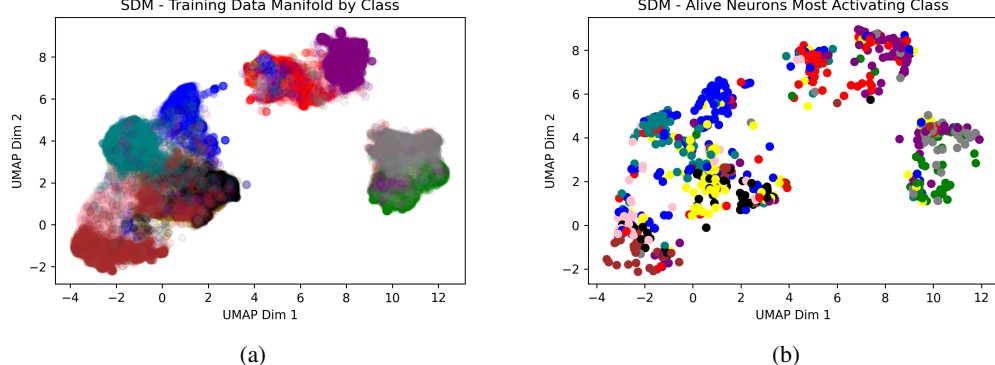

(a)                          (b)

Figure 29: **SDM neurons tile the regions of the manifold that they are most activated by.** We use the same UMAP plot as in Fig. 25 but color the data by its class in **(a)** and the neurons by the class that activates each the most in **(b)**.

