# OpenReview forum: "Sparse Distributed Memory is a Continual Learner"
_ICLR.cc/2023/Conference — ICLR 2023 poster_

### Official Review · Reviewer_HvB9 · 2022-10-23

**Confidence:** 3
**Correctness:** 3
**Technical Novelty And Significance:** 2
**Empirical Novelty And Significance:** 2
**Recommendation:** 5

**Clarity, Quality, Novelty And Reproducibility:**

- C: the paper is fairly clear and understandable at a first read. It involves a lot of parallelism with neuroscience terms and literature, that might not be easy to follow for all readers. However, the overall model is explained clearly.
- Q: the research seems conducted with good quality, in terms of literature review, technical formulation and experimental validation. Some issues in the latter point, as expressed above, emerge with the pretraining protocol used on CIFAR-10.
- N: beyond the whole biologically inspired motivation, the technical novelty of the paper reduces to the use of a top-k activation function in MLP hidden layers. In this perspective, the novelty is quite limited. Other contributions such as the annhealing of the top-k and the use of SGD without momentum can be considered tricks and implementation details, that don't add to the technical contribution of the paper.
- R: the authors illustrate in the paper and in the appendix all implementation details and hyperparameters needed to reproduce their work. Moreover, they release an implementation of their model in an anonymous repository, which is remarkable.

**Strength And Weaknesses:**

- +The paper overall well written, one read is sufficient to the reader to grasp the main ideas and contributions.
- +The authors remarkably place the proposed model within the literature, with a lot of references to both neuroscience and machine learning prior works.
- +The authors present a huge appendix that contains a lot of material useful for a deeper understanding of their research.
- -The motivation that the authors highlight for the employment of SDMs is only biological plausibility. Although I acknowledge that as a good trait, the text does not convey the reason why, technically, the proposed model avoids catastrophic forgetting. My personal interpretation is that, given that the Top-K activation functions create sparse activation patterns, the corresponding back-propagated gradient is also sparse, and less prone to overwrite important knowledge learned during prior tasks. The authors should provide more intuitive explanations about why the proposed model is less prone to forgetting in technical terms, beyond biologically-inspired considerations.
- -The paper describes how both read and write operations are performed in a SDM. However, the parallelism between SDM and a one-hidden-layer MLP is drawn only in the case of read operations (Eq. 3 and Eq. 2). How does the writing mechanism intervene in the authors model? Is the gradient-based learning of the MLP parameters considered the writing algorithm? If so, gradient-based updates do not conform with the writing mechanism for SDMs (Eq. 1). Can we still consider the model a SDM in that case? And if so, why?
- -The main experiment on CIFAR10 presents encouraging results, where the proposed SDM trained incrementally learns better than other models that specifically intervene to reduce catastrophic forgetting. However, the setting is quite arbitrary, as the authors use pretrained imagenet embeddings and an SDM on top of that. This raises two concerns. i) given that imagenet embeddings already separate CIFAR10 classes (as the cifar10 class space is a subset of the imagenet class space), is this an interesting learning setting? ii) are all compared methods in Tab. 1 using the same protocol? In my opinion, the case where *both* the embeddings and the classifiers need to be learned is the most interesting scenario for studying continual learning problems. Can the proposed models be extended to arbitrary backbones (e.g. ResNets), where every pair of consecutive convolutional layers can be considered as a SDM?
- -Table 1 would benefit a oracle where all classes are trained jointly, to put the results into perspective.
- -In Table 2, different optimizers showcase very different performances on the task. Are all baselines in Tab. 1 comparable, meaning that they all use SGD without momentum as an optimizer?

**Summary Of The Paper:**

This paper is concerned with continual learning, and it is interested in a class of biologically plausible models to overcome catastrophic forgetting. Specifically, the authors focus on the Sparse Distributed Memory (SDM) model, a long standing memory model in computer science. The paper highlights a parallelism between SDMs and a single-hidden-layer Multi Layer Perceptron (MLP) and introduce some tweaks to the latter for empowering continual learning capabilities. Specifically, the authors employ in the hidden-layer a top-K activation function, resembling the sparse read operation in SDMs. Since its naive application introduces the problem of dead neurons, an annhealing strategy is introduced for reducing k (the number of active neurons) throughout the optimization procedure. The authors further notice how momentum-based optimizers are problematic in continual learning tasks, and simply advice not to use them. Experimental results are presented on Split-CIFAR-10 against a number of baselines.

**Summary Of The Review:**

In my opinion the novelty of the work is limited for publication at ICLR. The authors reach remarkable performances on a custom task starting from pretrained embeddings, and I am lukewarm about how the model would perform in end-to-end continual learning (i.e. when representations are trained continually as well). The motivation of the work is purely biological, and there is limited discussion about why those ideas should work in practice.

---

> ### Author Response · Authors · 2022-11-15
> **First Reply to Reviewer**
>
> - **"The motivation that the authors highlight for the employment of SDMs is only biological plausibility. Although I acknowledge that as a good trait, the text does not convey the reason why, technically, the proposed model avoids catastrophic forgetting. "**
>
> Your interpretation is correct and we have made this intuition more clear in the Introduction section.
>
> It was not required reading for the review process but Appendix H also does a deep dive into the learning dynamics that make SDM successful, most relevant is Table 8 and Fig. 23 where we measure how specialized neurons are to specific tasks (avoid overwriting by later tasks) and how often each neuron is active (the capacity of the model to separately represent many different tasks). These two factors combined appear strongly correlated with continual learning ability.
>
> - **"The paper describes how both read and write operations are performed in a SDM. However, the parallelism between SDM and a one-hidden-layer MLP is drawn only in the case of read operations (Eq. 3 and Eq. 2). How does the writing mechanism intervene in the authors model? Is the gradient-based learning of the MLP parameters considered the writing algorithm? If so, gradient-based updates do not conform with the writing mechanism for SDMs (Eq. 1). Can we still consider the model a SDM in that case? And if so, why?"**
>
> This is a great question that we have added Appendix A.6 to explain and also refer to in the main text.
>
> The mechanism is gradient based as you suggest where during model training all inputs are considered write operations. After training, during inference, all inputs are considered queries that perform the SDM read operation. $\mathbf{p}_a$ corresponds to a CIFAR image and $\mathbf{p}_v$ is a one hot label of its encoding.
>
> The original SDM write operation directly updates the neuron value vector with the pattern value $\mathbf{x}_v = \mathbf{x}_v + \alpha \mathbf{p}_v$ where $\alpha$ weights the pattern by the amount each neuron was activated (we use lower case notation here to refer to a specific neuron).
>
> Meanwhile, the backpropgation write operation updates $\mathbf{x}_v$ with the error between the model output and the true class one-hot (using cross entropy loss).
>
> There are a few reasons why this difference is compatible with SDM:
>
> First, an optimal solution for $\mathbf{x}_v$ that will result in zero error is the one hot encoding used by the original SDM write operation.
>
> Second, the original SDM write is only appropriate when the neuron addresses are fixed and $\mathbf{x}_v$ is the only thing learnt. Otherwise, as neurons update their address $\mathbf{x}_a$, this changes the patterns they are activated by and what $\mathbf{p}_v$ they should store in $\mathbf{x}_v$. Using the backpropagation approach to continuously update $\mathbf{x}_v$ as a function of the patterns it is currently activated by is a viable solution.
>
> Finally, from a biological perspective, the error signal used by backpropagation is a closer approximation to how the cerebellar circuit that SDM maps to updates $\mathbf{x}_v$ (Bidirectional learning in upbound and downbound microzones of the cerebellum, De Zeeuw, 2021). While backpropagation through multiple layers of a deep network has been argued to be biologically implausible, this update to the output layer is directly connected to the error computation making it possible.
>
> In summary, SDM will explicitly write in $\mathbf{p}_v$ while the MLP with backprop will compute a delta between $\mathbf{p}_v$ and the network output but this approach is compatible with the same solution, works better when also learning neuron addresses, and is likely to be more biologically plausible.

---

> > ### Author Response · Authors · 2022-11-15
> > **Second Reply to Reviewer**
> >
> > - **"The main experiment on CIFAR10 presents encouraging results, where the proposed SDM trained incrementally learns better than other models that specifically intervene to reduce catastrophic forgetting. However, the setting is quite arbitrary, as the authors use pretrained imagenet embeddings and an SDM on top of that. This raises two concerns. i) given that imagenet embeddings already separate CIFAR10 classes (as the cifar10 class space is a subset of the imagenet class space), is this an interesting learning setting? ii) are all compared methods in Tab. 1 using the same protocol? In my opinion, the case where both the embeddings and the classifiers need to be learned is the most interesting scenario for studying continual learning problems. Can the proposed models be extended to arbitrary backbones (e.g. ResNets), where every pair of consecutive convolutional layers can be considered as a SDM?"**
> >
> > To address concerns with our pretrained ImageNet embeddings we ablated our training scenario in two ways: 1. we remove these embeddings and train directly on CIFAR/MNIST pixels (Appendix E.1) 2. we remove model pre-training (Appendix E.2). In both cases, while the final accuracy values change, the rank ordering of methods stays approximately the same such that our results and method generalize.
> >
> > All methods in Table 1 are compared using the same protocol. We have made this explicit with a footnote.
> >
> > We agree that the joint training of the embedding model and SDM model is interesting and note in the limitations section and more extensively in Appendix C.2 that this joint training results in dead neurons.
> >
> > *"Avoiding dead neurons currently requires training SDM on a static data manifold, whether it is image pixels or a fixed embedding. Initial experiments jointly training SDM modules either interleaved throughout a ConvMixer or placed at the end resulted in many dead neurons and failure to continually learn. We believe this is in large part due to the manifold continuing to change over time (Appendix C.2)."*
> >
> > In ongoing work we are exploring the ability to extend our results to deeper models that use Top-K throughout and dynamically learn $k$.
> >
> > - **"Table 1 would benefit a oracle where all classes are trained jointly, to put the results into perspective."**
> >
> > Thanks for this suggestion. Table 1 has an oracle for the 10,000 neuron model (bottom of the Table on the right, and as the horizontal dotted red line in Fig. 3). However, there is no oracle for the 1,000 neuron model, there should be and we have now added it.
> >
> > - **"In Table 2, different optimizers showcase very different performances on the task. Are all baselines in Tab. 1 comparable, meaning that they all use SGD without momentum as an optimizer?"**
> >
> > We tested both SGD and SGDM for all models including across a number of learning rates. We present the best performing models in Table 1 which for everything but our SDM combination with EWC/MAS is SGD (our SDM combination does slightly better with SGDM). These hyperparameters are available in our codebase scripts (e.g. https://github.com/anon8371/AnonPaper1/blob/main/exp_commands/cont_learn_on_SplitCIFAR10_embeddings.py) and we have added a footnote that makes this explicit.
> >
> > As an aside, the biggest performance difference between SGD and SGDM is for SDM (when not combined with other methods) which we theorize is because of its high degree of sparsity causing Stale Momentum to have the largest effect. (This theory is most thoroughly investigated in Appendix D).

---

> > > ### Comment · Reviewer_HvB9 · 2022-11-22
> > > **Final recommendation**
> > >
> > > I thank the authors for the detailed response to my review, which clarified to me many points. However, I am not fully convinced and I will keep my original borderline score.
> > > Specifically, my main concern is that the proposed model is illustrated as a SDM, but it is not exactly so. In this respect, reviewer 1XFe shared the same concern. It is true that, when equipped with a top-k activation function, the linear layer within an MLP resembles the read operation of an SDM. However, this is simply because both constitute of the same building block, namely linear projections, which are obiquitous in computer science.Following the same reasoning, almost every operation in deep learning (self-attention, squeeze and excitation, convolutions, recurrent networks) can be interpreted as SDMs with minor modifications.
> > > More importantly, the write operation seems somewhat different to me, and implicitly embedded into a gradient-based update. The authors, in their response, draw resemblance between SDM write operations and gradient-based updates, which I find unclear / unconvincing.
> > >
> > > Finally, the technical (not biological) argument about the reason why the proposed model is robust to forgettiing is that the top-k activation function yields sparse activation patterns, who in turn yield sparse gradient updates and therefore less forgetting. This finding is mostly known in the community and not particularily interesting.
> > >
> > > The paper has the merits of a clearly well conducted research, with extensive experiments and interpretation of results, which makes my score assess to borderline.

---

> > > > ### Author Response · Authors · 2022-11-22
> > > > **Reply to Reviewer**
> > > >
> > > > We thank the reviewer for their reply and additional comments.
> > > >
> > > > * **"Specifically, my main concern is that the proposed model is illustrated as a SDM, but it is not exactly so. In this respect, reviewer 1XFe shared the same concern."**
> > > >
> > > > While we cannot speak for reviewer 1XFe, they replied saying that our response cleared most of their concerns and did not re-raise the issue of the model being compatible with SDM.
> > > >
> > > > * **"The authors, in their response, draw resemblance between SDM write operations and gradient-based updates, which I find unclear / unconvincing."**
> > > >
> > > > Can the reviewer explain what in particular they found unclear/unconvincing? It is important that this part of our paper is both sound and understandable.
> > > >
> > > > * **"The top-k activation function yields sparse activation patterns, who in turn yield sparse gradient updates and therefore less forgetting."**
> > > >
> > > > One of our baselines is the Top-K activation function which catastrophically forgets. Our ablations in Table 2 show that the L2 normalization of weights and data along with having no bias terms are both crucial for Top-K to continuously learn. These modifications both came directly from their SDM inspiration along with the GABA switch that further increases performance.
> > > >
> > > > * **"It is true that, when equipped with a top-k activation function, the linear layer within an MLP resembles the read operation of an SDM. However, this is simply because both constitute of the same building block, namely linear projections, which are obiquitous in computer science."**
> > > >
> > > > Our above response shows that the SDM model modifications are more than just linear projections. They are: Top-K, L2 normalization of weights and data; no bias terms; the GABA switch. We believe this modifications, especially when combined are non-trivial and deviate from the norm in deep learning/computer science more broadly.
> > > >
> > > > * **"Following the same reasoning, almost every operation in deep learning (self-attention, squeeze and excitation, convolutions, recurrent networks) can be interpreted as SDMs with minor modifications."**
> > > >
> > > > The mapping to self-attention is highly non-trivial and requires utilizing the high dimensional properties of hypersphere intersections that approximate the exponential weighting of the softmax (Bricken & Pehlevan, 2021). The cerebellar circuit that is capable of implenting these operations also has a very unique architecture with the sheer number of granule cells (making up ~70\% of all neurons in the brain [2]) and three way convergence between granule cells, Purkinje cells, and Climbing fibers, being particularly unique. No claims of mappings to squeeze and excitation, convolutions, recurrent networks have been made and are not apparent to us.
> > > >
> > > > [1] "Attention approximates Sparse Distributed Memory" (https://proceedings.neurips.cc/paper/2021/hash/8171ac2c5544a5cb54ac0f38bf477af4-Abstract.html).
> > > > [2] “Updated Energy Budgets for Neural Computation in the Neocortex and Cerebellum” (https://pubmed.ncbi.nlm.nih.gov/22434069/).

---

> > > > > ### Author Response · Authors · 2022-12-08
> > > > > **Followup Reply**
> > > > >
> > > > > Dear reviewer,
> > > > >
> > > > > As the end of the discussion period is fast approaching, please let us know if you have any further comments or concerns to which we can respond. We'd be interested hearing if our comments and changes to the manuscript helped answer your questions!

---

### Official Review · Reviewer_krw5 · 2022-10-23

**Confidence:** 3
**Correctness:** 4
**Technical Novelty And Significance:** 3
**Empirical Novelty And Significance:** 4
**Recommendation:** 8

**Clarity, Quality, Novelty And Reproducibility:**

# Originality
The paper proposes multiple biologically plausible extensions to MLP that result in a strong continuous learning architecture. This is in contrast with most previous approaches that relied on artefacts to induce sparse networks and specialise them to different tasks.

The authors provide novel insights on how the proposed extensions are able to tackle the "dead neurons" problem. Moreover, the authors identify "state momentum" as a problem that has been missed by previous works.

The MLP formulation of SDM shades new light on its connection with the attention mechanism from Transformer architectures.


# Quality
All the architectural design decisions are discussed in detail, including extensive related work.
Experiments are well designed with promising results.
Limitations of the approach have been identified and discussed.

# Clarity
The paper bridges the gap between deep learning and computational neuroscience. This is not an easy task, but I think the authors did a fantastic job, making the neuroscience concepts accessible to the deep learning community.

# Reproducibility
The authors have made their code available, which together with the extensive details in the main text and appendixes make me confident the results are fully reproducible.



**Strength And Weaknesses:**

# Strengths
The paper aims to provide a biologically plausible extension of MLP that naturally results in continuous learning capabilities. This is an important and usually overlooked research area.

The connection between SDM, MLP and Transformer is very interesting and it is developed in further detail.

The numerical results are very promising. Although the results are not SOTA for CIFAR100, they are still competitive and outperform other baselines.

More importantly, the extensive but practical explanations behind their design decisions, the insightful ablation studies, and the suggested directions for future work, will likely have an impact in the community and will foster future research.

# Weaknesses
Simulation results are promising. However, it would be reassuring to see a common trend with other data modalities, like text. For example, the authors could use a pretrained modern Transformer architecture (e.g., DistilBERT) and evaluate continuous learning while fine tuning to different tasks.

The motivation of a single layer MLP is not very clear. It would be interesting to see whether the results will hold for a multilayer MLP.

The explanation of the three stages training regime in the main text is not very clear (probably too concise due to lack of space). The authors could clearly enumerate the three steps, explaining a ConvMixer is used for pretraining from the beginning, and making clear what SDM Module refers to their SDM-inspired MLP variant (maybe coining their model something like "SDM-MLP" could prevent the risk of a reader thinking that SDM and MLP are two different models).

The ablation study on training directly on image pixels would be more relevant if they were done by tuning the loss coefficient and the $\beta$ parameter for training directly on pixels.

**Summary Of The Paper:**

The paper proposes a number of modifications to the standard Multi-Layered Perceptron (MLP) to avoid catastrophic forgetting in continuous learning tasks. The proposed modifications are motivated as biologically plausible and include the use of sparse distributed memory, a Top-K activation function, no bias terms, and L2 normalization and non-negativity constraints on weights and data. These features are studied in isolation and combined with other approaches, like Elastic Weight Consolidation (EWC). In addition, the paper proposes two training techniques: i) an implementation inspired on the "GABA Switch" to avoid "dead neurons", showing this is equivalent to annealing number of active neurons (K in the Top-K activation function) when the non-negativity constraint is imposed; and ii) using an optimizer without momentum to avoid updating neurons that are not in the Top-K set and another potential exploding gradients issue. The authors consider a three stages during training: i) first, pretraining a ConvMixer on ImageNet; ii) pretraining the SDM using the pretrained embedding; iii) continuous learning of the pretrained SDM module with the pretrained image embedding. Experiments show the proposed modifications to the MLP, combined with EWC, provide SOTA results for CIFAR 10 in the challenging class incremental setting, and achieves second better performance for CIFAR 100 and Split MNIST.


**Summary Of The Review:**

This is a solid work, well written, full of insights, with promising results, and very relevant. I think it will foster future research on both SDM and continuous learning, and has the potential to help the representation learning community to be more aware of the potential of using neuroscience to improve standard deep learning architectures.

---

> ### Author Response · Authors · 2022-11-15
> **Reply to Reviewer**
>
> We thank the reviewer for their time, comments, and feedback. Replying to each comment in turn.
>
> - **"Simulation results are promising. However, it would be reassuring to see a common trend with other data modalities, like text. For example, the authors could use a pretrained modern Transformer architecture (e.g., DistilBERT) and evaluate continuous learning while fine tuning to different tasks."**
>
> Thank you for this suggestion. We would like to test our approach on a number of other datasets including RL environments where task boundaries don't exist and our approach may particularly shine. Text fine-tuning would be another excellent environment and with pre-trained models ready to use. In the space and time allotted, we only had time to test our approach on vision tasks; we are excited about future work in this direction.
>
> - **"The motivation of a single layer MLP is not very clear. It would be interesting to see whether the results will hold for a multilayer MLP."**
>
> We have made this motivation more clear -- it comes from the fact that Sparse Distributed Memory (SDM) can be directly related to a single layer MLP (Equations 2 and 3).
>
> We performed initial experiments with deeper models but the results were nuanced and require more investigation around the $k$ value for Top-K, which layers should use Top-K, and resolving dead neurons. As noted in the limitations section:
>
> *"Avoiding dead neurons currently requires training SDM on a static data manifold, whether it is image pixels or a fixed embedding. Initial experiments jointly training SDM modules either interleaved throughout a ConvMixer or placed at the end resulted in many dead neurons and failure to continually learn. We believe this is in large part due to the manifold continuing to change over time (Appendix C.2)."*
>
> In work that is ongoing we are learning optimal $k$ values across multiple layers that makes us optimistic the findings of this paper can generalize further.
>
> - "**The explanation of the three stages training regime in the main text is not very clear (probably too concise due to lack of space). The authors could clearly enumerate the three steps, explaining a ConvMixer is used for pretraining from the beginning, and making clear what SDM Module refers to their SDM-inspired MLP variant (maybe coining their model something like "SDM-MLP" could prevent the risk of a reader thinking that SDM and MLP are two different models).**"
>
> We have taken this great suggestion to rename our approach and have gone with the slightly shorter name "SDMLP" that is now used throughout the paper and figures. We have also re-written our explanation of the different training stages and re-designed Figure 2 to increase clarity.
>
> - "**The ablation study on training directly on image pixels would be more relevant if they were done by tuning the loss coefficient and the  parameter $\beta$ for training directly on pixels.**"
>
> We have hyper-parameter tuned the $\beta$ coefficient to make these results more relevant for both MNIST (Appendix F.3) and the new FashionMNIST dataset (Appendix F.4). Performance increased but still respects the rank ordering of model performances. Thank you for this suggestion.

---

> > ### Comment · Reviewer_krw5 · 2022-11-17
> > **Thank for the response**
> >
> > I thank the authors for their response. I have no further comments.

---

### Official Review · Reviewer_1XFe · 2022-10-24

**Confidence:** 4
**Correctness:** 3
**Technical Novelty And Significance:** 3
**Empirical Novelty And Significance:** 2
**Recommendation:** 6

**Clarity, Quality, Novelty And Reproducibility:**

The paper is not self-contained, which impairs its clarity significantly. The quality is good given the amount of investigation and study presented in the main text and appendix. The finding is interesting and new. However, as mentioned in the Weakness, the final method looks so different from the original SDM. Reproducibility is challenging if someone wants to apply the method to a new dataset. It will require a lot of tuning.

**Strength And Weaknesses:**

### Strength
- The paper gives a new insight: SDM can support continual learning, mainly based on its sparsity nature.
- The paper has done a non-trivial task to make SDM work for actual data, which involves many tips and tricks in training SDM.
### Weakness
- The presentation is hard to follow. The main text lacks details to understand the method. The paper is not self-contained and often requires references to Appendix, which is only optional for the reviewing process.
- Too many modifications to SDM make the final method no more SDM, which depreciates the paper's central message. Also, the amount of tuning hyperparameters and calibrating training procedures is enormous, making it hard to apply to a different CL data or task.
- The experiment is limited. The proposed method (combined with EWC) is only better than FlyModel in CIFAR10, and underperforms in CIFAR100 and MNIST

### Detailed comments and questions:
- Eq. 3, how do you write to $X_v$? If you use SDM's write, what are the $P_a$ and $P_v$ in the case of CIFAR10 data?
- The paper claims to fix the issue of random $X_a$ with top-K activation. However, Eq. 4 only presents the activation function, which affects the output $a$. How does it help $X_a$ model real-world data?
- Additional modification: (v) How do you use backpropagation in SDM? (to update $X_v$, $X_a$?)
- Please consider adding an algorithm to the main text to clarify how your proposed components work together.
- Please consider more datasets to validate the performance of your method: CelebA, FahsionMNIST, .... Given the current result; it is hard to say your method is more effective than FlyModel
- Why don't you make stronger baselines by combining methods just as you did with SDM? For example, FlyModel+EWC or use more baselines such as EWC, SI, HAT
- Check the section numbering. It should start from 1. Introduction ...

**Summary Of The Paper:**

This paper interprets Spare Distributed Memory (SDM) as a continual learner. Analogous to a one-hidden-layer MLP, the SDM is modified to support continual learning. There are several training tricks, such as the choice of optimizer and the training procedure. The most notable change is to replace the binarization activation in SDM with the top-k function, where k is annealed over training time. The method shows improved performance in the class-incremental setting with  CIFAR10 dataset.


**Summary Of The Review:**

The paper provides an interesting insight into SDM as a continual learner. The method, however, is over-complicated with many components and tuning efforts. The experimental result is weak.

---

> ### Author Response · Authors · 2022-11-15
> **First Reply to Reviewer**
>
> We thank the reviewer for their time, comments, and feedback. Replying to each comment in turn:
>
> - **"The presentation is hard to follow. The main text lacks details to understand the method. The paper is not self-contained and often requires references to Appendix, which is only optional for the reviewing process."**
>
> We appreciate this feedback, and have made additional efforts to clarify our method. We think clarity in scientific writing is of the utmost importance. This paper required lots of work that we couldn't include in the main text due to space constraints. In these instances, we chose to be rigorous and include clarifications in the Appendix, though we understand this makes the paper harder to read.
>
> - **"Too many modifications to SDM make the final method no more SDM, which depreciates the paper's central message."**
>
> We disagree with the characterization of the final method no longer being an instance of SDM. Most of our changes move a vanilla MLP closer to SDM (as opposed to the reverse). We do agree that we insufficiently explained how the SDM write operation is related to training with backpropagation and in response to one of your later questions have added a reference in the main text and next Appendix (A.6) to address this.
>
> Fundamentally, we take the continuous version of SDM (Bricken & Pehlevan, 2021), and implement the idea from Keeler (1988) of using Top-K to allow for neurons to learn their addresses, thus being capable of tiling data manifolds. This results in a version of SDM that is identical to an MLP that uses a Top-K activation function, has $L^2$ normalization of its weights and data, and no bias terms.
>
> In order for the model to be trained via backpropagation, we allow for neurons to use a continuous (ReLU or Exponential) activation function rather than binary (Heaviside), however we show in Appendix A.5 that this modification gives similar weightings to each pattern as the original binary SDM (determined by  the value of the read and write hypersphere intersection). When using exponential these weightings are the same, with ReLU they are closely approximated, we only use ReLU throughout the paper as exponential activation functions are uncommon.
>
> Training via backpropagation does use a slightly different update to $X_v$ than the SDM write operation. This is something we have made more clear by adding Appendix A.6 and we address in response to your later question (see three bullet points down).
>
> - "**Also, the amount of tuning hyperparameters and calibrating training procedures is enormous, making it hard to apply to a different CL data or task.**"
>
> We agree that carefully tuning hyperparameters makes methods  harder to apply to novel settings. Regardless, this calibration is common practice, which makes doing so necessary to construct proper baselines with previous work. We therefore apply careful tuning to *all* methods, even making a (minor) novel finding that adjusting the $\beta$ coefficient for Elastic Weight Consolidation (EWC) and Synaptic Intelligence (SI) can significantly increase their performance (Appendix G.1).
>
> Beyond our primary setting where we use latent image embeddings and ImageNet pretraining, we ablate both of these assumptions resulting in evaluations that are more readily comparable to other papers (Appendices E.1 and E.2). We also tested our method on Split MNIST and Fashion MNIST (Appendices F.3 and F.4).
>
> In addition, our method requires different hyperparameters but not a larger quantity than other baselines. It is only necessary to choose the terminal $k$ value used by Top-K and ensure it reaches its final value before the end of the first continual learning task. This is equivalent to tuning the regularization coefficient for the weight regularization methods, in addition to the EWC and SI $\beta$ coefficient. The FlyModel also requires tuning a $k$ value.
>
> Finally, we acknowledge that the continual learning field is highly fragmented in its use of different datasets and training procedures. For example, it is still somewhat uncommon to use the more challenging and realistic class incremental setting that we believe should be the standard. We hope that deciding to use this setting in our work helps to make it more of the norm.

---

> > ### Author Response · Authors · 2022-11-15
> > **Second Reply to Reviewer**
> >
> > - "**The experiment is limited. The proposed method (combined with EWC) is only better than FlyModel in CIFAR10, and underperforms in CIFAR100 and MNIST**"
> >
> > The objective of our paper is not solely focused on creating a new state of the art model for every task. Instead, we reveal that the FlyModel is an instance of SDM, recast SDM as a new Deep Learning model, connect this model to specific neurobiological mechanisms, and show that SDM derived models are collectively state of the art across domains. In this context, the FlyModel doeing better in MNIST and CIFAR100 supports the title of our paper that SDM is a strong continual learner.
> >
> > We have re-written the Related Work section to do a better job of highlighting the unique differences between our paper and the FlyModel. The FlyModel implements the Hyperplane design of SDM (Jaeckel, 1989), uses fixed weights, and does not attempt to cast their model in a deep learning framework.
> >
> > Our work instead implements Keeler's (1988) idea to enable neurons to learn their addresses, allowing for SDM to learn data manifolds. This approach does not appear to have continual learning advantages compared to the FlyModel, however, it does have other benefits that are useful for other tasks such as preserving the similarity between data points and creating more interpretable neurons (e.g. Figure 26 of Appendix H). Additionally, our approach beats the alternative deep learning baselines tested and creates new links with neurobiology. We believe the GABA switch in particularly provides a new example for how neuroscience can be used to inspire solutions in AI.
> >
> > - "**Eq. 3, how do you write to $X_v$? If you use SDM's write, what are the $P_a$ and $P_v$ in the case of CIFAR10 data?**"
> >
> > $X_v$ is updated with backpropagation where $\mathbf{p}_a$ is the input (a CIFAR10 image) and $\mathbf{p}_v$ is the one hot class label. In our single hidden layer model, backpropagation will update the $\mathbf{x}_v$ such that they more closely output the one-hot class label of the input. Training the model is analogous to SDM write operations when $X_v$ is updated while inference after the model is trained is analogous to SDM read operations where the input is now a query and $X_v$ is read from.
> >
> > The original SDM write operation directly updates the neuron value vector with the pattern value $\mathbf{x}_v = \mathbf{x}_v + \alpha \mathbf{p}_v$ where $\alpha$ weights the pattern by the amount each neuron was activated (we use lower case notation here to refer to a specific neuron).
> >
> > Meanwhile, the backpropgation write operation updates $\mathbf{x}_v$ with the error between the model output and the true class one-hot (using cross entropy loss).
> >
> > There are a few reasons why this difference is compatible with SDM:
> >
> > First, an optimal solution for $\mathbf{x}_v$ that will result in zero error is the one hot encoding used by the original SDM write operation.
> >
> > Second, the original SDM write is only appropriate when the neuron addresses are fixed and $\mathbf{x}_v$ is the only thing learnt. Otherwise, as neurons update their address $\mathbf{x}_a$, this changes the patterns they are activated by and what $\mathbf{p}_v$ they should store in $\mathbf{x}_v$. Using the backpropagation approach to continuously update $\mathbf{x}_v$ as a function of the patterns it is currently activated by is a viable solution.
> >
> > Finally, from a biological perspective, the error signal used by backpropagation is a closer approximation to how the cerebellar circuit that SDM maps to updates $\mathbf{x}_v$ (Bidirectional learning in upbound and downbound microzones of the cerebellum, De Zeeuw, 2021). While backpropagation through multiple layers of a deep network has been argued to be biologically implausible, this update to the output layer is directly connected to the error computation making it possible.
> >
> > In summary, SDM will explicitly write in $\mathbf{p}_v$ while the MLP with backprop will compute a delta between $\mathbf{p}_v$ and the network output but this approach is compatible with the same solution, works better when also learning neuron addresses, and is likely to be more biologically plausible.
> >
> > We have update our paper with a better explanation of how $X_v$ is updated by backpropagation and the relation to SDM in Appendix A.6 that we refer to in the main text.

---

> > > ### Author Response · Authors · 2022-11-15
> > > **Third Reply to Reviewer**
> > >
> > >
> > > - **"The paper claims to fix the issue of random $X_a$ with top-K activation. However, Eq. 4 only presents the activation function, which affects the output $a$. How does it help  model real-world data?"**
> > >
> > > The Top-K activation function is not necessary for the MLP model to learn $X_a$, this is possible with alternative activation functions and by simply allowing backprop to update $X_a$. The use of Top-K is necessary for this MLP model to be theoretically compatible with SDM.
> > >
> > > Replacing SDM's original fixed activation threshold with a dynamic  threshold calculated by Top-K enables SDM to learn $X_a$. This idea was first introduced in Keeler (1988) and is explained in Appendix A.2. Appendix A.3 provides historical context on why SDM originally assumed fixed neuron addresses, and why removing this constraint actually increases biological plausibility.
> > >
> > > Therefore, without the Top-K activation function, SDM would be unable to both maintain its theoretical underpinnings and also learn neuron addresses. We have modified our explanation of SDM and the Top-K activation function to better account for this source of confusion.
> > >
> > > - **"Additional modification: (v) How do you use backpropagation in SDM? (to update $X_a$, $X_v$?)"**
> > >
> > > Yes we use backprop to update $X_a$ and $X_v$?. Hopefully our answer to the earlier question about how $X_v$ is written is informative here too. We have made this explanation explicit and added Appendix A.6.
> > >
> > > - **"Please consider adding an algorithm to the main text to clarify how your proposed components work together."**
> > >
> > > Thanks for this suggestion. We have added Appendix A.1 with a reference in the main text. This algorithm box outlines step by step how the inputs are modified by the network and the contribution of each unique component from our architecture.
> > >
> > > - **"Please consider more datasets to validate the performance of your method: CelebA, FashionMNIST, .... Given the current result; it is hard to say your method is more effective than FlyModel."**
> > >
> > > Thank you for this suggestion. Have performed analysis of the FashionMNIST dataset to further compare our model with the FlyModel in Appendix F.4 In addition, we have re-writen portions of our Related Work section to make the differences with FlyModel more clear such that the contributions of each paper beyond their performance on a particular dataset are clearer.
> > >
> > > - **"Why don't you make stronger baselines by combining methods just as you did with SDM? For example, FlyModel+EWC or use more baselines such as EWC, SI, HAT"**
> > >
> > > Because the FlyModel does not use Deep Learning, it is incompatible with any of the other baselines. Concretely, the FlyModel does not use backprogagation for training which both EWC and SI use to infer the importance of each weight (and how much to protect them from being updated by later tasks). Moreover, the FlyModel uses fixed weights for its neuron addresses $X_a$ which are the most important for EWC and SI to protect from updates. This is because $X_a$  determines which neurons are active and what $X_v$ are active/updated downstream.
> > >
> > > This is an important difference between the FlyModel and ours that we've now better emphasized in the Related Work section.
> > >
> > > - **"Check the section numbering. It should start from 1. Introduction ...""**
> > >
> > > Thank you for spotting this formatting error that we have corrected.

---

> > > > ### Comment · Reviewer_1XFe · 2022-11-18
> > > > **One additional question**
> > > >
> > > > Thank you for providing a detailed response, which clears most of my concerns. From your explanation, I understand that the MLP "becomes" SDM thanks to:
> > > > - Top-K activation: SDM's KNN read
> > > > - Backprob update: SDM's write
> > > >
> > > > Your approach seems similar to this line of work [1,2], which should be mentioned in the paper. As I see, these works also perform KNN read (in form of attention) and backprob writing.  Could you compare your method with these papers and show the difference that makes your approach novel?
> > > >
> > > > [1] Le, Hung, Truyen Tran, and Svetha Venkatesh. "Neural Stored-program Memory." In International Conference on Learning Representations. 2019.
> > > >
> > > > [2] Le, Hung, and Svetha Venkatesh. "Neurocoder: General-Purpose Computation Using Stored Neural Programs"
> > > > International Conference on Machine Learning, 2022

---

> > > > > ### Author Response · Authors · 2022-11-22
> > > > > **Reply to Reviewer**
> > > > >
> > > > > We thank the reviewer for their reply and additional suggestions.
> > > > >
> > > > > * "**Your approach seems similar to this line of work [1,2], which should be mentioned in the paper. As I see, these works also perform KNN read (in form of attention) and backprob writing. Could you compare your method with these papers and show the difference that makes your approach novel?**"
> > > > >
> > > > > You are correct that the referenced papers (refered to as "Neurocoder") use an Attention based read and backprop for writing. However, there are a few significant differences.
> > > > >
> > > > > The biggest difference is that Neurocoder uses a highly complex meta network to produce weights for the feed-forward network. In other words, plugging different programs into the feed-forward network. Our SDM model only has a single primary network that learns sub-modules but always uses the same weights. For our SDM to be like the Neurocoder, we would replace the keys, values, and attention operation of the Neurocoder meta-network with our SDM single hidden layer MLP and then use the output from it as the weights for the primary network. However, our SDM model is not operating at this meta level.
> > > > >
> > > > > Additionally, the modularity/compositionality of each Neurocoder program (used to generate the network weights) allows for interaction effects between programs. This is cited as being a primary difference between Neurocoder and other approaches like Mixture of Experts. Our SDM model is much more similar to the Mixture of Experts, where each neuron is an "expert" and acts independently, potentially encoding redundant information.
> > > > >
> > > > > One minor difference that at first glance appears to be a similarity is that Neurocoder is not "organic" in the same way that SDM is. The paper claims that Neurocoder does not need to be given any labels/boundaries between tasks, however, it is only trained in the task and domain incremental settings rather than class incremental. As a result, the output heads are gated by the current task and so the task boundary information is provided implicitly.
> > > > >
> > > > > An additional minor difference is that their Attention update is not truly sparse in the same way that Top-K is (the softmax will produce many small but non-zero activations).
> > > > >
> > > > > Acknowledging these differences, there may be an interesting synergy between Neurocoder and our SDM model whereby our SDM model replaces the feed-forward layers inside the Neurocoder that are used to produce its query and residual weights (all of the fixed network portions of the meta network). It is noted in the papers that these components are problematic for continual learning.
> > > > >
> > > > > We will be sure to update our paper with citations to these papers as examples of alternative ways to perform continual learning with sub-networks and compare our approach with [1,2].
> > > > >
> > > > > Thank you again for all of your helpful comments and discussion that improved our work.

---

> > > > > > ### Author Response · Authors · 2022-12-08
> > > > > > **Followup Response**
> > > > > >
> > > > > > Dear reviewer,
> > > > > >
> > > > > > As the end of the discussion period is fast approaching, please let us know if you have any further comments or concerns to which we can respond. We'd be interested hearing if our comments and changes to the manuscript helped answer your questions!

---

> > > > > > > ### Comment · Reviewer_1XFe · 2022-12-08
> > > > > > > **Score updated**
> > > > > > >
> > > > > > > Given the change in your revision, I have increased my score to 6.
> > > > > > >
> > > > > > > I am still not entirely convinced about the message "SDM is a continual learner" because the continual learner in this paper is too far from SDM. So it is not an apparent acceptance to me.

---

### Official Review · Reviewer_2ptq · 2022-10-26

**Confidence:** 4
**Correctness:** 3
**Technical Novelty And Significance:** 3
**Empirical Novelty And Significance:** 3
**Recommendation:** 8

**Clarity, Quality, Novelty And Reproducibility:**

The paper is well written, and the methods are clearly explained. The overall quality of the paper is high.

The paper seems novel to my knowledge, esp. with the specific set of tricks the authors use to train the SDM+MLP architecture. But relation to (Shen et al. 2021) needs to be discussed in more detail.

The paper seems to have sufficient experimental details, esp. for the training, and hence looks to be reproducible.

**Strength And Weaknesses:**

## Strengths

- The authors clearly demonstrate that SDM in the specific form they use is suited for continual learning, with good ablation experiments
- Detailed experiments on the one benchmark that is chosen.
- The authors contribute a few tricks for training SDM-based architectures that are novel.
- Methods are clearly explained.

## Weaknesses

- Evaluation done only on one benchmark.
- Not clear if biological plausibility is a goal, and if so, how it is achieved.
- Differences from (Shen et al. 2021) not clear.

**Summary Of The Paper:**

The authors construct an architecture based on sparse-distributed memory (SDM) and multi-layer perceptrons and show that this architecture is naturally robust to catastrophic forgetting. This architecture uses sparse activations of neurons (top-K) with a couple of tricks to improve trainability. One one continual learning task (CIFAR-10), the authors show that this produces very good performance, esp. when combined with EWC.

**Summary Of The Review:**

Overall, it is a well-written high-quality paper with very interesting and relevant results for continual learning. The paper's biggest weakness is having results (albeit detailed) on only one benchmark. The paper needs comparisons on a more diverse set of benchmarks, and is the primary reason for my rating.

---

> ### Author Response · Authors · 2022-11-15
> **Reply to Reviewer**
>
> We thank the reviewer for their time, comments, and feedback. Replying to each comment in turn.
>
> - **"Evaluation done only on one benchmark."**
>
> We appreciate your concern and agree that ensuring performance across various benchmarks is a core priority for all deep learning research.
>
> Accordingly, we additionally tested on CIFAR-100 and MNIST, as well as adding the FashionMNIST dataset in response to this review (Appendices F.2, F.3, and F.4). We also ablate our training paradigm where we do not pretrain the models on ImageNet and also train them directly on image pixels rather than the learned embeddings (Appendices E.1 and E.2). We have made it more clear in the main text that these results exist and better summarize them.
>
> A dataset that does not have clear task boundaries would be particularly well suited to our "organic" learning capabilities but would require using different benchmark models that don't require this information (EWC, SI, MAS, NISPA could not be tested without adding an additional module to infer task boundaries).
>
> - **"Not clear if biological plausibility is a goal, and if so, how it is achieved."**
>
> Biological plausibility is certainly a sub-goal of this work that we believe makes it compelling. We have made this aspect of the work -- providing a bridge between Deep Learning practices and neurobiology -- clearer.
>
> Sparse Distributed Memory (SDM) has a compelling mapping to a particular circuit in the cerebellum (summarized in Appendix A.1). We support every modification made to the MLP with this underlying biological mapping (Appendix A.4). We also go into the neuroscience behind the GABA Switch trick used to avoid dead neurons (Appendix B.2). And how well the Top-K activation function can be approximated by inhibitory interneurons (Appendix C.1).
>
> - **"Differences from (Shen et al. 2021) not clear."**
>
> We agree and have re-written our discussion of Shen et al 2021 to better distinguish the contributions of each work (see Related Work section).
>
> Fundamentally, Shen et al. take an original formulation of SDM and apply it out of the box to continual learning. Most notably, they do not reconcile differences with MLPs nor do they resolve SDM's inability to update neuron weights and "tile" a data manifold. However, the work supports the title of our paper that SDM is a continual learner and shows the utility of random projections. The use of fixed weights also reduces parameter count, making the Shen et al. model easier to train.
>
> Our work both extends the theory of SDM by allowing it to  successfully learn data manifolds, and reconciles the differences between SDM and MLPs such that SDM can be trained in the Deep Learning framework (this includes having no fixed neuron weights and using backpropagation). Both of these contributions are novel and may inspire future work beyond the scope of continual learning. For example, learning the data manifold preserves similarity in the data and leads to more specialized, interpretable neurons (e.g. Figure 25 of Appendix H).

---

> > ### Comment · Reviewer_2ptq · 2022-11-16
> > **Discussing SDM -> Cerebellum mapping**
> >
> > Thank you for your response. The addition of additional benchmarks and comparison with (Shen et al. 2021) is definitely enough for me to increase my rating to an accept.
> >
> > I would suggest adding some of the new benchmarks to the main text instead of the appendix and add a discussion/analysis of why the FlyModel outperforms your model in some tasks if possible in the final version.
> >
> > But, I'm not completely convinced by the biological plausibility argument. I am/was aware of the mapping between the SDM and cerebellum, but to my knowledge this mapping is quite outdated and not meaningful since the overall consensus in neuroscience is that memory is not one of the primary functions of the cerebellum (do correct me if I'm mistaken about this though). Mapping to the Dentate Gyrus -> CA3 <-> Entorhinal Cortex loop is probably more meaningful. Moreover, using feed forward networks makes the entire setup quite implausible biologically.
> >
> > While this is not going to affect my rating significantly, I would strongly suggest making these caveats clearer so that the reader can focus on the algorithmic innovations in the paper, and not on a somewhat questionable mapping to biology.

---

> > > ### Author Response · Authors · 2022-11-17
> > > **Reply to Discussing SDM -> Cerebellum Mapping**
> > >
> > > We thank the reviewer very much for their additional suggestions and for raising their score.
> > >
> > > * **"I would suggest adding some of the new benchmarks to the main text instead of the appendix and add a discussion/analysis of why the FlyModel outperforms your model in some tasks if possible in the final version."**
> > >
> > > We will add more of these results and further discussion of why the FlyModel performs better in these instances to the main text, especially if our paper is accepted and this gives us an additional page of main text.
> > >
> > > * **"But, I'm not completely convinced by the biological plausibility argument. I am/was aware of the mapping between the SDM and cerebellum, but to my knowledge this mapping is quite outdated and not meaningful since the overall consensus in neuroscience is that memory is not one of the primary functions of the cerebellum (do correct me if I'm mistaken about this though)."**
> > >
> > > It is correct that the original mapping between SDM and the cerebellum is old, having been made 34 years ago (Kanerva, 1988). However, this mapping closely agrees with the Albus and Marr interpretations, which are still considered plausible theories of cerebellar function [1]. The Jaeckell Hyperplane variant of SDM in particular introduces sparse neuron weights that are more compatible with their mapping to granule cells [2]. However, in Appendix A.2 on SDM's biological plausibility we do acknowledge some lingering questions for the mapping to be complete.
> > >
> > > As for the function of the cerebellum, and its relation to memory, this depends upon the definition of memory. If one agrees that associative learning (both auto and hetero associative) is a form of memory then the cerebellum is certainly involved [3]. More recent work has also found that it is involved in higher order cognitive functioning, to take one excerpt from the following review [4]:
> > >
> > > > A review of 275 positron emission tomography (PET) and functional magnetic resonance imaging (fMRI) studies revealed that cerebellar activation was observed during a broad range of tasks, including orienting attention, olfaction, spoken and written language, verbal working memory, problem solving, spatial memory, episodic memory, skill learning, and associative learning (Cabeza & Nyberg, 2000). A broad range of neuropsychological deficits has also been documented following localized cerebellar pathology, with deficits across tasks of attention, working memory, language and naming, counting, visuospatial processing, planning, and abstract reasoning reported (Kalashnikova, Zveva, Pugacheva, & Korsakova, 2005).
> > >
> > > More recently, neuron tracers finding closed loop circuits between the cerebellum and almost every other brain region [5]. In Drosophila, its cerebellum-like structure -- the Mushroom Body -- is also highly plastic and the center of associative learning [6].
> > >
> > > A fact that supports this significance of the cerebellum is that of an estimated ~90 billion neurons in our brain, ~70% are Granule cells in the cerebellum. While these Granule cells are small, their numerosity incurs a high metabolic cost that must be worthwhile [7].
> > >
> > > References:
> > >
> > > [1] "50 Years Since the Marr, Ito, and Albus Models of the Cerebellum" (https://pubmed.ncbi.nlm.nih.gov/32599123/)
> > >
> > > [2] "A class of designs for a sparse distributed memory" (https://ntrs.nasa.gov/api/citations/19920002426/downloads/19920002426.pdf)
> > >
> > > [3] "Bidirectional learning in upbound and downbound microzones of the cerebellum" (https://www.nature.com/articles/s41583-020-00392-x)
> > >
> > > [4] "The cerebellum and neuropsychological functioning: A critical review" (https://pubmed.ncbi.nlm.nih.gov/22047489/)
> > >
> > > [5] "Homologous organization of cerebellar pathways to sensory, motor, and associative forebrain" (https://pubmed.ncbi.nlm.nih.gov/34551311/)
> > >
> > > [6] "The Drosophila Mushroom Body: From Architecture to Algorithm in a Learning Circuit" (https://www.annualreviews.org/doi/abs/10.1146/annurev-neuro-080317-0621333)
> > >
> > > [7] "Updated Energy Budgets for Neural Computation in the Neocortex and Cerebellum" (https://pubmed.ncbi.nlm.nih.gov/22434069/).
> > >
> > > * **"Mapping to the Dentate Gyrus -> CA3 <-> Entorhinal Cortex loop is probably more meaningful."**
> > >
> > > The two pathways Entorhinal Cortex -> CA3 and Entorhinal Cortex -> Dentate Gyrus -> CA3 is certainly an interesting connection that could perform auto-associative learning (less likely hetero-associative because unlike in the cerebellum where separate inputs can enter through the granule cells and the climbing fibers, here the same input comes from the Entorhinal Cortex.) We are certainly interested in mapping SDM to other neural circuits outside of the cerebellum.

---

> > > > ### Author Response · Authors · 2022-11-17
> > > > **Reply #2 to Discussing SDM -> Cerebellum Mapping**
> > > >
> > > > * **"Moreover, using feed forward networks makes the entire setup quite implausible biologically."**
> > > >
> > > > We are curious which operations the reviewer sees as being the most problematic? We acknowledge that using backpropagation to train $X_a$ is implausible but point to Hebbian learning alternatives that also learn to tile the data manifold (e.g. "Manifold-tiling Localized Receptive Fields are Optimal in Similarity-preserving Neural Networks" (https://proceedings.neurips.cc/paper/2018/file/ee14c41e92ec5c97b54cf9b74e25bd99-Paper.pdf)). We also acknowledge in Appendix A.2 that:
> > > >
> > > > > Within the scope of our model, the weakest biological plausibility is how well Top-K actually approximates the Golgi inhibitory interneuron (Appendix C.1) and how L2 normalization is implemented (Appendix A.5).
> > > >
> > > > * **"While this is not going to affect my rating significantly, I would strongly suggest making these caveats clearer so that the reader can focus on the algorithmic innovations in the paper, and not on a somewhat questionable mapping to biology."**
> > > >
> > > > We will update our limitations section to include the incomplete biological mapping in the main text.
> > > >
> > > > Thank you again for all of your helpful comments and discussion that improve our work.

---

> > > > > ### Author Response · Authors · 2022-12-08
> > > > > **Reply Followup**
> > > > >
> > > > > Dear reviewer,
> > > > >
> > > > > As the end of the discussion period is fast approaching, please let us know if you have any further comments or concerns to which we can respond. We'd be interested hearing if our comments and changes to the manuscript helped answer your questions!

---

> > > > > ### Comment · Reviewer_2ptq · 2022-12-12
> > > > > **Interesting information about cerebellum**
> > > > >
> > > > > Thank you for your detailed response and the interesting literature on cerebellum data. Looking at it, I agree that it can be implicated in associative memory and is activated during higher cognitive function (which does not necessarily mean it contributes its associative memory specifically for continual learning). It is still a big jump from there to say that your circuit is biologically plausible because even if each individual component is biologically plausible, it does not mean the overall circuit is biologically plausible for achieving continual learning.
> > > > >
> > > > > As I mentioned above, the presence of recurrent connections is another aspect that is not accounted for. Does adding recurrence break your architecture? Is there evidence that circuits responsible for continual learning do not have any recurrent connections?
> > > > >
> > > > > I am not so concerned about the backpropagation part since even if it is biologically implausible (which I think it is), one could just argue about the biological plausibility of the resulting circuit. That being said, I haven't seen strong evidence that Hebbian learning alternatives exhibit performance even close to comparable to BP for end-to-end training. While biological plasticity exhibits elements of Hebbian learning in recordings, it is far from clear if it can be the sole viable mechanism for learning complex cognitive tasks.
> > > > >
> > > > > I personally think it's borderline misleading to cherry pick specific data to make individual components of a model biologically plausible while ignoring the plausibility of the overall circuit, or doing more rigorous system level comparisons or any direct experimental comparisons for that matter. Nevertheless, I understand there is no consensus on what biologically plausible means in the field, and this is a very common approach. I appreciate the authors adding clarifications to the text in this regard.

---

> > > > > > ### Author Response · Authors · 2022-12-13
> > > > > > **Reply to reviewer**
> > > > > >
> > > > > > We thank the reviewer for their reply, additional comments, and the stimulating discussion!
> > > > > >
> > > > > > * **"Looking at it, I agree that it can be implicated in associative memory and is activated during higher cognitive function (which does not necessarily mean it contributes its associative memory specifically for continual learning)."**
> > > > > >
> > > > > > To the extent that the cerebellum does associative learning of long term information (e.g. how to ride a bike) and does not catastrophically forget, it seems like the null hypothesis should be that the cerebellum is implicated in continual learning? Given that biological organisms are so adept at continual learning and must be in order to adapt across different tasks, it would be surprising if any region of the brain that stored long term memories would catastrophically forget? We are curious if the reviewer would disagree but either way we will make our reasoning presented here more explicit in the paper.
> > > > > >
> > > > > > * **"It is still a big jump from there to say that your circuit is biologically plausible because even if each individual component is biologically plausible, it does not mean the overall circuit is biologically plausible for achieving continual learning."**
> > > > > >
> > > > > > We agree that there are parts of the cerebellar circuit that we are missing. Our current paper states in Appendix A.2:
> > > > > >
> > > > > > > While our improved version of SDM assigns functionality to five different cell types in the cerebellar circuit, there is more of the circuitry to be mapped such as the Basket cells and Deep Cerebellar
> > > > > > Nuclei (Sezener et al., 2021).
> > > > > >
> > > > > > We will ensure that these limitations are mentioned in the main text more clearly.
> > > > > >
> > > > > > * **"As I mentioned above, the presence of recurrent connections is another aspect that is not accounted for."**
> > > > > >
> > > > > > Our general impression of the literature is that the cerebellum is considered feedforward and we are unaware of any recurrence aside from the long range connections between cerebellar and neo-cortical regions observed in [1].
> > > > > >
> > > > > > To pre-empt a possible concern that SDM historically considered multiple updates to converge to a target pattern from a noisy query (recurrence), the closely related modern Hopfield Networks have shown that a great deal of convergence is possible with just a single update step [2].
> > > > > >
> > > > > > References:
> > > > > > [1] "Homologous organization of cerebellar pathways to sensory, motor, and associative forebrain" (https://pubmed.ncbi.nlm.nih.gov/34551311/)
> > > > > >
> > > > > > [2] "Hopfield Networks is All You Need" (https://arxiv.org/abs/2008.02217)
> > > > > >
> > > > > > * **"Does adding recurrence break your architecture?"**
> > > > > >
> > > > > > Our prior assumption is that the model should be robust to recurrence with the combination of L2 normalization + TopK + lack of bias term, allowing for the same sub-network formation to protect against catastrophic forgetting. But of course this should and would be interesting to empirically validate in future work.
> > > > > >
> > > > > > * **"Is there evidence that circuits responsible for continual learning do not have any recurrent connections?"**
> > > > > >
> > > > > > Given our answer to the first bullet point where we expect every long term memory system in the brain to be capable to some degree of continual learning, it seems likely that at least one of these circuits uses recurrence. However, we cannot say for certain.
> > > > > >
> > > > > > * **"I haven't seen strong evidence that Hebbian learning alternatives exhibit performance even close to comparable to BP for end-to-end training."**
> > > > > >
> > > > > > We agree that backprop especially for deeper models trained end to end remains state of the art. However, the following works may be of interest where alternative learning rules are still competitive and seeing new improvements [1, 2, 3, 4].
> > > > > >
> > > > > > [1] "Can a Fruit Fly Learn Word Embeddings?"(https://arxiv.org/abs/2101.06887)
> > > > > >
> > > > > > [2] "Algorithmic insights on continual learning from fruit flies" (https://arxiv.org/abs/2107.07617)
> > > > > >
> > > > > > [3] "Hybrid Predictive Coding: Inferring, Fast and Slow" (https://arxiv.org/pdf/2204.02169.pdf)
> > > > > >
> > > > > > [4] "Preventing Deterioration of Classification Accuracy in Predictive Coding Networks" (https://arxiv.org/abs/2208.07114)
> > > > > >
> > > > > > * **"While biological plasticity exhibits elements of Hebbian learning in recordings, it is far from clear if it can be the sole viable mechanism for learning complex cognitive tasks."**
> > > > > >
> > > > > > Also agreed! We will add a comment summarizing the above points on how more is to be elucidated on biological learning rules both in neuroscience and deep learning.

---

> > > > > > > ### Author Response · Authors · 2022-12-13
> > > > > > > **Reply continued**
> > > > > > >
> > > > > > > * **"I personally think it's borderline misleading to cherry pick specific data to make individual components of a model biologically plausible while ignoring the plausibility of the overall circuit, or doing more rigorous system level comparisons or any direct experimental comparisons for that matter. Nevertheless, I understand there is no consensus on what biologically plausible means in the field, and this is a very common approach. I appreciate the authors adding clarifications to the text in this regard."**
> > > > > > >
> > > > > > > We appreciate this concern and think the bar for true "biological plausibility" should indeed be very high. We also believe our work presents a closer mapping to biology than the status quo in this domain and hope that through our both our proposed edits and acknowledgements of limitations we can put forward this biological mapping while not overstepping.

---

### Comment · Area_Chair_rHGT · 2022-11-18
**Responses**

Dear Reviewers,

Thank you for your reviews and responses to authors replies. Do you have any further comments or questions, and if you have changed your opinion, have you updated your scores?

Kind regards,
AC

---

### Decision · Program_Chairs · 2023-01-20

**Decision:**

Accept: poster

**Justification For Why Not Higher Score:**

It would be good if method worked for full network (future work).

**Justification For Why Not Lower Score:**

Good results and interesting paper.

**Metareview: Summary, Strengths And Weaknesses:**

This paper proposes a new architecture for continual learning related to sparse distributed memory. It shows that the model works through theoretical analysis and experimental evaluation, relates it to biology and provides good insights.

Some drawbacks (potential future work) include the fact that it does not train the full network but pre-trains convent in a standard way on ImageNet and trains a network that corresponds to one layer MLP on the top of the features. I believe the authors are in the process of extending this but haven't gotten good results yet.

**Note From Pc:**

if the above contains the word "oral" or "spotlight" please see: "oral" presentation means -> notable-top-5% and "spotlight" means -> notable-top-25%. As stated in our emails, we are disassociating presentation type from AC recommendations

**Summary Of Ac-Reviewer Meeting:**

It was initially borderline, but through enough exchanges the paper became accept.